

# Medusa-Aqua system: simultaneous measurement and evaluation of novel potential halogenated transient tracers HCFCs, HFCs and PFCs in the ocean

Pingyang Li[1], Toste Tanhua[1]

[1] Marine Biogeochemistry, GEOMAR Helmholtz Centre for Ocean Research Kiel, Kiel 24105, Germany

*Correspondence to*: Toste Tanhua (ttanhua@geomar.de)

**Highlights**

- A method to evaluate the utility of a compound to be an oceanic transient tracer is provided.
- The Medusa-Aqua system to simultaneous measure selected CFCs, HCFCs, HFCs and PFCs in seawater is described.
- HCFC-22, HCFC-141b, HCFC-142b, HFC-134a and HFC-125 were measured in the Mediterranean Sea for the first time.
- HCFC-141b was found to be the most possible oceanic transient tracers.
- PFC-14 and PFC-116 were found to be the most potential oceanic transient tracers.

**Abstract**

This study evaluates the potential usefulness of the halogenated compounds HCFC-22, HCFC-141b, HCFC-142b, HFC-134a, HFC-125, HFC-23, PFC-14 and PFC-116 as the time-dependent oceanographic transient tracers in order to better constrain ocean ventilation processes. We collected seawater samples and improved on an established analytical technique, the Medusa-Aqua system, to simultaneous measure them, and estimate their stability in seawater following previous work on the atmospheric history and solubility. HCFC-22, HCFC-141b, HCFC-142b, HFC-134a and HFC-125 have been measured in profiles in the Mediterranean Sea for the first time. We estimated the historic surface saturation anomalies of transient tracers in the Mediterranean Sea by evaluating the historic record. Their stability in seawater was estimated by analysis of their ocean partial lifetimes, seawater surface saturations and concentrations compared to CFC-12 measurements by a well-established technique. Of the investigated compounds, HCFC-141b was found to be the most promising transient tracer in the ocean; it fulfills several essential requirements by virtue of well-documented atmospheric history, established seawater solubility, inertness in seawater and feasible measurements and indication of conservative behavior in seawater by having mean ages in agreement to be expected from both CFC-12 and SF$_6$ observations. However, more information on degradation is needed to further identify its stability in seawater, and HCFC-141b has restrictions on production and consumption imposed by the Montreal Protocol leading to its decreasing atmospheric mole fractions since 2017. The most potential oceanic transient tracers were PFC-14 and PFC-116 due to their stability in seawater, the long and well-documented atmospheric concentrations histories and constructed seawater solubility functions, although the low solubility in seawater creates challenging measurement conditions (i.e. low concentration). Measurements of PFCs can be potentially improved by modifying the Medusa-Aqua analytical system. With the exception of





providing the information on the novel potential alternative oceanic transient tracers, this study also provides a method on how to evaluate the feasibility for a compound to be a transient tracer in the ocean.

## 1 Introduction

### 1.1 Why do we look for new transient tracers?

Transient tracers consist of chronological transient tracers, such as dichlorodifluoromethane (CFC-12) and sulfur hexafluoride ($SF_6$), and radioactive transient tracers, such as Tritium ($^3H$), Argon-39 ($^{39}Ar$) and Carbon-14 ($^{14}C$). Dissolved CFC-12 is widely used as an oceanic transient tracer to study the oceanic ventilation, mixing and circulation processes since the 1980s, $SF_6$ since the 1990s, as they are stable in seawater; their seawater solubility functions are well-established (Warner and Weiss, 1985; Bullister et al., 2002), as are their atmospheric history

concentrations over time (Walker et al., 2000; Bullister, 2015). However, the use of CFC-12 was phased out as a result of the implementation of the Montreal Protocol (MP) on Substances that Deplete the Ozone Layer and subsequent amendments designed to curtail the degradation of the Earth's ozone layer. Therefore, the atmospheric concentration of CFC-12 has been decreasing since the early 2000s (Bullister, 2015), which reduced its usefulness to be an oceanographic transient tracer. Although CFC-12 is valuable for deep water transport, its use for recently

ventilated water masses is limited. Since the last decade, sulfur hexafluoride ($SF_6$) has been added to the suite of oceanic transient tracers (Tanhua et al., 2004; Bullister et al., 2006) as it is an inert gas whose atmospheric abundance is increasing. However, some local restrictions were in place for the production and use of $SF_6$ due to its very high global warming potential of around 23 500 over a 100-year time horizon (Carpenter et al., 2014). This may constrain $SF_6$ to be an oceanic tracer in the future. The radioactive isotope $^{39}Ar$ is in many ways an ideal tracer for

ocean circulation for older water masses, but its use has been impeded by difficult analytics. However, recent technological advancements have increased the feasibility of oceanic $^{39}Ar$ observations (Lu et al., 2014; Ebser et al., 2018). Since a combination of multiple transient tracers is needed to constrain ocean ventilation based on transit time distribution (TTD) model (Waugh et al., 2002; Stöven and Tanhua, 2014; 2015), it's necessary to explore novel transient tracers with positive growth in the atmosphere for better understanding ventilation and mixing processes in

the ocean.

There are a few general requirements for a transient tracer: 1) non-reactive and stable in seawater, 2) no (or well known) natural background, 3) known input history (including known solubility in seawater), 4) feasible measurement techniques and 5) large dynamic range. In this paper, we mainly address points 1 and 4 for a few compounds described below. In previous work, we focused on points 2, 3 and 5 (Li et al., 2019).

### 1.2 Potential alternative transient tracers

Based on the discussion by Li et al. (2019), potential alternative oceanographic transient tracers are hydrochlorofluorocarbons (HCFCs) such as HCFC-22, HCFC-141b and HCFC-142b, hydrofluorocarbons (HFCs) such as HFC-134a, HFC-125 and HFC-23 and perfluorocarbons (PFCs) such as PFC-14 ($CF_4$) and PFC-116. As the replacements of CFCs, the atmospheric abundances of most selected HCFCs and HFCs are rising. PFCs are

35 increasing in the atmospheric and have estimated atmospheric lifetimes over 10 000 years. Here we name the





potential chronological transient tracers HCFC-22, HCFC-141b, HCFC-142b, HFC-134a, HFC-125, HFC-23, PFC-14 and PFC-116 as "Medusa tracers", CFC-12 and $SF_6$ as traditional chronological transient tracers and $^3H$, $^{39}Ar$ and $^{14}C$ as radioactive transient tracers. Also CFC-11, CFC-113 and $CCl_4$ have been extensively used as the transient tracers, but have largely been discarded. CFC-11 was found to be degraded in anoxic marine waters (Bullister and

Lee, 1995) and has a time-history similar to that of CFC-12. Besides, the simultaneous measurement of $SF_6$ and CFC-11 is complicated. CFC-113 was found to be lost in warm upper waters (Roether et al., 2001) and $CCl_4$ was found to hydrolysis in warm waters and low oxygen regions (Wallace and Krysell, 1989; Huhn et al., 2001). For the radioactive transient tracers, the half-lives of the three tracer nuclides have different orders of magnitude, allowing them to cover a wide range of ages ("seawater timescales", Fig. 1). However, with the constraints of the weak signal

of $^3H$ and the decreasing atmospheric history of CFC-12, only $SF_6$ can be the relatively reliable transient tracer in the timescale range of 1-100 years despite local restrictions were in place. Fortunately, the different atmospheric histories of the potential transient tracers (Li et al., 2019) allow us to find one or several compounds to replace or supplement the established transient tracers.

### 1.3 Stability of alternative tracers in seawater

Chemical reactions (including hydrolysis process), adsorption to particles and biological degradation process should be considered for the stability of compounds in seawater. PFCs have atmospheric lifetimes on the order of ten thousands of years, that is, > 50 000 and > 10 000 years for PFC-14 and PFC-116, respectively. For PFC-14 ($CF_4$), it is thought to be stable and inert in the ocean (Ravishankara et al., 1993; Cicerone, 1979). First, $CF_4$ is stable at temperatures of at least 1200°C. Second, the rate of hydrolysis of $CF_4$ is immeasurably small. Thirdly, no known

marine natural product contains C-F bonds. Last but not least, no indications of biological processes that can break C-F bonds in $CF_4$.

As for other compounds discussed in this study, we are not aware of any publications directly discuss their stability in seawater. Therefore their stability is presented from several other perspectives that we could find from previous studies. One is the contribution of the oceanic partial lifetime to the total lifetimes. Considering the total fraction of a

compound in the ocean (as compared to the atmosphere), a small loss in the ocean is insignificant for the overall budget of the compound, but can still be important for a potential transient tracer. However, this is still discussed here as the background information for stability. Another one is the comparison of surface seawater saturations between HCFCs and $CCl_4$, a compound that has been identified to be an unstable tracer in the ocean. The other one is the biodegradation of compounds in freshwater or soil as no information was found in seawater. Though a compound

can be degraded in the freshwater or soil but maybe not in seawater, for instance, CFC-12.

As far as we know from previous studies (Yvon-Lewis and Butler, 2002; Carpenter et al., 2014), HCFCs and HFCs are relatively stable in seawater as their ocean partial lifetimes (partial atmospheric lifetimes with respect to oceanic uptake) ranged from thousands to millions of years. The estimated oceanic sinks yield lifetimes of 1 174, 9 190, 122 200, 5 909 and 10 650 years for HCFC-22, HCFC-141b, HCFC-142b, HFC-134a and HFC-125, respectively (Yvon-

Lewis and Butler, 2002). Judged against the environmental total lifetimes of 11.9, 9.4, 18, 14 and 31 years (SPARC, 2013), the ocean contributes approximately 1 %, 0.1 %, 0.01 %, 0.2 % and 0.3 % of the total environmental loss rates of these compounds based on the calculation method in Huhn et al. (2001). The ocean contributions are small enough



to be neglected. Note though that due to the low solubility of these compounds shown in Table 5 in Li et al. (2019), only a small fraction of the compounds enter the ocean in the first place. Sum up all the above discussions, the ocean partial lifetimes of HCFCs and HFCs with respect to hydrolysis in seawater are very long (Yvon-Lewis and Butler, 2002). This means that oceanic chemical degradation processes alone are possibly not significant sinks for selected

HCFCs and HFCs, although further research on this is needed.

The stability of HCFCs in the ocean is also suggested by their observed saturation anomalies from the equilibrium between the surface ocean and atmosphere (ftp://ftp.cmdl.noaa.gov/hats/ocean/, last access: 13 September 2019) based on the comparison with those for CFC-11, CFC-12 and $CCl_4$ (Butler et al., 2016). Seawater surface saturations of HCFCs are not as under-saturated as that of $CCl_4$ by the comparison of their saturations in various oceans based

on the results of the National Oceanic and Atmospheric Administration (NOAA) cruises in 1992-2008. This suggests that HCFCs are more stable than $CCl_4$ and possibly well suited to be a tracer in the ocean.

Based on these discussions, HCFCs seem to be relatively stable in the ocean considering only chemical degradation process and surface seawater saturation. However, oceanic uptakes for these compounds mentioned above were calculated without considering the oxygen dependence and biological degradation processes as spatial and temporal

distributions of these degradation processes were not investigated (Yvon-Lewis and Butler, 2002). Although Chang and Criddle (1995), Oremland (1996) and Streger et al. (1999) observed the aerobic bacterial degradations of selected HCFCs and HFC-134a in very high oxygen concentrations and substrate levels, these aerobic microorganisms are common inhabitants of soil and aquatic systems (Table 1). The lifetime of a compound in soil or freshwater can be considerably shorter than the one in open ocean waters with few particles, although rapid removal

in the soil can be an indication of non-conservative behavior in the ocean. Considering all the above discussions, not enough information is known on the stability of the selected HCFCs and HFCs in the ocean. More studies on their stability should be done.

### 1.4 Purpose and structure of this study

In order to explore if the target halogenated compounds can be used as oceanic alternative transient tracers, their

atmospheric histories and seawater solubility have already been reported by Li et al. (2019). This study is an extension of that work, with the main focus on the evaluation of the usefulness of these halogenated compounds as the time-dependent oceanographic transient tracers. We would discuss if these compounds are conservative in the oceanic environment and are capable of rapid, relatively inexpensive and accurate measurements based on observations of these tracers in the Mediterranean Sea.

The structure of this study is organized as follows. The next section introduces ocean ventilation and a one-dimensional inverse Gaussian transit time distribution (IG-TTD) model, combined with more information on time range, tracer age, mean age and TTD elaborated from the atmospheric histories of selected HCFCs, HFCs and PFCs. Section 3 overviews the Medusa-Aqua system. Section 4 describes the sampling and measurement. Section 5 discusses the historical and current seawater surface saturations, stabilities and comparison of estimated mean ages

of selected HCFCs and HFCs based on observations. The usefulness of all the alternative transient tracers was evaluated and discussed in Section 6. The last section provides the main conclusions.



## 2 Transient tracer interpreting methods

### 2.1 Ocean ventilation and transit time distribution (TTD) model

Ocean ventilation and mixing processes play a significant role in the climate change as they are the most prominent processes to propagate perturbations on the ocean surface to the interior, largely controlling the accumulative uptake

of anthropogenic carbon ($C_{ant}$) at mid- and high latitudes and oxygen supply. In order to quantitatively describe these processes, we introduce a conceptual but well-established ocean ventilation model.

The Transit Time Distribution (TTD) model is based on the Green's function described the propagation of tracer boundary conditions into the interior (Hall and Plumb, 1994). As shown in Eq. (1), $G(t,r)$ is the Green's function and $c(t_s,r)$ is the concentration of a transient tracer at year $t_s$ and location $r$. The boundary concentration $c_0(t_s,r)$ is

the concentration at source year ($t_s - t$) related to the input function of a tracer. The exponential term ($e^{-\lambda t}$) is the decay rate of radioactive transient tracers. This function is based on a steady and one-dimensional flow model with time-invariant advective velocity and diffusivity gradient. One commonly used solution to Eq. (1) is the one-dimensional inverse Gaussian transit time distribution (IG-TTD), simplified and expressed as Eq. (2). $G(t)$ is defined based on the mean age $\Gamma$, the width of the distribution $\Delta$ and the time range $t$ (Waugh et al., 2003). In the case of IG-

TTD, the location $r$ is immaterial because the distribution is compiled for every distinct sampling site of a transient tracer (Stöven et al., 2015).

$$c(t_s,r) = \int_0^\infty c_0(t_s - t)e^{-\lambda t} \cdot G(t,r)dt \qquad (1)$$

$$G(t) = \sqrt{\frac{\Gamma^3}{4\pi\Delta^2 t^3}} \cdot \exp(\frac{-\Gamma(t-\Gamma)^2}{4\Delta^2 t}) \qquad (2)$$

The $\Delta/\Gamma$ ratio (0.0-1.8) of the TTD corresponds to the proportion of advective transport and eddy-diffusive characteristics of the mixing processes for a water parcel; the higher $\Delta/\Gamma$ ratio (1.2-1.8), the more dominant the diffusion; the lower $\Delta/\Gamma$ ratio (0.4-0.8), the more dominant the advection.

### 2.2 Time range, tracer age, mean age and Transient Time Distribution

**Time range.** The time range where a tracer can be used as a transient tracer is defined by its input function. Atmospheric histories of tracers should be monotonic increasing in the atmosphere for proper applicability. For chronological transient tracers, the input functions are described by their atmospheric histories and seawater surface saturations. Figure 2 shows the atmospheric histories of HCFC-22, HCFC-141b, HCFC-142b, HFC-134a, HFC-125,

HFC-23, PFC-14, PFC-116, CFC-12 and SF$_6$ in the Northern Hemisphere (Bullister, 2015; Li et al., 2019). Except for CFC-12, HCFC-141b and HCFC-142b, the concentrations of the potential tracers are still increasing.

**Tracer age.** Water mass ages are defined as the elapsed time since a water parcel left the mixed layer into the interior ocean. Tracer age has been used to estimate the age of a water parcel based on a purely advective flow in the ocean. Its $\Delta/\Gamma$ equals zero if put in the IG-TTD concept.

Related to possible age information, each tracer has a specific time and application range. Figure 3 shows the relation between the relative tracer concentrations in percent and the corresponding tracer ages with the reference year 2018,





2010 and 2000, which highlights tracer similarities and the specific application range for each tracer. Relatively similar time trends are found for the following couples: HCFC-141b and HCFC-142b, HFC-134a and HFC-125, SF$_6$ and HFC-23, PFC-14 and PFC-116. Assuming that all these compounds fulfill the other criteria to be a transient tracer, one of each couple could be chosen for further studies depending on their relative tracer concentrations.

In Fig. 3, the relative tracer concentrations, i.e. normalized to the contemporaneous atmospheric concentrations, for 11 transient tracers are shown for three different sampling years. If the relative tracer concentrations are over 100 % then there has been a decrease in atmospheric concentrations. For instance, the tracer age range is 0-30 years for CFC-12 in the reference year 2018 (Fig. 3a), which was produced by the decreasing atmospheric mole fractions (see Fig. 2). When the atmospheric history of a compound is not monotonically changing, the equilibrium atmospheric

mole fraction (and ultimately the age associated with that mole fraction) calculated from its concentration in the ocean is not unique, reducing its potential as a transient tracer (Li et al., 2019). Therefore, the tracer age ranges are dependent on the sampling years for the chronological tracers. From Fig. 3ac, the useful tracer age ranges of CFC-12 are 30-80 years and 1-60 years for sampling in 2018 and 2000, respectively. This indicates that the ability of CFC-12 to be a transient tracer is decreasing with time. Similarly, CFC-12 is not suitable to be a transient tracer for "young"

waters due to the limit of decreasing atmospheric concentrations but still provide important time information for intermediate and deep water layers with moderate ventilation timescales.

It's worth pointing out that PFCs have longer tracer ages compared to other compounds, even CFC-12, but shorter than that of CCl$_4$ (Fig. 3) among the chronological transient tracers. As CCl$_4$ is not used frequently as a transient tracer any more due to its non-conservative behavior in the ocean (Huhn et al., 2001) and CFC-12 was limited to be

used as a tracer in the upper ocean, PFCs will obtain more attention if they are evaluated to be transient tracers in the ocean. The specific application ranges of tracer ages for other tracers can be found in Fig. 3 with the compiled results shown in Fig. 1.

**Mean age and Transient Time Distribution (TTD).** When $\Delta/\Gamma > 0$ and ranges in 0.2-1.8, the calculated age is mean age considered both advection and mixing processes in the ocean. The input functions of chronological

transient tracers can be used to calculate theoretical tracer concentrations $c(t_s, r)$ for a range of $\Delta/\Gamma$ ratios (0.2-1.8) and a mean age spectrum between 1 and 1500 years (see Eq. 1 and Fig. 4). The input function is the combination of the atmospheric concentrations and the seawater surface saturations during water mass formation, in Fig. 4 we have assumed 100% saturation and the atmospheric histories from Li et al. (2019). Observed concentrations, $c(t_s)$, yield different mean ages for different assumptions of the $\Delta/\Gamma$ ratios for the IG-TTD based on the method presented in

Stöven et al. (2015). That is, the mean age (i.e. the TTD) of a water parcel can be determined depending on the assumed or determined $\Delta/\Gamma$ ratios (assuming an IG-TTD) and measured tracer concentrations in seawater. A $\Delta/\Gamma$ ratio of 1.0 is commonly applied for the mean age calculation. Figure 4 can be compared with the corresponding figures for the traditional transient tracers, as reported in Stöven et al. (2015).

Figure 5 shows the mean age matrices of the blue lines ($\Delta/\Gamma = 1.0$) in Fig. 4 for each compound and describes tracer

concentration distributions depending on different mean ages and sampling years. That is to say, a number of characteristics, such as mean age, mode age and width of the distribution, of a water parcel, can be calculated based on the sampling year and measured tracer concentrations in seawater on the assumption of a certain TTD. By




combining the information in Fig. 4 and Fig. 5, the mean age lookup tables of the IG-TTD for transient tracers can be constructed by considering sampling years, measured tracer concentrations in seawater and $\Delta/\Gamma$ ratios.

As seen in Fig. 5, the mean age distributions of the IG-TTD for the tracers are different depending on their atmospheric concentration histories (Fig. 2), part of the input functions. Such a suite of transient tracers with

5 sufficiently different input functions could support the empiric determination of more complex TTDs, as reported in Stöven and Tanhua (2014). As the first step, these Medusa tracers have been measured, sometimes for the first time, in this study and interpreted based on the IG-TTD concept to identify their possibility to be transient tracers in the ocean.

### 3 Medusa-Aqua system

**3.1 Progress in analytical technology of selected HCFCs, HFCs and PFCs**

Measurement of halogenated compounds is normally done by a "gas-solvent extraction" technique, i.e. purge-and-trap, where a clean gas is bubbled through a seawater sample to move the analytes from the sample into a cold trap for concentrated. By desorbing the content of the trap, the sample can then be injected into a gas chromatograph (GC) and an electron capture detector (ECD) for separation and detection. Sometimes, a pre-column and a

15 reconcentration trap are used to improve the analytical performance of the system. These are all well-established techniques that have been used successfully for CFCs and $SF_6$ (Bullister and Weiss, 1988; Bullister and Wisegarver, 2008) and where accuracies in the order of 1 % can be achieved (Bullister and Tanhua, 2010). However, several HCFCs and HFCs (i.e. HCFC-22, HFC-134a and HFC-125) have low responses and large uncertainties when they are measured by the ECD (Lobert et al., 1995; Beyer et al., 2014). Besides, the HCFCs tend to be more soluble,

making it more difficult to quantitatively purge all of the tracers from a water sample. One alternative is to use a mass spectrometer (MS) for detection. This has the advantage of scanning on unique masses for different compounds, i.e. quantification and identification simultaneously. MS as the detector is becoming increasingly popular because the sensitivity is approaching that of ECD. Previous studies also reported that CFCs, HCFCs and HFC-134a can be measured by GC-MS (Lobert et al., 1996; Yvon-Lewis et al., 2008; Ooki and Yokouchi, 2011).

Medusa-GC-MS system, shorted as Medusa system, for precision and simultaneous analysis of a wide range of volatile trace gases has been developed at Scripps Institution of Oceanography (Miller et al., 2008). This system is based on trapping of the volatile gases on two traps kept at accurately controlled temperatures. The packing material of the traps and the temperature during the trapping stage are designed in a way that allows fractionation of the compounds on two traps. In this way, interferences from atmospheric permanent gases can be avoided and hard-to-

measure gases like PFC-14 ($CF_4$) can be measured. This analytical system was designed to automatically and continuously measure air samples at the Advanced Global Atmospheric Gases Experiment (AGAGE) remote field stations (Prinn et al., 2018) and is unique in that it provides high accuracy measurements of more than 40 compounds including halocarbons, hydrocarbons and sulfur compounds involved in ozone depletion and/or climate forcing from the same sample. The measurement precisions of the majority of the halogenated compounds are less than 0.5 % in 2

L ambient air. Based on the Medusa system, the Medusa-Aqua system was developed to measure halogenated compounds in seawater samples. In this study, the Medusa-Aqua system can measure the majority of the 40





compounds in seawater samples. Target HCFCs, HFCs and PFCs are chosen for quantification and assessment of their potential as alternative oceanic transient tracers.

### 3.2 Difference between Medusa-Aqua and Medusa system

Medusa-Aqua system consists of a Medusa system (Miller et al., 2008) and a seawater sample pretreatment system (Fig. 6). The Medusa system comprises a cryogenic pre-concentration unit, named Medusa, and an Agilent 6890N gas chromatograph (GC) and an Agilent 5975B quadrupole mass spectrometer (MS). The seawater sample pretreatment system was developed to degas the samples from gaseous tracers before injecting into the Medusa system, replacing the air sampling device of the original Medusa system. The technology is based on a purge-and-trap technology where Medusa serves as the trap unit and chromatographically separates the sample for detection in MS. This seawater sample pretreatment system for purge, named Ampoule-Cracker-System, is used for off-line seawater samples. The seawater samples were sampled and flame-sealed in ~1.3 L ampoules at sea and measured in the laboratory. The Ampoule-Cracker-System, as designed by Vollmer and Weiss (2002) and modified by Stöven (2011), has been used to extract CFC-12, $SF_6$ and $SF_5CF_3$ from seawater samples in the laboratory in Kiel for over five years and is a mature technique (Stöven, 2011; Stöven and Tanhua, 2014). The system was used in this study to work for the Medusa-Aqua system.

The main difference between Medusa and Medusa-Aqua system is that the former used an air pump module as the gas sample pretreatment system and the sample volume is determined by an integrating mass flow controller (MFC), while the latter used a seawater purge module as the seawater sample pretreatment system and a gravimetrically calibrated standard loop to qualify standard gases. Another difference comes from data processing. The integrated peak areas or heights of compounds measured by the Medusa system are directly processed by the GCWerks, a custom Linux-based software which used to control the hardware, while the peak area or height counts measured by the Medusa-Aqua system are exported to MATLAB (R2016b) for data processing.

## 4 Sampling and Measurement

Here we describe the results where we sampled for the Medusa-Aqua system from mainly one cruise (MSM72) to the Mediterranean Sea in 2018 to demonstrate the method. Over the past years, we have collected samples on a few cruises (MSM18/3, MSM23, M130, NORC2017-09, KBP523, KBP 524 and Al516) and empirically improved our method.

### 4.1 Sample collection

Seawater samples were collected on three areas (Fig. 7, Table S1), Southern Ionian Sea (SIS), Tyrrhenian Sea (TS) and Western Mediterranean Sea (WMS), on the cruise MSM72 by the research vessel *Maria S. Merian* in the Mediterranean Sea from March 2$^{nd}$ to April 3$^{rd}$, 2018 along the GO-SHIP line MED-01 (Hainbucher et al., 2019). These seawater samples were collected in glass ampoules by connected to a Niskin bottle of CTD (conductivity, temperature, and depth) via a stainless steel mounting system (Vollmer and Weiss, 2002). Around 5 min is spent for the seawater to fill up a whole glass ampoule. The sampling process lasted 15 min for the seawater to flush the whole





ampoule volume three times. After removing and closing the ampoule with a screw, the ampoule was flame-sealed as soon as possible under a flow of high purity $N_2$ (grade 6.0) and then sent back to the laboratory in Kiel for measurement. As seen in Fig. 7, three sampling stations (in red) were selected for HCFCs, HFCs, PFCs and CFC-12 measured by the Medusa-Aqua system and blue dots mark stations where CFC-12 and $SF_6$ were measured onboard

by the PT-GC-ECD. No on-board CFC-12 and $SF_6$ measurements were conducted on the stations we sampled for the Medusa-Aqua system. Instead, we interpolated (on density surfaces) the CFC12 and $SF_6$ concentrations to the Medusa sampling sites based on adjacent stations. The distance between stations on this cruise was 15 nm (nautical miles), and normally we sampled for chemistry on every second station.

### 4.2 Gas extraction, separation and detection

The flow scheme for the Medusa-Aqua system is shown in Fig. 6. Before measurement, each ampoule sample was immersed in a warm water bath at 65 °C to enhance the purging efficiency by driving the gases into the headspace. The stem of the ampoule is inserted vertically up into the cracking chamber and is held by a screw-nut with nylon ferrule. Then the cracking chamber is flushed by $N_2$ for 10 min to remove all impurities. A blank test for the cracking chamber is made by simulating an extraction without breaking the glass ampoule. For analysis, the tip of the

ampoule's stem is shattered into pieces inside the enclosed cracking chamber by rotating the cracking paddle. A straight purge tube is then inserted down into the ampoule until touching the ampoule bottom for finer bubbles. These bubbles will help strip the compounds out of the seawater and enhance the purge efficiencies for the dissolved gases.

The extraction process is started by purging the gases in the ampoule with $N_2$ (grade 6.0) for 20 min at a flow rate of

100 ml min$^{-1}$. The purge time is the same as it is for standard gas and air injections. The extracted gas is then introduced into Medusa via port no. 1 of V1. Two Nafion dryer of 1.8 m length and one Nafion dryer of 0.6 m length are used to remove water vapor from the samples. The flow rate of Nafion dryer gas ($N_2$, grade 5.0) was set to 120 ml min$^{-1}$. After flushing the gaseous samples into Medusa, the following path is the same as described by Miller et al. (2008). The gases will experience separation and be detected by MS. The carrier gas for MS is Helium (grade 6.0).

The chromatograms for seawater samples are similar to those for air samples in Miller et al. (2008) but with smaller peak areas and heights.

### 4.3 Weighting the ampoule

The cracked ampoules (with glass sprinters) were weighed right after removing them from the cracking device. Then they were emptied and rinsed with distilled water. The glass splinters were sorted back into the ampoules. The

ampoule, as well as glass splinters, was together put into an electro-thermal drying oven for around 96 h. Finally, the ampoules were weighed again to quantify the masses of the seawater samples.

### 4.4 Standard and calibration

The standard gas used in the laboratory in Kiel is a tertiary standard from the AGAGE relative scale "SIO-R1". The tertiary standard is calibrated by the secondary standard and propagated to the primary calibration from the central

calibration facility at the Scripps Institution of Oceanography (SIO). More details about the propagation of the



standard see Miller et al. (2008). Gravimetric calibration scales and calibrated errors of compounds in the tertiary standard are reported in Table 2. Measurements in seawater will be reported on the latest SIO absolute calibration scales for HFC-125 (SIO-14), HFC-23 and PF-114 (SIO-07) and other compounds (SIO-05). The calibrated errors of all compounds in standard gas are below 0.5 %.

The tertiary calibration scale is then propagated to a working standard. This working standard is used to determine calibration curves for the MS responses within the range of compounds concentrations measured. These calibration measurements are made by multiple injections of a 10.0 ml gravimetrically calibrated sample loop. Each injection lasts 40 s at a flow rate of 50-60 ml min$^{-1}$. The calibration curve was made once per week. Detector responses for compounds in Table 2 are relatively linear in the range of our measurements. During the analysis of seawater

samples, three standard measurements at the same volume at ambient pressure are done once a day to correct the short-term detector response fluctuations. After applying a three standard deviation recursive filter, concentrations of measured compounds were corrected.

## 4.5 Purge efficiency, detection limit and precision

Considering the instability of purge efficiencies, each sample was measured 3-6 times until none of the compounds

in Table 2 could be detected in the seawater sample. That is, the purge efficiencies for all compounds are 100 %. The detection limits for measurements of all compounds by the Medusa-Aqua system are shown in Table 2 based on the signals corresponding to the blank values or noises plus ten standard deviations.

The precisions of the measurement are dependent on the size of ampoules (i.e. sampling size) and sampling concentrations; they tend to be higher for larger samples and shallow samples (i.e. high concentration). The samples

with a higher amount of tracer have better precisions than those with a lower amount. The precision (or reproducibility) for seawater sample measurements were determined by the relative standard deviations of the concentrations for two pairs of duplicate samples (Table 2).

The concentrations of SF$_6$, PFC-14 and PFC-116 in most seawater samples are lower than the detection limit, and HFC-23 has an unstable non-zero blank in blank, standard and seawater measurements, preventing us from

evaluating those results. Observations of CFC-12, HCFC-22, HCFC-141b, HCFC-142b, HFC-134a and HFC-125 measured by the Medusa-Aqua system in seawater from the cruise MSM72 are shown in Table S1 and discussed in the following sections.

## 4.6 Comparison of instruments measuring CFC-12

In order to explore the accuracy of seawater measurements by the Medusa-Aqua system, CFC-12 was measured by

both the Medusa-Aqua system and a purge and trap GC-ECD instrument (Syringe-PT-GC-ECD) used onboard the cruise MSM72. This is a mature system to measure CFC-12, SF$_6$ and SF$_5$CF$_3$ (Stöven, 2011; Stöven and Tanhua, 2014; Stöven et al., 2016; Bullister and Wisegarver, 2008). For comparison, a similar purge and trap system set-up (Cracker-PT-GC-ECD) to measure flame sealed ampoules is added. A detailed comparison of the three instruments is shown in Table 3. Compared to other systems, the Medusa-Aqua system has lower purge efficiency due to using a

bigger sampling volume if only considering purge once (although we used multiple purge cycles to increase the





effective purge efficiency and reduce the uncertainty); has lower precision than that of the Syringe-PT-GC-ECD as well as Cracker-PT-GC-ECD system but can measure more compounds.

## 5 Results

### 5.1 Historical seawater saturation in the Mediterranean Sea

As mentioned in Sect. 2.2, input functions are described by the combination of atmospheric histories and seawater surface saturations for chronological transient tracers. The atmospheric histories of selected HCFCs, HFCs and PFCs have been discussed in Li et al. (2019). In order to obtain the complete input functions, the seawater saturation of compounds in the Mediterranean Sea is discussed in this section.

Seawater saturations in the winter mixed layer was calculated from historical cruises data in the Mediterranean Sea
to determine the historical seawater saturations. The depth of winter mixed layers in summer and winter are shown in Fig. 8. The seawater saturation in the warm surface tends to be higher than the values during winter, which are the relevant saturation levels for deep and intermediate water formation, and thus for the input functions. Therefore, the relative warm layer was not considered in the calculation of historic seawater saturation in the winter mixed layer for summer cruises. The depth ranges of winter mixed layer for each historical cruise from 1987 to 2018 from Schneider
et al. (2014) and Li and Tanhua (in preparation) were commonly determined by profiles of temperature, potential density and CFC-12 concentrations shown in Fig. S1.

Seawater saturations of CFC-12 and $SF_6$ in winter mixed layers from each historical cruise are shown in Fig. 9. Historical seawater saturation of CFC-12 and $SF_6$ in the Mediterranean Sea are determined to be $94 \pm 6$ % and $94 \pm 4$ %, respectively, by averaging the average seawater saturation from every cruise in their specific winter mixed layer
(Fig. S1). Neither the historical seawater saturation of CFC-12 or $SF_6$ does show a clear trend over time. For CFC-12, this is different to the situation in the North Atlantic Ocean (Tanhua et al., 2008), and is probably an indication of the different oceanographic setting where the inflowing Atlantic Water (to the Mediterranean Sea) has a long time to equilibrate with the atmosphere.

For the following calculation, the historical seawater saturations are assumed to be a constant 94 % (over time) for
all transient tracers in this study as little data exists to determine the historical seawater saturation of selected HCFCs and HFCs in the Mediterranean Sea (see Sect. 5.3). The historical seawater saturation and the atmospheric concentration histories of all compounds together describe their input functions.

### 5.2 Observations

The observations of CFC-12 measured by the Medusa-Aqua system are generally comparable with those in adjacent
stations measured by the PT-GC-ECD onboard (Fig. 10). The averaged misfit of CFC-12 concentrations measured by the two different instruments is $5.9 \pm 4.6$ % focusing on only the data with quality flagged as "good" (blue dots in Fig. 10). Based on the reasonable correlation between CFC-12 observations from the Medusa-Aqua system and the onboard PT-GC-ECD system, we can move on and interpret the profiles of the Medusa-only compounds. Figure 11 shows the observations of CFC-12 and $SF_6$ from profiles 51, 83 and 105 measured by the PT-GC-ECD and
observations of CFC-12, HCFC-22, HCFC-141b, HCFC-142b, HFC-134a and HFC-125 from the nearby profiles 52,



84 and 106 measured by the Medusa-Aqua system. Nearly monotonous decline trends in concentrations are found for all compounds from the surface to the deep ocean but with slightly higher bottom concentrations.

### 5.3 Seawater surface saturation

Seawater surface saturations of $SF_6$ and CFC-12 measured by the PT-GC-ECD in profiles 83 and 105 and CFC-12,

HCFC-22, HCFC-141b, HCFC-142b, HFC-134a and HFC-125 measured by the Medusa-Aqua system in profiles 84 and 106 are shown in Table 4. The averaged saturations of $SF_6$ and CFC-12 measured by the PT-GC-ECD are 94.5 % and 91.5 %, respectively, which are close to the ones estimated from historical seawater saturations (Sect. 5.1). The seawater surface saturation of CFC-12 measured by the Medusa-Aqua system is ~20 % lower than the adjacent ones by the PT-GC-ECD. The averaged surface saturations of HCFC-22, HCFC-141b and HFC-125 measured by the

Medusa-Aqua system are 43 %, 52 % and 37 %, which are lower than the ones expected. There are a few possible reasons for the lower than expected saturations: 1) problems in measurements/calibration; 2) poorly defined solubility functions; 3) degradation in seawater. Note that the samples were transported from the sampling sites in the Mediterranean Sea to Kiel, and were not measured until July, leading to roughly 4 months-time between sampling and measurement. During this time the samples were stored at ~20°C, although temperatures in the

container transport could have been higher occasionally. The averaged saturation of HCFC-142b measured by the Medusa-Aqua system is 90 %, which is comparable to the ones of $SF_6$ and CFC-12. The averaged saturation of HFC-134a measured by the Medusa-Aqua system is 139 %, which is supersaturated.

### 5.4 Stability based on interior ocean observations

In order to do a first-order validation of the stability of HCFCs and HFCs, concentrations of CFC-12 in profiles 52,

84 and 106 are vertically interpolated to match the PT-GC-ECD measurements in each sampling area (Fig. 7). This was done by a piecewise cubic hermite interpolating method on potential density surfaces; the PT-GC-ECD profiles were averaged by the arithmetic mean of the interpolated profiles (Tanhua et al., 2010; Schneider et al., 2014). The concentrations of HCFC-22, HCFC-141b, HCFC-142b, HFC-134a and HFC-125, measured by the Medusa-Aqua system, are plotted against CFC-12 from the PT-GC-ECD instrument and shown in Fig. 12. The thick black line is

the atmospheric history of the tracer pair. Although the atmospheric histories of CFC-12 in each plot are the same, the thick black lines are different due to the different atmospheric histories of HCFC-22, HCFC-141b, HCFC-142b, HFC-134a and HFC-125. The thin black line is the theoretical mixing line between contemporary concentrations and pre-industrial. All samples have to fall between the thin and thick black lines if conservative in seawater. Compounds where the samples below the thick line are not stable (assuming that CFC-12 is stable), and for samples above the

thin mixing line there are issues with too high values (see below). HCFC-22 is found close to the atmospheric CFC-12/HCFC-22 line (samples would fall on this line if there were no mixing but only advection in the ocean); HCFC-141b and HFC-125 are well in the allowed range; HCFC-142b and HFC-134a are around or above the lower-bound line (2018). That is to say, HCFC-141b and HFC-125 are well in the ranges, HCFC-22 is on the lower bound, while HCFC-142b and HFC-134a are mostly higher than the allowed ranges. There are a few possible reasons for too high

concentrations: 1) contamination of the samples during sampling process or measurement in the laboratory; 2)





problem in solubility functions and 3) some other issues within the measurements in the laboratory causing our observations to be high.

For HFC-125, there are three potential seawater solubility functions of HFC-125 reported in the study of Li et al. (2019) depending on the three different freshwater solubility functions. In Fig. S2e in Li et al. (2019), freshwater

solubility curve 1 (MB), 2 (MH) and 3 (RA) are estimated based on data from studies (Miguel et al., 2000; Battino et al., 2011), (Mclinden, 1990; HSDB, 2015) and (Reichl, 1996; Abraham et al., 2001), respectively. Freshwater solubility curve 3 was picked as the likely correct one as only the freshwater solubility data from the two studies are obtained based on measurements. When we think about the solubility curve of HFC-125 in this study, seawater solubility function derived from freshwater solubility curve 1 seems to be the best one to fit the atmospheric and

oceanic concentrations. Thus, a dedicated study to determine the solubility of HFC-125 in seawater needed.

### 5.5 Comparison of mean age estimates

In order to compare the mean age of selected HCFCs and HFCs with the ones of CFC-12 and $SF_6$, the $\Delta/\Gamma$ ratio of IG-TTD is assumed to be 1.0 and the saturation of all traces is assumed to be 94 %, see Sect. 5.1. The TTD of the Mediterranean Sea is complicated by the variable ventilation and the influence of different source regions for interior water, see Stöven and Tanhua (2014). The assumption of a 1IG-TTD with $\Delta/\Gamma$ = 1.0 can certainly be questioned but

serves as an initial assumption to evaluate the new tracers. Input functions (including atmospheric histories and historical seawater saturation), sampling year, measured tracer concentrations in seawater and predefined $\Delta/\Gamma$ ratio are used to calculate the mean age, see Sect. 2.2. The calculated mean ages of the Medusa tracers are shown in Fig. 13. The mean age of HCFC-141b is similar to those of CFC-12 and $SF_6$, but the mean age is higher for HCFC-22 and

significantly lower for the other three tracers (HCFC-142b, HFC-134a and HFC-125). One explanation can be a different (i.e. shorter) atmospheric history that makes them sensitive to uncertainty in the $\Delta/\Gamma$ ratio in a different way. Since the mean age is lower than expected, it means that the concentration is higher than expected, maybe due to uncertainty in the solubility function or analytical error, see above.

However, as we know from the work of Stöven and Tanhua (2014), the TTD in the Mediterranean Sea is rarely a

1IG-TTD so this is only an indication of the utility of the new tracers. The Mediterranean Sea was chosen for this study because of its rapid ventilation, rendering transient tracers to penetrate most of the water column. However, the time-variant ventilation and the contribution of several deep water sources make the TTD concept difficult for the Mediterranean Sea.

### 6 Discussions

The atmospheric histories of the Medusa tracers have been given by Li et al. (2019), who also used indirect methods to estimate the solubility functions. HCFC-22, HCFC-141b, HCFC-142b, HFC-134a and HFC-125 can be measured by the Medusa-Aqua system. Based on the combined results from Li et al. (2019) and this study, the evaluation of the potential of the Medusa tracers to be transient tracers in the ocean is summarized in Table 5 by evaluating the (i.e. the confidence or feasibility of) atmospheric history, solubility, stability and ease of measurement. In this table, we

start with the results of CFC-12 and marked their confidences or feasibilities to be three stars (the highest mark)





because 1) the atmospheric history of CFC-12 is well-documented (Walker et al., 2000; Bullister, 2015); 2) seawater solubility function are well-established (Warner and Weiss, 1985); 3) the stability in warm waters, as well as poorly oxygenated waters, have been proven; 4) CFC-12 has been observed for several decades by mature analytical techniques. Therefore, we took the confidences or feasibilities of the results of CFC-12 as references to separately

evaluate selected HCFCs, HFCs and PFCs in the following.

*HCFC-22*. The atmospheric history is well-established by a combination of the model results and observations (Li et al., 2019), rendering the result of high confidence in Table 5. The seawater solubility function has been constructed by combining a model (CGW) on the experimental freshwater solubility data and another model (poly-parameter linear free-energy relationships, pp-LFERs) on the salting-out coefficients (Li et al., 2019). The results of freshwater

solubility matched the ones published in Deeds (2008) on measurements and the CGW model fitted results in 298-348 K, and the ones published in Abraham et al. (2001) on observations and the pp-LFERs model results at 298 K and 310 K. Thus, our ability to estimate the seawater solubility was marked as two stars due to lacking the experimental seawater solubility empirical data to verify the function. The stability was evaluated by combining analyzing the ocean partial lifetimes and seawater surface saturations in Sect. 1.3 and measurements compared to

CFC-12 in Sect. 5.4. The clustering of HCFC-22 values in the lower range in Fig. 12 could be an indication of slow degradation in warm seawater, which was also supported by the weak hydrolysis of HCFC-12 in tropical and subtropical waters (Lobert et al., 1995). Also, the seawater surface saturation was lower than expected, which may support the instability of HCFC-22. The mean ages of HCFC-22 were found to be higher than those of CFC-12 and $SF_6$, which is an indication of non-conservative behavior, too. Therefore, the "stability marks" for HCFC22 was

determined to be one star only; more measurements should be added for the stability analysis, especially in poorly oxygenated and warm waters. As to measurement, HCFC-22 has been measured in several cruises (Lobert et al., 1996; Yvon-Lewis et al., 2008) by GC-ECD and GC-MS instruments and in this study by the Medusa-Aqua system. Therefore, its measurement was marked as high feasibility. However, the atmospheric history is similar to that of $SF_6$ but different enough to provide some constraints on the TTD. These all indicate that HCFC-22 seems not suitable to

be a potential new transient tracer in the warm ocean, for instance, the Mediterranean Sea, but could be better used in colder high-latitude areas.

*HCFC-141b*. The knowledge of the atmospheric history (Li et al., 2019) was marked as high confidence (Table 5). But the result of seawater solubility was marked as low confidence because the seawater solubility function was constructed for the first time (Li et al., 2019) and the freshwater solubility only matched the ones in Abraham et al.

(2001) at the two temperatures. The low seawater surface saturation may associate with some physical processes (Butler et al., 2016). As for stability, HCFC-141b was identified to be potentially stable in seawater (two stars) since they are in the range of likely concentrations in the interior ocean (Fig. 12), assuming that the solubility function is valid. HCFC-141b has been measured on cruises (Lobert et al., 1996; Yvon-Lewis et al., 2008) and also in this study. Thus three stars were given to its measurement feasibility. The input function of HCFC-141b is different enough

from those of other tracers (Sect. 2.2), and the observed mean ages of HCFC-141b are similar to both CFC-12 and $SF_6$ (Fig. 13). All these analyses indicated the possibility of HCFC-141b to be a new transient tracer.

*HCFC-142b*. High and low confidences (Table 5) were given to the results of the atmospheric history and seawater solubility function for the same reasons as HCFC-141b. Our ability to estimate the stability of HCFC-142b only got





one star as no as some higher than expected concentration points to another issue (see Sect. 5.4) though its seawater surface saturation was similar to the ones of $SF_6$ and CFC-12. More investigations on solubility and stability of HCFC-142b are needed in the future. However, HCFC-142b has been measured in some cruises (Lobert et al., 1996; Yvon-Lewis et al., 2008) and also in this study. Thus its measurement feasibility is high. Although the input function

of HCFC-142b is different from those of most other tracers (only similar to that of HCFC141b), HCFC-142b has currently low limited to be a tracer in the ocean due to little reliable information on the solubility and stability and the lower mean ages than those of CFC-12 and $SF_6$ (Fig. 13).

*HFC-134a*. The confidences of atmospheric history and seawater solubility function (Li et al., 2019) were marked as three stars and two stars (Table 5), respectively. Seawater solubility function was constructed by combining the

CGW model fitted on the experimental freshwater solubility data and the pp-LFERs model estimated the salting-out coefficients (Li et al., 2019). The results of freshwater solubility matched both the observations (Deeds, 2008) and model results (Abraham et al., 2001). Therefore, two stars marked for our ability to estimating seawater solubility due to lacking the completely experimental seawater solubility data to construct the function. Our ability in estimating the stability of HFC-134a only got one star due to higher than the expected seawater surface saturation

and concentration, but also not identified to be unstable, see Sect. 5.3 and 5.4. As to being measured, HFC-134a was measured in Ooki and Yokouchi (2011) by GC-MS and in this study by the Medusa-Aqua system, led to high feasibility. As presented in Li et al. (2019), HFC-134a could be a tracer for "young" waters only when considering its short atmospheric history. Based on all these discussions, HFC-134a has a lower possibility than HCFC-141b but a higher possibility than HCFC-22 to be a tracer because of the lower mean ages than those of CFC-12 and $SF_6$ (Fig.

13) and the higher concentrations than expected in Fig. 12.

*HFC-125*. HFC-125 was found to have a strange trend in relative tracer concentration in the early 1990s (Fig. 3), which only marginally influences its ability to be a tracer. Considering its reconstructed atmospheric history (Li et al., 2019), three stars marked in Table 5 as the confidence. As to solubility, three seawater solubility functions of HFC-125 have been constructed (Li et al., 2019). Curve 1 was supported by the similar results from Deeds (2008) in

freshwater solubility and stability analysis based on measurements against with CFC-12 (Fig. 12). While curve 3 was supported by the observations and model results from Abraham et al. (2001) at the two temperatures. This led the seawater solubility function of HFC-125 still to be unknown and marked as low confidence. If curve 1 is chosen to be the seawater solubility function, HFC-125 was relatively stable based on the measurements against CFC-12 (Fig. 12), though its seawater surface saturation was relatively low. Our ability to estimating stability was marked as one

star because unidentified solubility led to undetermined stability. The feasibility of being measured was marked as medium because HFC-125 was measured for the first time in seawater as we know and we can't compare the results with other studies. Besides, measured HFC-125 in freshwater was inconsistent in different studies indicated by three freshwater solubility functions (Li et al., 2019), which may also lead to the problems of measurements in seawater. The lower mean ages than CFC-12 and $SF_6$ will not support HFC-125 to be a tracer, although HFC-125 could be a

tracer for "young" water when only considering its short atmospheric history. Therefore, all these discussions render HFC-125 to currently low possibility to be a tracer in the ocean due to the confusing information on solubility and stability, possible problems in seawater observations and the lower mean ages than CFC-12 and $SF_6$.





*HFC-23*. HFC-23 could not be reliably quantified due to a non-zero blank (Sect. 4.5), and led to the unknown stability, although its atmospheric history and seawater solubility function have been constructed (Li et al., 2019; Simmonds et al., 2018). Therefore, our ability to estimating stability was marked as one star (Table 5) and the same one star was marked for the feasibility of the measurements. The atmospheric history of HFC-23 was marked as

medium confidence as its atmospheric mole fraction doesn't start from zero (Simmonds et al., 2018) due to limited data. Our ability to estimating the seawater solubility function was marked as two stars with the same reason for HFC-134a. That is, the freshwater solubility function matched with results from Deeds (2008) and Abraham et al. (2001) but the seawater solubility function was not constructed by the experimental data completely. In consequence, unknown stability and inability to be measured can't support HFC-23 to be a tracer in the ocean at this moment.

*PFC-14 and PFC-116*. Atmospheric histories of PFC-14 and PFC-116 have been completely established (Li et al., 2019) and these results were marked as both high confidences. Seawater solubility functions were also constructed (Li et al., 2019). The confidence of seawater solubility was marked as two stars for PFC-14, but its confidence was higher than the ones of other compounds with two stars. This can be explained by that seawater solubility of PFC-14 in 273-328 K matched the previous measurements at 288, 293, 298 and 303 K (Scharlin and Battino, 1995) and

freshwater solubility matched both the previous observations and model results (Clever et al., 2005; Abraham et al., 2001). Low confidence was found for PFC-116 seawater solubility as freshwater solubility only matched the one from Deeds (2008) and lacked support information from Abraham et al. (2001). PFC-14 and PFC-116 are very stable (Sect. 1.3) but can't be easily measured in seawater because of low solubility (Li et al., 2019). This led to three stars of the "stability marks" and one star for ease of measurement (Table 5). The high stability makes PFCs the most

potential transient tracers in the ocean, but also with the large challenge of difficulty to be measured. The long atmospheric histories and tracer ages of PFCs (Sect. 2.2) among the chronological transient tracers increased their potentials to be transient tracers.

## 7 Conclusions

This study, combined with Li et al. (2019), provides a method to identify and evaluate if a compound is suitable to be

used as a transient tracer in the ocean. As the replacements of CFCs, representative HCFCs, HFCs and PFCs were selected to be evaluated. The evaluation considered four aspects: atmospheric history, seawater solubility, stability and the feasibility of measurement. The atmospheric histories, combined with historical seawater saturations, form the input functions. The atmospheric histories have been reconstructed in our last study, and the seawater saturation over time in the Mediterranean Sea was estimated based on the ones from historical cruises data. The historical

seawater saturation of CFC-12 and $SF_6$ in the Mediterranean Sea was determined to be a constant 94%. For CFC-12, this is different from the situation in the North Atlantic. The seawater solubility functions have been constructed by Li et al. (2019) by combining a model on the experimental freshwater solubility data and another model on the salting-out coefficients. But the results from this study identify some questions for HFC-125 so that, seawater solubility functions constructed based on experimental methods are needed. The stability of the tracers is analyzed

from three aspects: ocean partial lifetimes, surface seawater saturations and comparison to the expected concentrations deduced from CFC-12 observations. Measurements of CFC-12 by the Medusa-Aqua system were





compared to the ones measured by the onboard PT-GC-ECD system at adjacent stations. Based on the reasonable correlation between CFC-12 observations from the two systems, we interpreted the observations of the Medusa-only compounds for further analysis of the stability and mean age. We conclude that HCFC-141b, and possibly HFC-125, are probably stable in seawater whereas there are indications of slow degradation of HCFC-22 in warm seawater. We

were not able to estimate the stability of HCFC-142b and HFC-134a, and although not evaluated in this study, it seems that the PFCs are stable in seawater. Further studies on the stability of selected HCFCs and HFCs in seawater will be needed in future work. HCFC-22, HCFC-141b, HCFC-142b, HFC-134a and HFC-125 were successfully measured by the Medusa-Aqua system, a novel analytical technique, in the Mediterranean Sea for the first time, although the measurements of the poorly soluble PFCs could not be quantified with our current analytical system.

By comprehensive evaluation of these four aspects (see Table 5), HCFC-141b was found to be the most promising new oceanic transient tracers for following a few decades. However, there are still some challenges: 1) more information on biodegradation is needed to further identify the stability of HCFC-141b in seawater; 2) HCFC-141b probably will not work as transient tracers more than a few decades considering its restrictions on production and consumption imposed by the Montreal Protocol, and its decreasing atmospheric mole fractions since 2017 (Li et al.,

2019). The most potential transient tracers in the ocean still belong to PFC-14 and PFC-116 because of their high stability in seawater, the long and well-document atmospheric concentrations histories and constructed seawater solubility functions (Li et al., 2019). But the biggest challenge is still how to measure the PFCs accurately due to their low concentrations in seawater. Similarly, the work of Deeds et al. (2008) concludes that PFC-14 ($CF_4$) is a very potential oceanic transient tracer. Possible ways forward are to modify the Medusa system according to Arnold

et al. (2012) to improve the sensitivity for PFC-14 ($CF_4$) and try field measurement by the vacuum-sparge method by Law et al. (1994) to improve the speed of gas extraction.

This study is considered as the first step for exploring the novel alternative transient tracers in the ocean. For the tracers HCFC-22, HFC-134a, HFC-125 and HFC-23, more information on the solubility, stability and the feasibility of measurement should be added to further identify their possibility to be transient tracers in the ocean.

**Data availability**

Cruises data worked for historical seawater saturation of CFC-12 and $SF_6$ in the Mediterranean Sea (Sect. 5.1) are from https://www.nodc.noaa.gov/ocads/oceans/Coastal/Meteor_Med_Sea.html. Observations of CFC-12 and $SF_6$ measured by the PT-GC-ECD and observations of CFC-12, HCFC-22, HCFC-141b, HCFC-142b, HFC-134a and HFC-125 measured by the Medusa-Aqua system in seawater from cruise MSM72 are shown in Table S1.

**Author contributions**

TT conducted the sampling. PL developed the instrument and carried out the measurements. PL interpreted the data and analyzed the results based on the discussion with TT. PL wrote the paper with contributions from TT.



**Competing interests**

The authors declare that they have no conflict of interest.

**Acknowledgments**

We acknowledge the great support by the captain, the crew and the scientists from expeditions MSM18/3, MSM23,
5   M130, NORC2017-09, KBP523, KBP524, MSM72 and Al516, which work for the development of Medusa-Aqua
system. We also thank the captain, the crew and the scientists from NOAA expeditions OAXTC, BLAST1, BLAST2,
BLAST3, GasEx98, RB9906, PHASE1 and GOMECC, which provide the saturation data of HCFCs, CFCs and $CCl_4$.
Special thanks go to Boie Bogner and Tim Steffens for their technical support on instrument Medusa-Aqua system;
Prof. Minggang Cai, Dr. Peng Huang and Weimin Wang for supporting the sampling at sea on cruise NORC2017-09
10   in the Western Pacific Ocean. This research was supported by the GEOMAR Innovationsfonds Technologie-Seed-
Funding (Transient Tracers 70090/03) and the China/Germany Joint Research Programme (Programm des
Projektbezogenen Personenaustauschs, PPP, D1820) supported by the Deutscher Akademischer Austausch Dienst
(DAAD) in Germany. The author also gratefully acknowledges support through the scholarship program from the
China Scholarship Council (CSC).





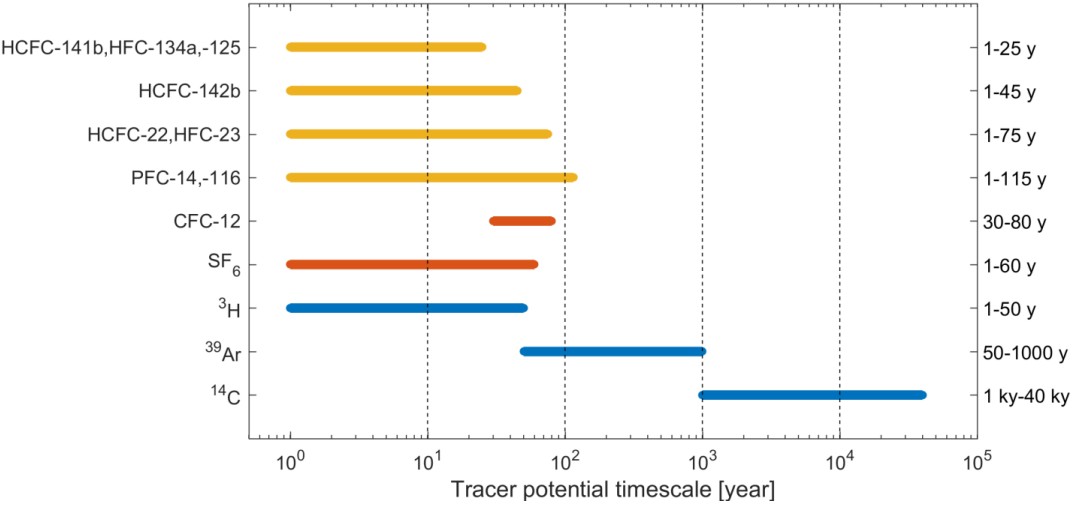

**Figure 1**. Tracer age ranges ("seawater timescales") of halogenated compounds dating using potential chronological transient tracers (selected HCFCs, HFCs and PFCs, orange) as well as traditional chronological transient tracers (CFC-12 and SF₆, red) combined with radioisotope dating using radioactive transient tracers ($^3$H, $^{39}$Ar and $^{14}$C, blue). Tracer age ranges of chronological transient tracers are estimated from Fig. 3 (see below), while tracer age ranges of radioactive transient tracers are from Aggarwal (2013).

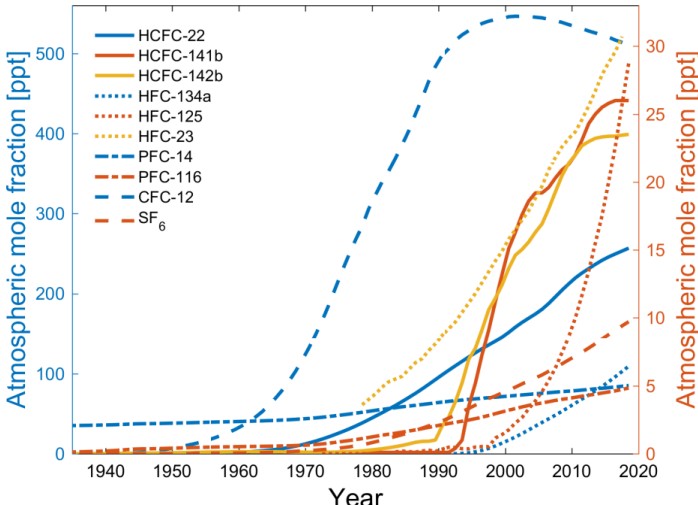

**Figure 2.** Atmospheric histories of HCFC-22, HCFC-141b, HCFC-142b, HFC-134a, HFC-125, HFC-23, PFC-14, PFC-116, CFC-12 and SF₆ in the Northern Hemisphere. HCFC-22, HFC-134a, PFC-14 and CFC-12 share the left y-axis scale; other compounds share the right y-axis scale.

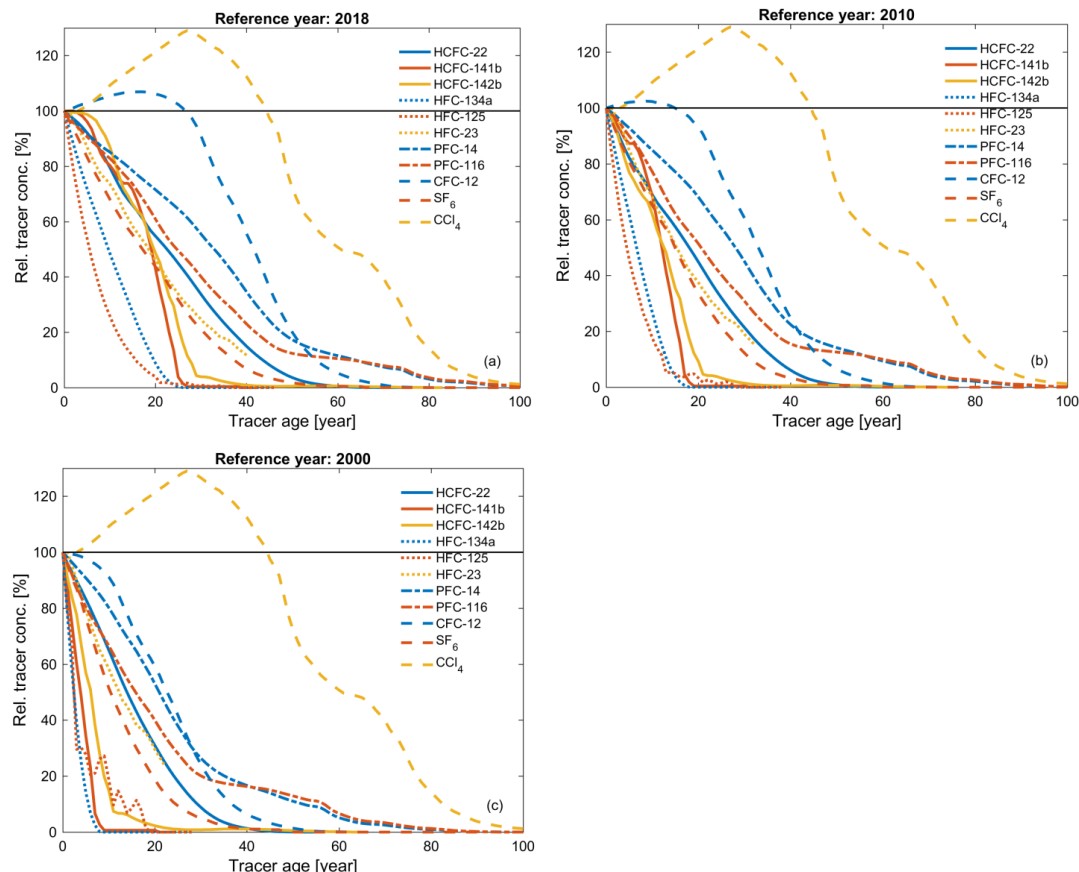

**Figure 3.** Relative tracer concentrations in percent and corresponding tracer age for HCFC-22, HCFC-141b, HCFC-142b, HFC-134a, HFC-125, HFC-23, PFC-14, PFC-116, CFC-12, SF$_6$ and CCl$_4$ in the Northern Hemisphere. Reference year: (a) 2018, (b) 2010, (c) 2000.



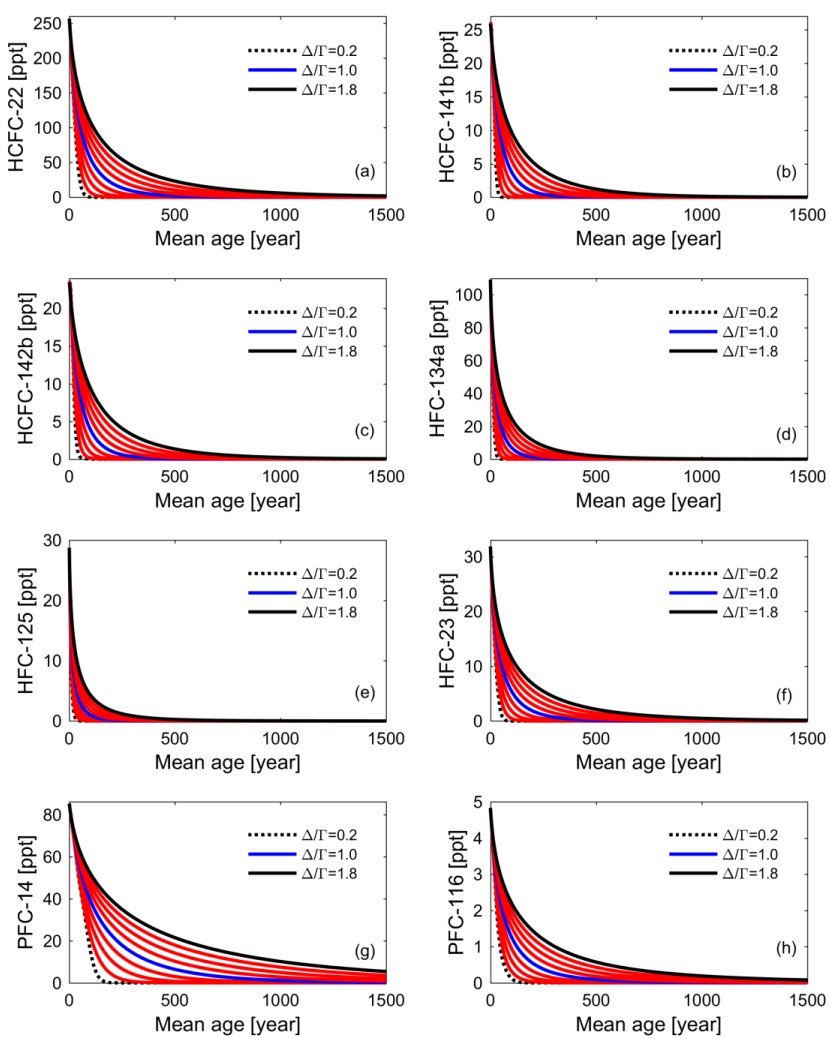

**Figure 4.** Transient tracer concentrations (ppt, parts per trillion) of HCFC-22, HCFC-141b, HCFC-142b, HFC-134a, HFC-125, HFC-23, PFC-14 and PFC-116 vs. mean age for different $\Delta/\Gamma$ ratios (a range of 0.2-1.8) in the Northern Hemisphere. The unity ratio of 1.0 is shown as a blue line.






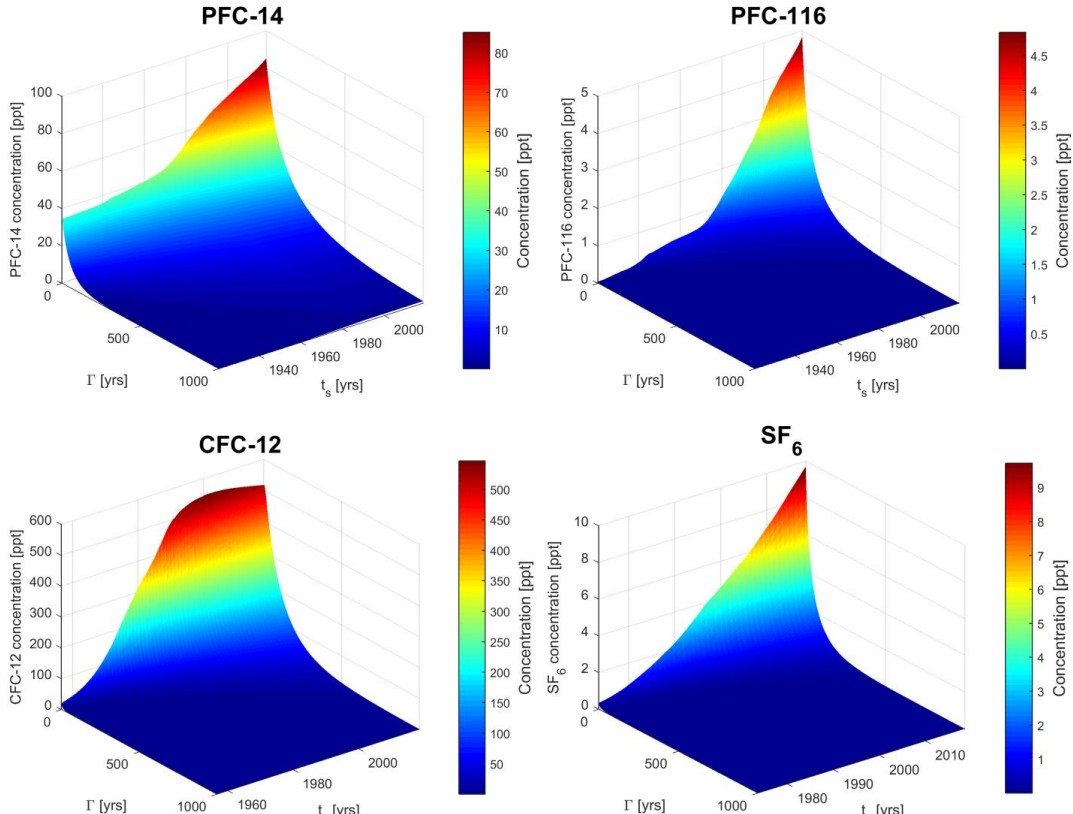

**Figure 5.** HCFC-22, HCFC-141b, HCFC-142b, HFC-134a, HFC-125, HFC-23, PFC-14, PFC-116 CFC-12 and SF$_6$: concentrations (ppt) in different sampling year (t$_s$) and mean age ($\Gamma$) in the Northern Hemisphere with $\Delta/\Gamma = 1.0$ based on the IG-TTD with 100 % saturation.



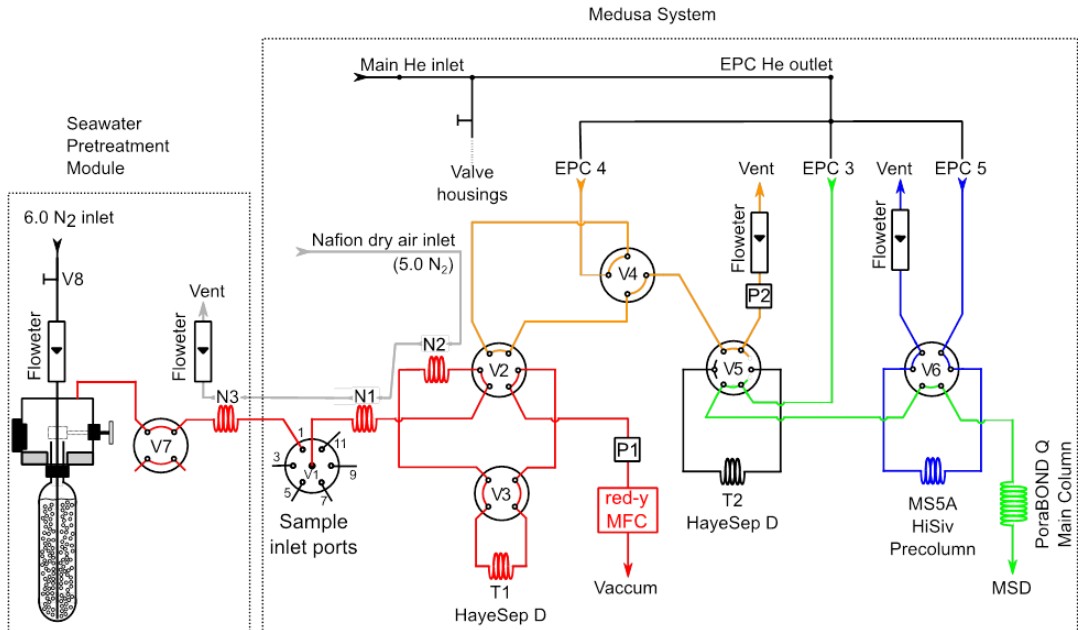

**Figure 6**. Medusa-Aqua system flow scheme. The Medusa system remains identical to that given by Miller et al. (2008). The seawater pretreatment module is added to degas the samples from gaseous tracers before injecting into Medusa. Electronic Pressure Controllers (EPC3, EPC4, and EPC5) supply helium throughout the system. The Mass Flow Controller (MFC) is used to measure the sample volume downstream of trap 1 (T1) but not used in this study. The cryogenic packing materials are 200 mg of 100/120 mesh HayeSep D (HSD) for Trap 1 (T1) and 5.5 mg of HSD adsorbent for Trap 2 (T2).

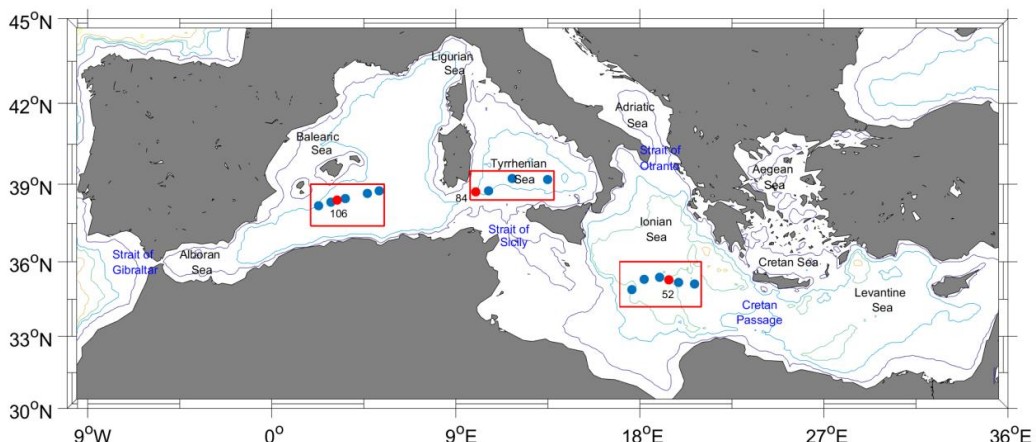

**Figure 7**. Sampling sites distributed in the Mediterranean Sea from the cruise MSM72 in three areas: Southern Ionian Sea (SIS), Tyrrhenian Sea (TS) and Western Mediterranean Sea (WMS). Sampling sites in red solid circles indicate samples measured by the Medusa-Aqua system for HCFCs, HFCs, PFCs and CFC-12, and the ones in blue solid circles stations were for CFC12 and $SF_6$ measured by the PT-GC-ECD. The depth contours are 500 m, 2000 m, 3000 m, 4000 m, 5000 m and 6000 m.



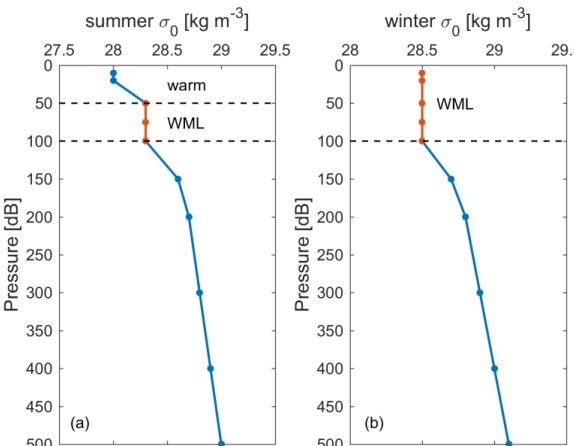

**Figure 8**. Example of the winter mixed layer (WML) depth (marked as red) determined in summer and winter in potential density ($\sigma_0$) profiles especially for historical seawater saturation calculation.

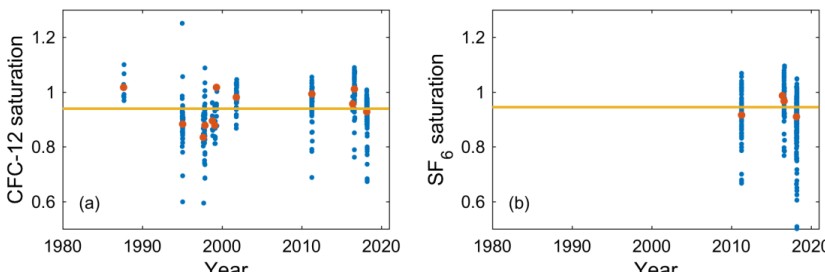

**Figure 9.** Historical seawater saturations in winter mixed layers (blue solid circles) for (a) CFC-12 from 12 cruises from 1987 to 2018 and (b) SF$_6$ from 4 cruises in the Mediterranean Sea. In addition to the data from Schneider et al. (2014), data from the cruises CRELEV2016 and TALPro2016 in 2016 and MSM72 in 2018 (Li and Tanhua, in preparation) were added. Red solid circles denote the means of seawater saturation for each cruise.





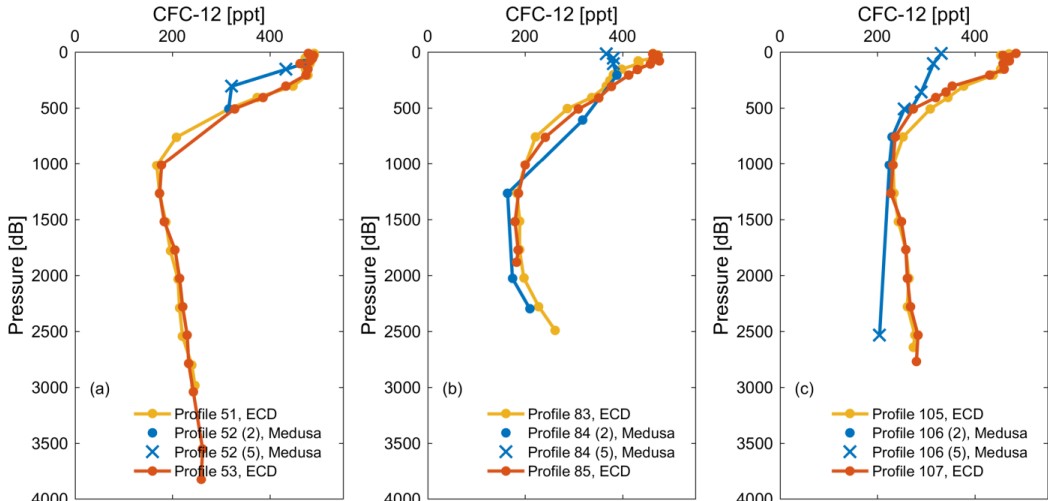

**Figure 10.** Comparison of CFC-12 observations measured by PT-GC-ECD and Medusa-Aqua system in three areas: (a) Southern Ionian Sea, (b) Tyrrhenian Sea and (c) Western Mediterranean Sea from cruise MSM72. For profiles 52, 84 and 106, (2) and (5) refer to the quality flags of the data (Table S1). We used normal quality control routines and looked for outliers vs. pressure, such outliers got a flag "3" (probably bad) and are not further considered. Flag "2" represent data "good" when compared over pressure. We did one more step and compared the Medusa-Aqua system with the PT-GC-ECD observations; samples that received a flag "2" in the first step, but where the CFC-12 values are inconsistent with the CFC12 values from PT-GC-ECD measurements, was flagged "5" in Table S1, indicating a possible issue during the measurement process. In the following plots we show all data with a quality flag of "2" or "5", values flagged "2" are marked as dots, whereas values flagged "5" are marked as crosses in the following.



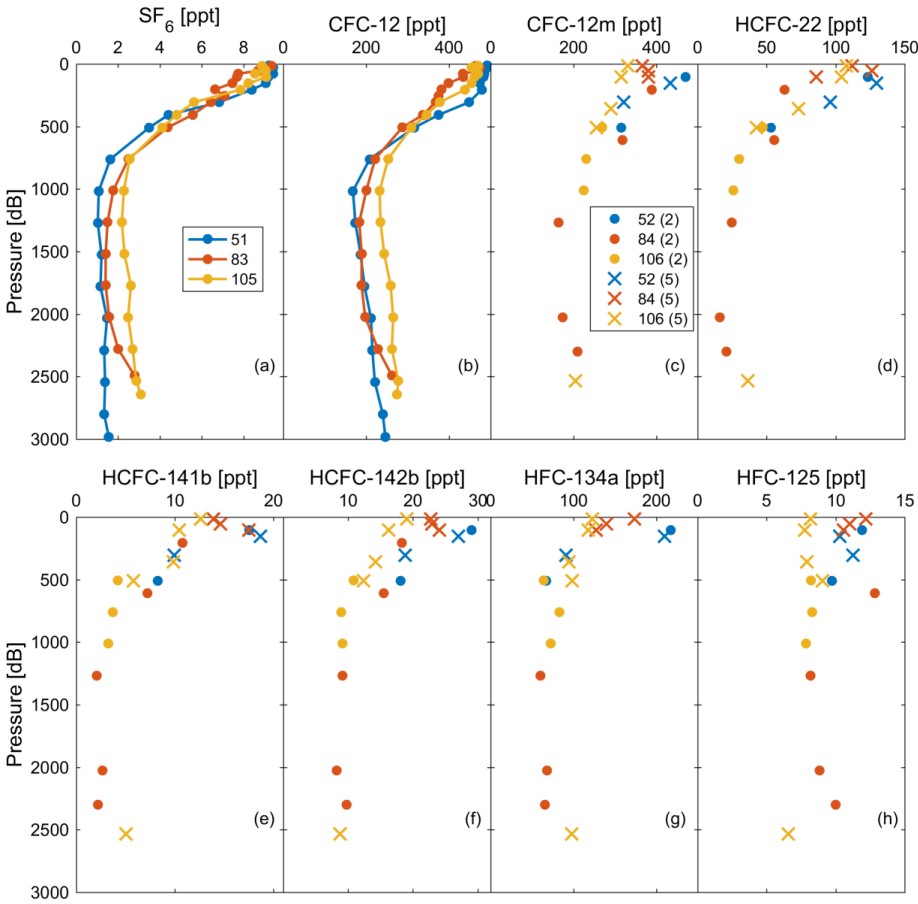

**Figure 11**. Observations of (a) SF$_6$ and (b) CFC-12 in profiles 51, 83 and 105 measured by PT-GC-ECD and (c) CFC-12 (marked as CFC-12m), (d) HCFC-22, (e) HCFC-141b, (f) HCFC-142b, (g) HFC-134a and (h) HFC-125 in profiles 52, 84 and 106 measured by Medusa-Aqua system. For the explanation of (2), (5), dots and crosses, refer to Fig. 10.

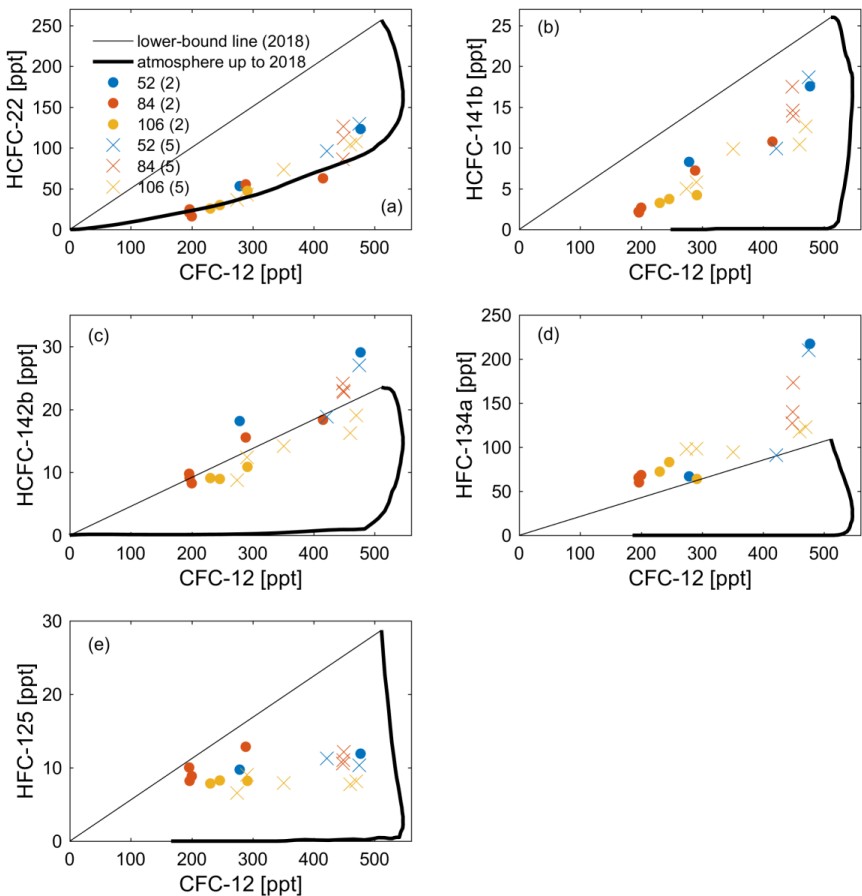

**Figure 12**. Observations of (a) HCFC-22, (b) HCFC-141b, (c) HCFC-142b, (d) HFC-134a and (e) HFC-125 measured by Medusa-Aqua system plotted against the interpolated CFC-12 based on measurements by PT-GC-ECD. The thick black line is the atmospheric history of the tracer pair and the thin black line is the theoretical mixing line between contemporary concentrations and pre-industrial. For the explanation of (2), (5), dots and crosses, refer to Fig. 10.



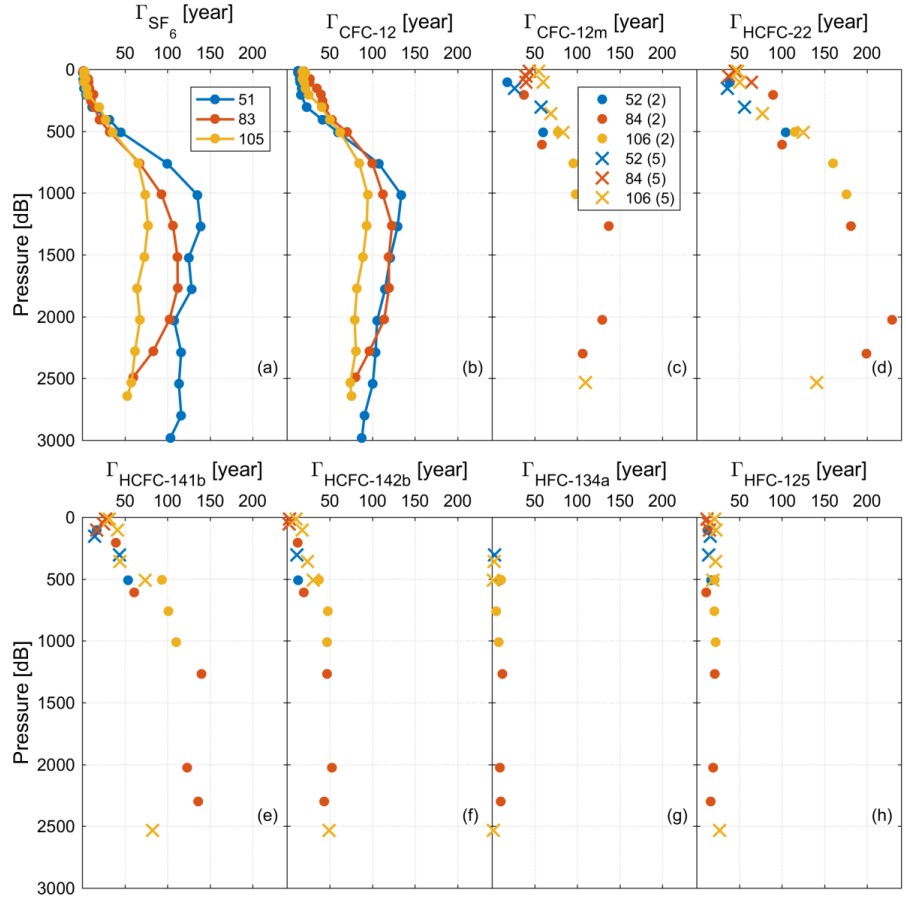

**Figure 13.** Mean age of (a) SF$_6$ and (b) CFC-12 in profiles 51, 83 and 105 and (c) CFC-12, (d) HCFC-22, (e) HCFC-141b, (f) HCFC-142b, (g) HFC-134a and (h) HFC-125 in profiles 52, 84 and 106 based on $\Delta/\Gamma = 1.0$ of IG-TTD. For the explanation of (2), (5), dots and crosses, refer to Fig. 10.





**Table 1.** Review on biodegradation of HCFCs and HFCs in freshwater or soil

| Microorganisms or culture | HCFC-22 | HCFC-141b | HCFC-142b | HFC-134a | HFC-125 | HFC-23 | References |
|---|---|---|---|---|---|---|---|
| Methanotrophic bacterium *Methylosinus trichosporium OB3b*(pure culture) | | √ [a] | x [b] | x | | | (DeFlaun et al., 1992) (Streger et al., 1999) |
| Mixed methanotrophic culture (MM1) with many heterotrophs | √ | | √ | √ | | | (Chang and Criddle, 1995) |
| Cell suspensions of M. capsulatus, methanotrophs in natural assemblages | √ | | | | | | (Oremland, 1996) |
| Methanotrophic mixed culture ENV2040 | | x | x | | | | (Streger et al., 1999) |
| Unidentified methanotroph ENV2041 | | x | x | x | | | (Streger et al., 1999) |
| Propane-oxidizing bacteria, *M. vaccae JOB5* | x | √ (0.1 μmol h⁻¹) | √ | x | x | x | (Streger et al., 1999) |
| Methylococcus capsulatus (Bath) | √ | | | | | | (Matheson et al., 1997) |
| Aerobic condition closed bottle tests | | x | | x | x | | (Berends et al., 1999) |
| Anoxic sediments | | √ | | | | | (Oremland, 1996) |
| Landfill soil | √ [c] | x | | x | | | (Scheutz et al., 2004) |
| Anaerobic conditions in sewage sludge and aquifer sediment slurries | x | x | x | x | | | (Balsiger et al., 2005) |

[a] √: Biodegradation in freshwater/soil; [b] x: No biodegradation in freshwater/soil; [c] in the oxidative zone

**Table 2.** Selected Medusa-Aqua analytes, calibration scales and errors in standard gas, detection limits and precision of seawater measurements

| Industrial name (abbreviation) | Chemical formula | Full name | Standard scale | Calibrated errors of the standard gas (%) | Detection limit (fmol kg⁻¹) | Precision [a] (%) |
|---|---|---|---|---|---|---|
| SF₆ | $SF_6$ | sulfur hexafluoride | SIO-05 | 0.37 | 0.48 | -- |
| CFC-12 | $CCl_2F_2$ | dichlorodifluoromethane | SIO-05 | 0.08 | 29.83 | 0.4 |
| HCFC-22 | $CHClF_2$ | chlorodifluoromethane | SIO-05 | 0.22 | 13.75 | 3.1 |
| HCFC-141b | $CH_3CCl_2F$ | 1,1-dichloro-1-fluoroethane | SIO-05 | 0.20 | 4.01 | 6.1 |
| HCFC-142b | $CH_3CClF_2$ | 1-chloro-1,1-difluoroethane | SIO-05 | 0.21 | 3.24 | 1.8 |
| HFC-134a | $CH_2FCF_3$ | 1,1,1,2-tetrafluoroethane | SIO-05 | 0.21 | 7.31 | 9.7 |
| HFC-125 | $CHF_2CF_3$ | pentafluoroethane | SIO-14 | 0.23 | 1.19 | 2.0 |
| HFC-23 | $CHF_3$ | fluoroform | SIO-07 | 0.49 | 6.71 | -- |
| PFC-14 | $CF_4$ | carbon tetrafluoride | SIO-05 | 0.30 | 0.44 | -- |
| PFC-116 | $CF_3CF_3$ | hexafluoroethane | SIO-07 | 0.32 | 1.41 | -- |

[a] Precision (reproducibility) of tracers in seawater was determined from samples (~1.3 L) at 23.5 dbar from cruise Al516 in the Baltic Sea in September 2018.





**Table 3.** Comparison of instrument performance measuring CFC-12

| System | Medusa-Aqua | PT-GC-ECD | PT-GC-ECD |
|---|---|---|---|
| Instrument | Cracker-Medusa-GC-MS | Syringe-PT-GC-ECD | Cracker-PT-GC-ECD |
| Workplace | Laboratory in Kiel | Onboard | Laboratory in Kiel |
| Purge efficiency (%) | $99.5 \pm 0.5$ [a] | 99.2 - 3.6 [b] | $99.6 \pm 0.08$ [c] |
| Precision (%) | 2.0 | 0.34 | 1.4 |
| Sampling volume (L) | ~1.3 | ~0.25 | ~0.3 |
| Sampling sites | Red dots in Fig. 7 | Blue dots in Fig. 7 | - |
| Measured compound | CFCs, HCFCs, HFCs, PFCs, etc. | CFC-12, $SF_6$, $SF_5CF_3$ | CFC-12, $SF_6$, $SF_5CF_3$ |

[a] after purge third times; [b] after purge once; [c] after purge twice

**Table 4.** Seawater surface saturations (%) of $SF_6$ and CFC-12 in profiles 83 and 105 (measured on-board with the PT-GC-ECD system) and CFC-12, HCFC-22, HCFC-141b, HCFC-142b, HFC-134a and HFC-125 in profiles 84 and 106 (measured in Kiel with the Medusa-Aqua system).

| Profile | Pressure (dbar) | $SF_6$ | CFC-12 | Profile | Pressure (dbar) | CFC-12 | HCFC-22 | HCFC-141b | HCFC-142b | HFC-134a | HFC-125 |
|---|---|---|---|---|---|---|---|---|---|---|---|
| 83 | 13.4 | 97 | 91 | 84 | 14.7 | 72 | 44 | 54 | 97 | 163 | 44 |
| 105 | 14.3 | 92 | 92 | 106 | 14.0 | 65 | 42 | 49 | 82 | 115 | 30 |

**Table 5.** Evaluating the possibilities of selected HCFCs, HFCs and PFCs to be transient tracers in the ocean

| Compound | Atmospheric history | Solubility in seawater | Stable in seawater | Can be measured in seawater | Possibility to be transient tracer |
|---|---|---|---|---|---|
| CFC-12 | *** | *** | *** | *** | *** |
| HCFC-22 | *** [a] | ** | * | *** | * |
| HCFC-141b | *** | * | ** | *** | ** |
| HCFC-142b | *** | * | * | *** | * |
| HFC-134a | *** | ** | * | *** | * |
| HFC-125 | *** | * | * | ** | * |
| HFC-23 | ** | ** | * | * | * |
| PFC-14 | *** | ** | *** | * | ** |
| PFC-116 | *** | * | *** | * | ** |

[a] the number of stars represents the low, medium and high confidence or feasibility of the results: one star (*) means "low confidence or feasibility or largely unknown", two stars "reasonable well resolved" and three stars "well documented or resolved".

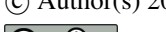



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
