# Peer review of "Medusa-Aqua system: simultaneous measurement and evaluation of novel potential halogenated transient tracers HCFCs, HFCs and PFCs in the ocean"

_Ocean Science, 2019_

## Referee Comment (RC2)

I have reviewed the manuscript "Medusa-Aqua system . . . ", by Pingyang Li and Tosta Tanhua and, while I find it appropriate for publication, it needs considerable modification before that. The authors describe a promising approach for continuing our ability to date ocean water masses with transient tracers as their emission histories evolve and new methods for their analysis emerge. The community faces a significant challenge in monitoring these gases in the ocean and interpreting results, as they are sequentially phased out and their atmospheric histories are no longer monotonic. Tracers useful today may not be useful tomorrow, so it is important that analytical and interpretive skills improve over time as well. This paper describes such an improvement and demonstrates its capabilities with verification in the field.

However, the manuscript is loosely written in some parts and more wordy than it needs to be. The authors need to seek out and eliminate redundant statements and tighten up the language where they can. I also worry about the organization and think it would be best to start with the instrument description, show the data, then interpret and explain it. Because of its organization, the manuscript tends to bounce the reader around rather than build a case from observations. Part of me wants to suggest that the authors write two papers – one about the method and results and one about the value of the tracers. But I think they can get this into one if it is better organized and tightened up.

Some of the figures are excellent and appropriate, while others have too much information and are not useful. The overall important message is that applying this new technology allows for the measurement of new transient tracers that will likely be useful over the next decade in understanding ocean transport. Going too far beyond that is superfluous and should be avoided.

SOME SPECIFIC SUGGESTIONS

PAGE 1

The abstract overall could be shortened. Summary statements are better in the abstract than explanations, which are in the text.

Line 13 – replace "be the most possible" with "have the greatest potential as"

Line 14 – Can these two compounds be put into the sentence above? The way it's worded, it's not clear what the difference between "possible" and "potential" is.

Line 15 – delete "the" after "as"

Line 32 – "most potential" is awkward. Try "the compounds that have the greatest potential as tracers in the future are  . . ."

**PAGE 2**

Line 8 – Place "and" before SF6 and add "has been used" after SF6; replace they with both

Line 9 – replace "as are" with "and"

Line 10 – Add "are known" after "time"

Lines 13,14 – Replace "to be" with "as"

Line 27 – "solubility" is not a part of the "input history"

**PAGE 3**

Line 7 – Hydrolysis is not a significant mechanism for destroying CCl4 in seawater, but in some locations microbial degradation in low oxygen waters is.

Lines 9-11 – Awkward sentence; please revise.

Line 22 – Add "ing" to "discuss"

Lines 23-24 – Awkward sentence

Lines 24-25 – Incomplete sentence

Line 32 – replace "as" with "and"

Lines 33-37 – Consider tabulating these numbers instead of putting them into text.

**PAGE 4**

Line 2 – Add "s" to "enter"

Lines 4, 5 – Hydrolysis is only one mechanism; low hydrolysis does not rule out other possible chemical or biological removal mechanisms. Revise or delete the sentence.

Line 11 – delete "a"; add "s" to "tracer"

Line 22 – delete last sentence

Line 24 – change "target" to "targeted" (?)  First sentence here is a little awkward; perhaps it can be improved.

Line 27 – delete "would"

Lines 30-36 – This structure for the paper feels awkward. Would it be better to describe the instrument first, show some results from it, and then go into the modeling and descriptions of how the data can be used? I think the paper would be easier on the reader if that approach were taken.

**PAGES 5-7**

Section 2.2 is good background but seems to meander. Can it be condensed and still deliver what is needed for the reader to interpret the data that are later presented?

**PAGE 5**

Line 22 – Add "ally" to "monotonic"

**PAGE 10**

Line 10 – add "s" to "standard", delete "measurements", replace "done" with "measured" or "analyzed"

**PAGE 11**

Line 1 – delete "effective"

Lines 5-9 – This could probably be deleted with no impact on the paper.

Line 9 – change "was" to "were"

Line 10 – add "s" to "depth"

Line 20 – Sentence is awkward.

Line 21 – replace "to" with "than"

**PAGES 12, 13**

Section 5.3 – These low values for surface saturations seem like a serious concern for the capability of the Medusa-Aqua system. Although loss during transport of samples is possible with some compounds, it is not likely for CFC-12. I hope this is not a "show-stopper", meaning that the Medusa-Aqua system is not capable of quantitatively analyzing these compounds from seawater. The authors need to do a better job of explaining this so as not to mislead the reader about the capabilities of this system. If it has issues, what do they suggest for improving it?

Section 5.4 – It's always difficult to follow description of a figure while reading the text and this section is no exception. I would prefer such a description to appear in the figure caption and then for the authors to use the text simply to state what the figure means. That lets the reader get on with the story and not get bogged down looking back and forth.

Section 5.5 – This discussion seems to meander about. It is not clear and it's hard to follow. It takes several readings to get to what the authors are trying to say. What exactly are the points the authors are trying to make? I recommend they state those points and use the figure references (parenthetically) to support them. Such an approach would be good for all three of these sections.

**PAGE 14**

Section 6 – This section, too, meanders. Here and elsewhere (e.g., previous sections), the authors get into trouble when they start describing the content of figures (or in some cases tables) in the text. The authors would do well to consider referring to figures and tables only parenthetically, then using the text to make their important points, i.e., treat the figures much as they do references to other papers. Figure descriptions and explanations should go into the figure captions, thus freeing the authors (and readers) to engage solely in scientific meaning in the text. The idea is that text and figures should be able to stand alone. When this is done, it makes for a much easier read and quicker understanding.

Line 12 and elsewhere – The reference to number of stars is mentioned in the text without reference to their meaning. This is another example of not keeping the text and figures independently coherent. The reader should not have to refer to a figure to understand what is being said in the text, and *vice versa*.

Line 16 – Should HCFC-12 be HCFC-22?

Lines 24-26 – This reminds me that here and in the "highlights" section at the beginning of the paper, the authors refer to "potential" and "possible" as different things, but do not state how they define that difference. They may want to drop this distinction or else provide definitions for the terms.

**PAGE 15**

Line 15 – delete "and concentration". Also, this sentence is a bit awkward

Line 29 – "estimating" should be "estimate"

**PAGE 16**

Line 3 – "estimating" should be "estimate"

Section 7 – This section needs to be more succinct. It is very important that the reader can understand immediately what the paper found. Statements about how things were done are not necessary for the conclusion, only the findings.

**FIGURES AND TABLES**

Figure 3 – For the purposes of this paper, I don't believe it is necessary to show the same plot three times with different reference years. I recommend choosing one as an example to make the point that each gas has a different history that is recorded differently in the ocean.

Figure 4 – As it stands, this figure is useless. There is little to be obtained from this plethora of miniscule plots. I think the authors are trying to make the point that for any one of these gases there is some uncertainty as to their age distributions in the ocean. They can do that with one plot and a nice caption supporting it.

Figure 5 – I'm not sure this figure adds anything. If there is a point the authors are trying to make with this figure, they should do so clearly and, as for Figure 4 use one plot (maybe two, depending upon the point they are trying to make) and a clear description in the caption.

All other figures are valuable and I believe necessary to the paper.

The tables are all useful in the manuscript. In Table 2, it would be good to note whether the reproducibility is expressed as one or two sigma or if another approach is used. Tables 4 and 5 support the text, but, as noted above, the text in the manuscript needs to draw out the meaning of what these tables contain in a way that is readily understood by the reader.

FINAL NOTE: Finally, I would appreciate it if the authors would have someone else in their institute go through the final version of the next draft and catch typos, grammatical errors, etc. Independent eyes are always useful for finding things that authors miss simply because they've worked the text so many times.

---

## Referee Comment (RC1) · Anonymous Referee #1 · 5 Nov 2019

This manuscript examines the usefulness of a suite of halogenated compounds as ocean transient tracers to better constrain ocean ventilation. I think this is an important topic, and can see many potential benefits of having measurements of the suite of tracer with different atmospheric histories. However, I think the manuscript needs some revisions before it is suitable for publication.

MAJOR COMMENTS

1. I think the overall conclusions of the study need to be more clearly presented. I

like Table 5, and accompanying discussion in Section 6, as a summary of the analysis/paper, but the Introduction, and Conclusions are less clear. I think some one just reading these (as a lot of people will do before reading the whole paper) will come away a bit confused. I don't have any specific suggestions, but I think some rewriting is required so it is clearer which tracers are definitely not suitable, and which are (and may be a third category of tracers where not sure and more measurements / tests needed).

As part of this I think you need to clarify what is meant (or remove usage) in the highlights by "most possible" and "most potential" tracers. I think I understand what is meant but only after reading the paper in detail.

2. As written the manuscript seems to suggest that a conserved tracer is only useful as a transient tracer if it is linearly increasing, and can then be used to estimate the mean age. However, a tracer with nonlinear history can still be used as a transient tracer, and be used to help constrain the TTD (especially if multiple tracers are available). For example, a linear and quadratically increase can be used in combination to estimate both the mean and the width. Also, the fact that the atmospheric concentration of the tracers is essentially zero at differing times in the past (e.g. HFC-125 in late 1990s, HCFC-142b in late 1980s, and HCFC-22 in late 1960s) might be very useful for containing aspects of the TTD, even if can't estimate the mean age from each tracer. The authors make a few references to determining more complex TTDs as in Stoven and Tanhua, but even if you still assume IG TTD multiple tracers could be used to estimate both parameters of the IG.

It might be asking too much for a detailed multi-tracer analysis in this paper, but even if this is the case I think the possibility a multi-tracer analysis that does not aim solely at estimating the mean age needs to be discussed. I would encourage the authors to think if there are more appropriate approaches than just estimate the mean age assuming D/G=1, and whether this may change conclusions on whether a tracer is useful.

3. The description of how the mean age is estimated from each tracer needs to be

clearer. I don't find the description on top of page 7 and figs 4 and 5 very helpful, and if I wasn't already familiar with the approach would be confused.

Related to this the text "mean age of HCFC-141b" (and other such usage throughout the manuscript) needs to be replaced with "mean age estimated from HCFC=141b". This may seem like petty semantics, but it is important to be clear that you are trying to estimate the same mean age from different tracers and not a tracer dependent quantity (as written it implies the latter). I think a paragraph where you clearly describe how you estimate the mean age from measurements of a given tracer, and then define your terminology, will help.

Actually, what are the take home points from figures 4 and 5? I understand what is shown but I don't know what I am meant to take home from them (and if purpose is to describe the calculation of the mean age, I don't think it works). Note, I find figures like fig 5 nice to look at but difficult to interpret / read off values.

4. The shorter atmospheric history is given as a possible cause of the young estimates of mean age from HCFC-142b, HFC-134a, and HFC-25 (pg 13, line 21), but what does this not cause a similar bias for HCFC-141b? Can the differences in mean age estimates be linked to the location of points on figure 12? I think fig 12 and 13 need to be linked together better than they currently are.

Would it help to include curves for different Delta/Gamma on figure 12 (thick curve is Delta=0, and could add curves for a couple of values D/G)? This might show whether data consistent with different values of D/G.

MINOR COMMENTS:

Pg 6, line 5: This description of fig 3 needs to be earlier, when fig 3 is first mentioned. Also, I think what is meant by relative tracer concentration should be in the caption.

Pg 6, line 23: The sentence "... the calculated age is mean age considered both advection ..." does not make sense.

Pg 12, lines 27-34 are repetitive.

Pg 13, line 15. Remove "1" from "1IG".

Pg 15, line 1: "one star as no as some higher ..." does not make sense.

---

## Author Comment (AC1) · 6 Feb 2020

pli@geomar.de, ttanhua@geomar.de

6 February 2020

We thank two anonymous referees for your constructive suggestions and comments. Below, we address all the comments and describe our responses to them where we refer to the revised manuscript (Italic) by listing the pages or sections of changes. In the following Revised Manuscript and Supplement, all changes from the original text are marked.

**Anonymous Referee #1**

MAJOR COMMENTS

Comment: 1. I think the overall conclusions of the study need to be more clearly presented. I like Table 5, and accompanying discussion in Section 6, as a summary of the analysis/paper, but the Introduction, and Conclusions are less clear. I think someone just reading these (as a lot of people will do before reading the whole paper) will come away a bit confused. I don't have any specific suggestions, but I think some rewriting is required so it is clearer which tracers are definitely not suitable, and which are (and maybe a third category of tracers where not sure and more measurements/tests needed).

As part of this I think you need to clarify what is meant (or remove usage) in the highlights by "most possible" and "most potential" tracers. I think I understand what is meant but only after reading the paper in detail.

Response: Thank you very much for your comment. All the "most possible" and "most potential" have been removed. We have rewritten the Introduction and Conclusion, see the Revised Manuscript. Of the investigated halogenated compounds, HCFC-142b and HCFC-141b are found to be the most promising transient tracers currently and PFC-14 and PFC-116 for the future (medium confidence). HCFC-22 is not suitable as a transient tracer in warm waters but possible for cold waters. We are not able to evaluate the potential of HFC-134, HFC-125, and HFC-23 as tracers fully due to lacking information on their solubility and stability in seawater and with potential analytical challenges. But we could no longer consider HCFC-22, HFC-125 and HFC-23 as tracers since they could be replaced by mature or better transient tracers based on the similar atmospheric history.

Comment: 2. As written the manuscript seems to suggest that a conserved tracer is only useful as a transient tracer if it is linearly increasing, and can then be used to estimate the mean age. However, a tracer with nonlinear history can still be used as a transient tracer, and be used to help constrain the TTD (especially if multiple tracers are available). For example, a linear and quadratically increase can be used in combination to estimate both the mean and the width. Also, the fact that the atmospheric concentration of the tracers is essentially zero at differing times in the past (e.g.

HFC-125 in late 1990s, HCFC-142b in late 1980s, and HCFC-22 in late 1960s) might be very useful for containing aspects of the TTD, even if can't estimate the mean age from each tracer. The authors make a few references to determining more complex TTDs as in Stoven and Tanhua, but even if you still assume IG TTD multiple tracers could be used to estimate both parameters of the IG.

It might be asking too much for detailed multi-tracer analysis in this paper, but even if this is the case I think the possibility of a multi-tracer analysis that does not aim solely at estimating the mean age needs to be discussed. I would encourage the authors to think if there are more appropriate approaches than just estimate the mean age assuming D/G=1, and whether this may change conclusions on whether a tracer is useful.

Response: Many thanks for your comment. Of course, we could try more methods. However, there are still too many unknowns in the calculation involving new tracers with uncertain stability, and sometimes atmospheric history will make it impossible to be applied by some methods. Besides, these methods may lead to a bigger bias based on limited measurements. Actually, we did try some based on current observations, but the time-variant ventilation in the Mediterranean Sea made our attempts fail. Therefore, we would like to try these methods after we obtain more observations. Since our main purpose in this study is to solve our primary task – evaluating the usefulness of the new tracers, it will be future work to explore more appropriate approaches based on multi-tracers rather than only TTDs.

Comment: 3. The description of how the mean age is estimated from each tracer needs to be clearer. I don't find the description on top of page 7 and figs 4 and 5 very helpful, and if I wasn't already familiar with the approach would be confused. Related to this the text "mean age of HCFC-141b" (and other such usages throughout the manuscript) needs to be replaced with "mean age estimated from HCFC-141b". This may seem like petty semantics, but it is important to be clear that you are trying to estimate the same mean age from different tracers and not a tracer dependent quantity (as written it implies the latter). I think a paragraph where you clearly describe how you estimate the mean age from measurements of a given tracer, and then define your terminology, will help.

Actually, what are the take-home points from figures 4 and 5? I understand what is shown but I don't know what I am meant to take home from them (and if the purpose is to describe the calculation of the mean age, I don't think it works). Note, I find figures like fig 5 nice to look at but difficult to interpret/read off values.

Response: Thanks for your comment. Considering two referees commented on the original Figures 4 and 5, the two figures have been moved to Supplement as Figures S1 and S2. All "*mean age of*" in the manuscript has been changed to "*mean age (estimated) from*". Actually, the concept of TTD and Equations (1) and (2) are used to calculate the mean age. Through the TTD, we obtained the mean age look-up figures (Fig. S1 and S2) based on the input functions of tracers, sampling year, theoretical tracer concentrations in seawater, and assumed or determined $\Delta/\Gamma$ ratio. When the observed tracer concentrations match the theoretical ones, the mean age can be determined by looking up in the mean age look-up figures. The method to estimate the mean age from a given tracer based on observations can be found in the subsection (named "Mean age and Transient Time Distribution") of Sect. 4.2 in the Revised Manuscript.

Comment: 4. The shorter atmospheric history is given as a possible cause of the young estimates of mean age from HCFC-142b, HFC-134a, and HFC-25 (pg 13, line 21), but what does this not cause a similar bias for HCFC-141b?

Can the differences in mean age estimates be linked to the location of points in figure 12? I think fig 12 and 13 need to be linked together better than they currently are.

Would it help to include curves for different Delta/Gamma in figure 12 (thick curve is Delta/Gamma=0, and could add curves for a couple of values D/G)? This might show whether data consistent with different values of D/G.

Response: Thanks for your comment. The reasons for the young estimates of mean age from some tracers have been rewritten as uncertainty in the TTD ratio that will affect tracers with different input function differently, and/or uncertainty in the solubility function or analytical error and so on. For more details, see the second paragraph of Sect. 5.5 in the Revised Manuscript.

In order to better link the differences in mean age estimates in original Fig. 13 (now Fig. 11) to the location of points in Fig. 12 (now Fig. 10), Fig. 10 and 11 have been updated with the values of the top two points of profile 52 marked with a bigger size. As seen in Fig. 10, concentrations in the shallow layers are more located in the upper right corner and vice-versa.

In Fig. 10, curves of $\Delta/\Gamma$ ratios from 0.2 to 1.8 have been added. Thus curves of $\Delta/\Gamma = 0.0-1.8$ have been plotted. Besides, $SF_6$ concentrations from profiles 51, 83 and 105 again CFC-12 concentrations have also been added for comparison (Fig. 10f). Here both $SF_6$ and CFC-12 were measured by the PT-GC-ECD. Obviously, the data is not consistent with different values of $\Delta/\Gamma$. The reason for assuming $\Delta/\Gamma=1$ see the first paragraph of Sect. 5.5 in the Revised Manuscript.

Figures 10 and 11, their captions and related contents in Sect. 5.4 and 5.5 have been revised, see the Revised Manuscript.

MINOR COMMENTS

Comment: Pg 6, line 5: This description of fig 3 needs to be earlier when fig 3 is first mentioned. Also, I think what is meant by relative tracer concentration should be in the caption.

Response: The description of the original Fig. 3 (now Fig. 5) has been moved upwards, see the first paragraph of subsection (named "Tracer age") of Sect. 4.2 in the Revised Manuscript. We also added the meaning of relative tracer concentrations in the caption of Fig. 5.

Comment: Pg 6, line 23: The sentence "… the calculated age is mean age considered both advection …" does not make sense.

Comment: Pg 12, lines 27-34 are repetitive.

Response: The related sentences have been removed.

Comment: Pg 13, line 15. Remove "1" from "1IG".

Response: "assumption of a 1IG-TTD" has been changed to "assumption of an IG-TTD".

Comment: Pg 15, line 1: "one star as no as some higher …" does not make sense.

Response: The sentence "*Our ability to estimate the stability of HCFC-142b only got one star as no as some higher than expected concentration points to another issue (see Sect. 5.4) though its seawater surface saturation was similar to the ones of $SF_6$ and CFC-12*" has been rewritten as "*We have medium confidence in our ability to estimate the stability of HCFC-142b because of slightly higher than the expected concentrations in the interior ocean (Fig. 10), lower than expected mean ages particularly in the Atlantic Water Layer (Fig. 11), and its surface saturation similar to the ones of CFC-12/$SF_6$ in seawater*".

**Anonymous Referee #2**

GENERAL COMMENTS

Comment: I have reviewed the manuscript "Medusa-Aqua system . . . ", by Pingyang Li and Toste Tanhua and, while I find it appropriate for publication, it needs considerable modification before that. The authors describe a promising approach for continuing our ability to date ocean water masses with transient tracers as their emission histories evolve and new methods for their analysis emerge. The community faces a significant challenge in monitoring these gases in the ocean and interpreting results, as they are sequentially phased out and their atmospheric histories are no longer monotonic. Tracers useful today may not be useful tomorrow, so it is important that analytical and interpretive skills improve overtime as well. This paper describes such an improvement and demonstrates its capabilities with verification in the field.

However, the manuscript is loosely written in some parts and wordier than it needs to be. The authors need to seek out and eliminate redundant statements and tighten up the language where they can. I also worry about the organization and think it would be best to start with the instrument description, show the data, then interpret and explain it. Because of its organization, the manuscript tends to bounce the reader around rather than build a case from observations. Part of me wants to suggest that the authors write two papers – one about the method and results and one about the value of the tracers. But I think they can get this into one if it is better organized and tightened up.

Some of the figures are excellent and appropriate, while others have too much information and are not useful. The overall important message is that applying this new technology allows for the measurement of new transient tracers that will likely be useful over the next decade in understanding ocean transport. Going too far beyond that is superfluous and should be avoided.

Response: Thank you very much for your comment. As suggested, the sequence of the original Sect. 3, 4 and 2 have been changed to Sect. 2, 3 and 4, that is, the interpreted method has been followed with the instrument description. The original Figures 4 and 5 have been moved to the Supplement as Figures S1 and S2. A new Table 1 (total lifetimes, ocean partial lifetimes and ocean contributions for tracers) and Table 6 (evaluating the stability of selected HCFCs and HFCs based on seawater measurements in relation to observations of CFC-12) have been added. The sequence of figures and tables has also been revised correspondingly. Besides, we have eliminated redundant statements and tighten up the language carefully.

SOME SPECIFIC SUGGESTIONS

PAGE 1

Comment: The abstract overall could be shortened. Summary statements are better in the abstract than explanations, which are in the text.

Response: Highlights (Summary statements) have been removed. The main points in the Highlights have been added in the abstract and the abstract has been shortened, see the Revised Manuscript.

Comment: Line 14 – Can these two compounds be put into the sentence above? The way it's worded, it's not clear what the difference between "possible" and "potential" is.

Response: The two sentences on "possible" and "potential" in the Highlights (Summary statements) have been removed.

Comment: Line 13 – replace "be the most possible" with "have the greatest potential as"

Response: "be the most possible" has been changed to "be the most promising"

Comment: Line 18 – delete "the" after "as"

Line 32 – "most potential" is awkward. Try "the compounds that have the greatest potential as tracers in the future are . . ."

Response: Done as suggested.

PAGE 2

Comment: Line 8 – Place "and" before $SF_6$ and add "has been used" after $SF_6$; replace they with both

Line 9 – replace "as are" with "and"

Line 10 – Add "are known" after "time"

Lines 13, 14 – Replace "to be" with "as"

Response: Done as suggested.

Comment: Line 27 – "solubility" is not a part of the "input history"

Response: "*(including known solubility in seawater)*" has been removed.

PAGE 3

Comment: Line 7 – Hydrolysis is not a significant mechanism for destroying $CCl_4$ in seawater, but in some locations microbial degradation in low oxygen waters is.

Response: "*CFC-113 was found to be lost in warm upper waters (Roether et al., 2001) and $CCl_4$ was found to hydrolysis in warm waters and low oxygen regions (Wallace and Krysell, 1989; Huhn et al., 2001)*" has been changed to "*Both CFC-113 and $CCl_4$ have been found to be degraded in warm waters (Roether et al., 2001) as well as in low oxygen waters (Wallace and Krysell, 1989; Huhn et al., 2001)*".

Comment: Lines 9-11 – Awkward sentence; please revise.

Response: "*However, with the constraints of the weak signal of $^3H$ and the decreasing atmospheric history of CFC-12, only $SF_6$ can be the relatively reliable transient tracer in the timescale range of 1-100 years despite local restrictions were in place*" has been changed to "*However, with the constraints of the weak signal of 3H and the decreasing atmospheric mole fraction of CFC-12, only SF6 is a relatively reliable transient tracer in the seawater timescale range of 1-100 years (Fig. 1)*".

Comment: Line 22 – Add "ing" to "discuss"

Line 32 – replace "as" with "and"

Response: Done as suggested.

Comment: Lines 23-24 – Awkward sentence

Response: "*Therefore their stability is presented from several other perspectives that we could find from previous studies*" has been changed to "*Therefore, their stabilities are inferred from other studies with slightly different perspectives and environmental foci*".

Comment: Lines 24-25 – Incomplete sentence

Response: "*Considering the total fraction of a compound in the ocean (as compared to the atmosphere), a small loss in the ocean is insignificant for the overall budget of the compound, but can still be important for a potential transient tracer*" has been changed to "*Considering the low fraction of these mainly non-polar compounds in the ocean, a small loss in the ocean is insignificant for the overall budget of the compound, but can still be of significance for a potential transient tracer*".

Comment: Lines 33-37 – Consider tabulating these numbers instead of putting them into text.

Response: The new Table 1 has been added on page 26 in the Revised Manuscript. The sequence of other tables has also been revised correspondingly. The related descriptions have been changed in the second paragraph of Sect. 1.3 in the Revised Manuscript.

**PAGE 4**

Comment: Line 2 – Add "s" to "enter"

Response: The related sentence has been removed.

Line 11 – delete "a"; add "s" to "tracer"

Line 22 – delete the last sentence

Line 27 – delete "would"

Response: Done as suggested.

Comment: Lines 4, 5 – Hydrolysis is only one mechanism; low hydrolysis does not rule out other possible chemical or biological removal mechanisms. Revise or delete the sentence.

Response: The sentence "*This means that oceanic chemical degradation processes alone are possibly not significant sinks for selected HCFCs and HFCs, although further research on this is needed.*" has been removed.

Comment: Line 24 – change "target" to "targeted" (?) The first sentence here is a little awkward; perhaps it can be improved.

Response: "*In order to explore if the target halogenated compounds can be used as oceanic alternative transient tracers, their atmospheric histories and seawater solubility have already been reported by Li et al. (2019)*" has been changed to "*In order to explore the possibility of the use of the targeted compounds as oceanic transient tracers, their atmospheric histories and seawater solubility have already been reported by Li et al. (2019)*".

Comment: Lines 30-36 – This structure for the paper feels awkward. Would it be better to describe the instrument first, show some results from it, and then go into the modeling and descriptions of how the data can be used? I think the paper would be easier on the reader if that approach were taken.

Response: As suggested, the sequence of original Sect. 3, 4 and 2 have been changed to Sect. 2, 3 and 4, that is, the modeling and interpreted method has been followed with the instrument description.

**PAGES 5-7**

Comment: Section 2.2 is a good background but seems to meander. Can it be condensed and still deliver what is needed for the reader to interpret the data that are later presented?

Response: Many thanks for your comments. The original Sect. 2.2 (now Sect. 4.2) is not only background information (time range and tracer/mean age definitions) but also the interpreted results (tracer similarities and the specific application range shown in Fig. 5) based on atmospheric histories and seawater solubility of HCFCs, HFCs and PFCs. The shortened Sect. 4.2 can be found in the Revised Manuscript.

**PAGE 5**

Comment: Line 22 – Add "ally" to "monotonic"

Response: The related sentence has been removed.

**PAGE 10**

Comment: Line 10 – add "s" to "standard", delete "measurements", replace "done" with "measured" or "analyzed"

Response: The related sentence has been removed.

**PAGE 11**

Comment: Line 1 – delete "effective"

Lines 5-8 – This could probably be deleted with no impact on the paper.

Line 10 – add "s" to "depth"

Line 21 – replace "to" with "than"

Response: Done as suggested.

Comment: Line 9 – change "was" to "were"

Response: "*Seawater saturations in the winter mixed layer was calculated from historical cruises data*" has been changed to "*we calculated seawater saturation in the winter mixed layer (WML) from historical cruises data*".

Comment: Line 20 – Sentence is awkward.

Response: "*Neither the historical seawater saturation of CFC-12 or SF$_6$ does show a clear trend over time*" has been changed to "*These historical seawater saturations do not show a clear trend over time*".

**PAGES 12, 13**

Comment: Section 5.3 – These low values for surface saturations seem like a serious concern for the capability of the Medusa-Aqua system. Although loss during transport of samples is possible with some compounds, it is not likely for CFC-12. I hope this is not a "show-stopper", meaning that the Medusa-Aqua system is not capable of quantitatively analyzing these compounds from seawater. The authors need to do a better job of explaining this so as not to mislead the reader about the capabilities of this system. If it has issues, what do they suggest for improving it?

Response: Section 5.3 has been rewritten by adding more observations from the Baltic Sea to describe the reasons for the low surface saturations. For details, see the Sect. 5.3 in the Revised Manuscript.

Comment: Section 5.4 – It's always difficult to follow the description of a figure while reading the text and this section is no exception. I would prefer such a description to appear in the figure caption and then for the authors to use the text simply to state what the figure means. That lets the reader get on with the story and not get bogged down looking back and forth.

Response: The caption and the related description in the text (Sect. 5.4) of the original Fig. 12 (now Fig. 10) has been changed. The captions of figures and texts related to this question have also been revised in the Revised Manuscript.

Comment: Section 5.5 – This discussion seems to meander about. It is not clear and it's hard to follow. It takes several readings to get to what the authors are trying to say. What exactly are the points the authors are trying to make? I recommend they state those points and use the figure references (parenthetically) to support them. Such an approach would be good for all three of these sections.

Response: Thanks for your comments. We have deleted some sentences and stated points by using the figure references (parenthetically) to support them, see the Sect. 5.3, 5.4 and 5.5 in the Revised Manuscript.

**PAGE 14**

Comment: Section 6 – This section, too, meanders. Here and elsewhere (e.g., previous sections), the authors get into trouble when they start describing the content of figures (or in some cases tables) in the text. The authors would do well to consider referring to figures and tables only parenthetically, then using the text to make their important points,

i.e., treat the figures much as they do references to other papers. Figure descriptions and explanations should go into the figure captions, thus freeing the authors (and readers) to engage solely in scientific meaning in the text. The idea is that text and figures should be able to stand alone. When this is done, it makes for a much easier read and quicker understanding.

Response: "in Sect. xx", "in Fig. xx" and "in Table xx" have been changed to "(see Sect. xx)", "(Fig. xx)" and "(Table xx)" in Sect. 6 and previous sections.

Comment: Line 12 and elsewhere – The reference to the number of stars is mentioned in the text without reference to their meaning. This is another example of not keeping the text and figures independently coherent. The reader should not have to refer to a figure to understand what is being said in the text, and *vice versa*.

Response: The number of stars has been replaced with the low, medium and high confidence/feasibility in Sect. 6. More footnotes have been added for the original Table 5 (now Table 7) to explain the stars inside the table.

Comment: Line 16 – Should HCFC-12 be HCFC-22?

Response: "HCFC-12" has been changed to "HCFC-22".

Comment: Lines 24-26 – This reminds me that here and in the "highlights" section at the beginning of the paper, the authors refer to "potential" and "possible" as different things, but do not state how they define that difference. They may want to drop this distinction or else provide definitions for the terms.

Response: All sentences related "most possible" and "most potential" has been removed.

**PAGE 15**

Comment: Line 15 – delete "and concentration". Also, this sentence is a bit awkward

Response: "*Our ability in estimating the stability of HFC-134a only got one star due to higher than the expected seawater surface saturation and concentration, but also not identified to be unstable, see Sect. 5.3 and 5.4*" has been changed to "*We have only poor knowledge on stability of HFC-134a since higher than expected surface saturation (Table 5) and concentration (Fig. 10), as well as lower than expected mean ages (Fig. 11) don't suggest degradation, and the compound is not identified to be unstable (see Sect. 5.4), but the stability of HFC-134a is still largely unknown considering the issues on seawater solubility function and/or measurements*".

Comment: Line 29 – "estimating" should be "estimate"

Response: All "ability to estimating" has been changed to "ability to estimate".

**PAGE 16**

Comment: Line 3 – "estimating" should be "estimate"

Response: All "ability to estimating" has been changed to "ability to estimate".

Comment: Section 7 – This section needs to be more succinct. It's very important that readers can understand immediately what the paper found. Statements about how things were done are not necessary for the conclusion, only the findings.

Response: Section 7 has been shortened, see the Revised Manuscript.

**FIGURES AND TABLES**

Comment: Figure 3 – For the purposes of this paper, I don't believe it is necessary to show the same plot three times with different reference years. I recommend choosing one as an example to make the point that each gas has a different history that is recorded differently in the ocean.

Figure 4 – As it stands, this figure is useless. There is little to be obtained from this plethora of minuscule plots. I think the authors are trying to make the point that for any one of these gases there is some uncertainty as to their age distributions in the ocean. They can do that with one plot and a nice caption supporting it.

Figure 5 – I'm not sure this figure adds anything. If there is a point the authors are trying to make with this figure, they should do so clearly and, as for Figure 4 use one plot (maybe two, depending upon the point they are trying to make) and a clear description in the caption.

All other figures are valuable and I believe necessary to the paper.

Response: From Fig. 3 (now Fig. 5), we compared the difference of relative tracer concentrations of CFC-12 in two reference years 2018 and 2000, so we suggest to keep the original Fig. 3a and Fig. 3c (now Fig. 5a and Fig. 5b).

The original Fig. 4 and 5 are combined to form the mean age look-up figure for each tracer, which has been moved to the Supplement as Fig. S1 and S2.

Comment: The tables are all useful in the manuscript. In Table 2, it would be good to note whether the reproducibility is expressed as one or two sigmas or if another approach is used. Tables 4 and 5 support the text, but, as noted above, the text in the manuscript needs to draw out the meaning of what these tables contain in a way that is readily understood by the reader.

Response: As suggested, we added a new Table 1 (ocean partial lifetimes and ocean contributions for tracers).

For the original Table 2 (now Table 3), the reproducibility is expressed as one sigma. The footnote "Precision (reproducibility)" has been changed to "Precision (reproducibility, $1\sigma$)".

The original Table 4 (now Table 5) added more surface saturations in seawater from the Baltic Sea.

A new Table 6 (evaluating the stability of tracers based on seawater measurements in relation to observations of CFC-12) has been added.

The original Table 5 (now Table 7) has been revised by adding red stars and rewriting the footnote, see the Revised Manuscript.

Comment: FINAL NOTE: Finally, I would appreciate it if the authors would have someone else in their institute go through the final version of the next draft and catch typos, grammatical errors, etc. Independent eyes are always useful for finding things that authors miss simply because they've worked the text so many times.

Response: Many thanks for your kind reminder. Some colleagues have helped us check the language carefully.

[revised manuscript text omitted]

[a] Meaning of the quality flag, this is modified from the WOCE flagging system (https://cchdo.github.io/hdo-assets/documentation/manuals/pdf/90_1/chap4.pdf, last access: 20 January 2020) only in that we added flag "5" for the purpose of this study

| Quality flag number | Meaning |
|---|---|
| 2 | Normal data; data for sampling sites that measured CFC-12 by Medusa-Aqua system matched the one by PT-GC-ECD |
| 3 | Questionable data: may not fit the profile or some other doubts |
| 4 | Problem data definitely |
| 5 | Data for sampling sites that measured CFC-12 by Medusa-Aqua system doesn't match the one by PT-GC-ECD; data quality between 2 and 3 |
| 6 | Mean of two or more measurements |
| 9 | Missing (null) data |

---

## Editor Comment (EC1) · Piers Chapman (Editor) · 25 Aug 2020

The authors have responded to the comments of the reviewers and made a large number of changes to the manuscript. As a result, the paper is somewhat shorter and easier to read than previously and more of the details have been placed in the supplementary material. I believe that the paper now needs only a few relatively minor changes to be suitable for publication, but would like to give reviewers #2 and #3 a chance to comment on the revised version.

[Figure]

Some comments I have are:

1. The abstract does not mention the samples from the Baltic; this seems odd given that these samples are used to justify some of the conclusions in the paper. 2. Nowhere in the paper do the authors state specifically that at present none of the suggested compounds can be considered to be suitable transient tracers. The last paragraph of the authors' overall response to reviewer #2 is a very clear statement of this (starting "The main result of this study..." and finishing "to guide further, more targeted experiments."). This could be added to the conclusion. 3. In section 5.2 the authors discuss variability in their CFC-12 measurements. Reviewer #2 queried this, and the authors produced Table A1 in response. While these numbers can presumably be calculated from Table S6, a bit more explanation in the text would help understanding. 4. Like reviewer #3, I feel the paper seems to suggest liabilities of the present Medusa system for seagoing use as much as the problems of using these particular tracers. The authors have made some suggestions for future improvement; these could pehaps be expanded.

---

## Referee Report (RR1)

This manuscript is a companion piece to a previous study by these two authors (along with additional co-authors) to identify potential new anthropogenic transient tracers for studies of ocean ventilation on decadal timescales. While the first manuscript (Li et al., 2019) focused on reconstructing the atmospheric histories and estimating the solubilities in water and seawater for several halogenated compounds, this manuscript is focused on the feasibility of measuring these compounds in seawater, demonstrating whether they behave conservatively, and finally rating their potential as tracers. In my view, the key to the publication of this manuscript is to have demonstrated the capability to measure the seawater concentration of these tracers. As such, the authors should have focused on the methods necessary to produce high-quality "Medusa" tracer data. In the data presented from the Mediterranean Sea, only 9 of the 21 samples are evaluated as good. Another 10 are given the flag of 5. The data from the Baltic Sea may be better, but the authors spend very little time on their interpretation. The remainder of the manuscript consists of hypothetical data interpretation (e.g. Fig. 5, Figs. S1 and S2) that is not worthy of publication on their own. I cannot recommend that this manuscript be published without major revisions that may include further method development. Developing the implementation of new tracers for studying the ocean is important, and I encourage the authors to understand the issues affecting their measurements. I would suggest some laboratory measurements using seawater samples equilibrated with atmospheric gases as a potential method for future studies.

In addition to the major scientific issue, the manuscript has many other issues with both writing and with some of the interpretation. I have listed many of them below:

p. 1, L21 – "potential be tracers"
p. 1, L31 - The source function for bomb tritium is not well constrained for much of the ocean. It was the $^3$He-$^3$H tracer pair which are used as transient tracers. On the pedantic side, $^{39}$Ar is not a "transient" tracer as it's assumed to be at steady-state.

p.2, L 3 – the industrial use of CFC-12 was phased out; assign a year to the Montreal Protocol
p.2, L9 – Although restrictions on $SF_6$ may be implemented, the concentrations in the atmosphere will continue to rise for the foreseeable future due to its long atmospheric life-time. Note that the PFCs also have high GHG potential, yet the authors conclude that they should be considered as transient tracers.
p. 2, L15 – Note that CFC-11 is a Level 1 (required) measurement for the US GO-SHIP program (https://usgoship.ucsd.edu/level-1-data/). The measurements of tracers are complicated – fortunately, the Bullister and Wisegarver (2008) paper referenced by the authors describes an analytical system capable of measuring seawater concentrations of CFC-11, CFC-12, $SF_6$, and CCl4 precisely and accurately aboard a research vessel.

p. 2, L20 – There should never be a 1-sentence paragraph

p.2, L 24 – "Well-quantified sources and sinks" instead of "known input function"

p.2, L 29 - two phrases beginning with "as" makes the sentence confusing to read

P.2, L 35 - CFC-12 is difficult to use by itself as a tracer; with some caveats, it can be used as a tracer pair with e.g. $SF_6$. Based upon the reported blank level (0.48 fmol/kg), the utility of $SF_6$ extends back about 50 y instead of 100 y.

p. 3, L8 – see https://www.ncbi.nlm.nih.gov/pubmed/12608868 for a marine natural product with a C-F bond.

p. 3, L9 – grammar: "reasoning true for PFC-116"

p.3, L 19 – replace of with to; where is "here"?

p. 3, L 24 – when I went to this ftp site, the data were available through 2004 and the only HCFC data were for HCFC-22

p. 3, L27+ - If these studies of degradation in soils have no relevance for CFC-12 in the seawater, why should they have relevance for the Medusa tracers? This contributes nothing to this manuscript.

p. 3, L. 36 – "not enough information…"

p.4, L 13 – grammar: "…rendering transient tracers to penetrate…"

p. 4, L14 – TTDs assume a time-invariant circulation. In other words, they should not be expected to produce useful information (i.e. Sec. 5.5 is just a data exercise)

p. 4, L 25 – Lobert et al., 2015 only reported HCFC-22; Beyer et al., 2014 only report CFC-1301. How are these relevant specifically to HFC-134a and HFC-125?

p.4, L 27-28 – citation for the precision of MS vs. ECD?

p. 4, L 28-29 – The last sentence in this paragraph seems to be misplaced.

p. 4, L 30 – Shortened not shorted

p. 4, L 32 – An important component of the Medusa system is the trapping system – I wouldn't say the system is "based" upon it.

p. 5, L 28 – This is the first mention of the samples from the Baltic in the manuscript. It seems like a late addition.

p. 5, L 32 – I doubt that the glass ampoule was closed with a screw.

p. 6, L 10 – Is the difference in bubble size noticeable over the depth of the ampoule? I could understand placing the purge tube near the bottom of the ampoule to increase the physical stirring of the entire water sample. In addition, the bubbles are primarily responsible for stripping the compounds out of the sample – exchange across the gas-water interface would be extremely slow.

P. 6 and Fig. 2 – The manuscript and/or figure caption need more details. For example, I assume N1, N2 and N3 are the Nafion driers mentioned in the text. If the Methods section is the major component of the paper, elaborate.

p. 6, L13 – In this method, approximately 2 L of gas is used to purge 1.3 L of sample. In comparison, the Bullister and Wisegarver CFC/SF6 system purges a 200-cc sample with approximately a liter of UHP N2. It's not surprising that it takes multiple purges.

p. 6, L 33 – What is the "stripping efficiency" for the first purge? I.e. how much of the total tracer remains in solution?

p. 7, L3 – Clarify what is meant by "noises"

p. 7, L16 – "Larger" rather than "bigger"; a single purge

p. 7 L 18 – Are the blanks additive? Since each analysis of a single purge has an error associated with it, don't those errors add up (or at least the square root of the sum of the blanks squared)?

p. 8, L 6 – monotonically

p. 8, L8 - Why does the HFC-125 source function start in the mid-1990s? In Li et al. (2019) there is a reconstructed atmospheric concentration prior to that.

p. 8, L25 – What sets the maximum useful age? Blank level? Source function?

p. 8, L31 – I'm not sure what evaluated to be transient tracers means. I would probably say demonstrated to be useful (and that includes easily measured).

p. 9 first paragraph – If both figure referred to in a paragraph are in the supplemental material, then perhaps that is where the paragraph belongs as well.

Section 5.1 – The authors stress the importance of the WML for determining the long-term average saturation, then never return to this concept. I'm not sure why this is here, since they evaluate the Medusa tracers by their surface saturation (Section 5.3)

p. 9, L 13 – Examples of density profiles – rather than exemplary

p. 9, L 23 – Takes a long time or has a long time or has been isolated from the atmosphere for a long time?

p.9, L29 – "generally comparable" is meaningless. The authors should plot CFC-12(MS) vs. CFC-12(ECD), and let the reviewer decide if they are comparable.

p. 9, L31 – Is 5.9% an absolute difference? This is much larger than the precision of the measurements. How are data flagged as "good"? Note that only 9 of 21 samples are classified as "good".

Section 5.2 – This section is entitled "Observations of Medusa Tracers in Seawater" yet the Medusa tracers are never discussed. The final sentence points to Fig. 9.

p. 10, L 5 – In Section 5.1, the authors wrote that there is too much variability in the surface saturation to be useful.

p. 10 , L7 – Do not forget bubble injection and/or vertical mixing

p. 10, L10 – If the Medusa Aqua measurements of CFC-12 are 20% too low at the surface of the Mediterranean Sea, how can the reviewer have confidence in any of the other tracer measurements? Unless the authors can provide a reasonable explanation for this discrepancy for CFC-12, the remainder of the data are suspect in my opinion. Likewise, the surface CFC-12 saturations of 122% in the Baltic Sea also seem unreasonable. Since data from only one station in the Baltic Sea is reported, I assume that the uncertainties in the surface saturation are an indication of the precision of the Medusa tracer measurements. These are also not very reassuring for the most part (except for HCFC-142b at 2%).

The manuscript would greatly benefit from measurements of the Medusa tracers in the laboratory under controlled settings. I need to be convinced that Medusa Aqua can reliably measure the tracer concentration under laboratory conditions in order to have confidence in the reported measurements from the Mediterranean and Baltic Seas.

p. 10, L 25 – grammar: easy to soluble

p. 11, L 21 – Why not take the approach of finding a IG-TTD with a delta/gamma ratio that matches CFC-12 and SF6 together (instead of assuming the ratio is 1), and then applying this IG-TTD to the Medusa tracers?

p. 12, L 31 – The discussion of solubilities should have been in Li et al. (2019)

---

## Referee Report (RR2)

This is a worthwhile study, and I think it should see the light of day (i.e. be published) after some work on presentation. The authors pursue the potential of synthetic compounds that are regularly measured/monitored in the atmosphere as new tracers for quantifying ocean ventilation timescales. New tracers are needed as the most commonly used tracers, the chlorofluorocarbons, are now decreasing in the atmosphere, and their replacement/complement, Sulfur Hexafluoride, is much more difficult to measure and is likely to suffer from regulatory limits in the near future as well. To evaluate alternate synthetic tracers (almost too many to list), the authors used the analytical equipment built to quantify their atmospheric abundances by adapting/attaching a sparging apparatus to the intake. This allowed the authors to evaluate a number of potential tracers in terms of measurement feasibility, likely stability in seawater, and known solubility.

One issue of presentation has to do with the discussion. HFC-125 is easily quantifiable with the Ampule-Medusa system, and based on one solubility estimate, may be stable in seawater. Rather than discount the tracer (p. 15, lines 35 onward), one would ordinarily recommend that empirical (laboratory) studies of HFC-125 solubility (a la Warner and Weiss, 1998) in seawater be performed. With known solubility, then, HFC-125's stability can be addressed. (The lower mean ages from TTDs are not a hindrance, newer tracers will be expected to yield younger ages). The authors don't discuss anything about possible next steps towards pursuing the promising tracers.

Figure 1 can be improved by dropping 39Ar and  $\frac{14C}{2}$  – allowing the distinction between the anthropogenic gases to be easier to see. Tritium-helium might be a good addition.

The same perspective could be applied to PFC-14 and PFC-16. Although Agua-Medusa is not sensitive enough for these gases, perhaps they could be pursued with an electron-capture detector? ECDs love C-F bonds, after all. Looking at Table 5, one concludes that PFC-14 and PFC-16 are too hard to measure, but one wonders if that's true using 'traditional' purge-trap GC-ECD techniques.

The discussion would benefit, greatly, with some discussion of what to do next, is what I'm saying. The second author, in particular, is a known leader in developing custom analytical system for aquatic trace trace gases. It is disappointing to have that expertise not weighing in on the potential of PFCs as future tracers. The bottom line is that Agua-Medusa's utility on in oceanography is limited by its poor sensitivity (relative to ECDs). This lower sensitivity requires a water budget (5 liters, by my estimation) that is simply not sustainable on regular oceanographic cruises.

My recommendation is that a few hard passes through the text are required before the paper is ready for publication. The figure and analyses all make sense. The text is confusing and hard to follow, and the conclusions rely too much on "applicability within Agua-Medusa" of these tracers, rather than on the perspective of "applicability within oceanography". I applaud the intent of the effort and am happy to review a draft that is easier to follow.

---

## Referee Report (RR3)

This is a worthwhile study, and I think it should see the light of day (i.e. be published) after some work on presentation.

This draft is greatly improved over the prior. Unfortunately the ms. is not currently ready. Although the text has been streamlined considerably (and I applaud the first author on the effort) the text is still confusing to follow.

The discussion of TTDs, for example, has been omitted from Section IV and moved to the supplement. Please, at least introduce, in the main manuscript, the idea (somewhere in section 4) of using D/G = 0.2 and D/G = 1.8 as a validity range (Waugh et al., 2003, or one of Hall's or Stoven's or Sonnerup's papers. Oh, and incidentally, this "valid range" is based on what is visible using the CFC / SF6 pair and does not reflect what may actually be happening in the ocean.) Without this introduction, the TTD-based curves in Fig. 10, and the discussion of TTDs on p. 9 (line 32) and p. 10 (line 8) have little explanation and appear to come out of nowhere.

The summary section (and Fig. 10) often disagrees with the final sentence of section 5.3, in which are listed sea surface saturations from the Baltic station:

| CFC-12     | 122 +- 8 %  |
|------------|-------------|
| HCFC-22    | 77 +- 8 %   |
| HCFC-141b  | 74 +- 12 %  |
| HCFC-142b  | 114 +- 2 %  |
| HFC — 134a | 125 +- 23 % |
| HFC – 125  | 252 +- 35 % |

One would ordinarily conclude, from this list, that HCFC-142b, HFC-134a have solubility functions that are approximated reasonably, and that they are stable in seawater.

Jumping ahead to page 11:

For HCFC-141b one should note that the low saturation measured *in this paper* points to a problem with the solubility function, or degradation in seawater, or both.

For HCFC-142b – are the Baltic data omitted from Fig. 10? I interpret the Baltic results as promising and wonder if some of the analytical issues identified in the Med. are the reason for the high (they are profoundly high, not 'slightly', as suggested) values in Fig. 10?

I would recommend that the Baltic measurements be included in the 'Synopsis' for each tracer.

I started the process of identifying individual sentences that are confusing or could use some clarification, but concluded that the list would be too exhaustive. This is the authors' responsibility.

In summary, while the ms. is greatly improved, I would appreciate it if the authors would take three steps: 1) put some more effort into clarifying the presentation, 2) have peers/ colleagues go over it once or twice, and 3) let it sit for a few days, then go over it again before resubmitting. I stand by my original review statement that this is a worthwhile effort that should be published eventually.

---

## Referee Report (RR4)

**Review of Li & Tanhua (OS-2019-101, revised)**

This version of the manuscript is much improved in focus. Many of my previous concerns have been addressed. However, my major issue is with the quality of the reported measurements from the Aqua-Medusa system. As a proof of concept that this instrument can detect the proposed tracers in the water column, I think the manuscript is fine. However, I have low confidence in the quality of the data - primarily based upon the inability to accurately measure CFC-12. The decision on which data are good (quality flag = 2) seems to be based on an undefined cutoff for consistency with the CFC-12 concentrations measured using the standard PT-ECD instrumentation. The assumption follows that the accuracy issue is due to the sampling or extraction, such that good CFC-12 concentrations are a necessary and sufficient condition for the other measured Medusa concentrations to also be good. I would prefer an explanation, supported by data, for the "bad" CFC-12 data. As Tanhua is aware, it is possible to measure one compound accurately but still have issues with other compounds using a PT-ECD system. Without confidence in the accuracy of the tracer data, I find the interpretation of the data to be interesting but not convincing. In my opinion, laboratory experiments conducted using seawater equilibrated with ambient air, sampled into glass ampoules and analyzed in the same manner as the samples from the Mediterranean and Baltic Sea, would greatly improve my confidence in their water column measurements. If I return to their list of five requirements for a transient tracer, I am not convinced of #4 and cannot therefore evaluate #5.

Other specific comments – both editorial and scientific – are listed below.

p. 1, Line 29 -  Ventilation is usually defined as a process, not a time (e.g. the process that transports water and climatically important trace gases from the surface mixed layer into the ocean interior).

p. 2, line 12 – add the word "respectively" for clarity
p.2, line 16+– grammar ("have been" or "might be" imposed); will continue to rise?; "have readily measured"
p. 2, line 19 – None of the US tracer groups have stopped measuring CFC-11. It is clear that the Tanhua group has stopped measuring it. I'd just leave out the discussion entirely. In addition, Lee & Bullister found evidence for the degradation of CFC-11 in permanently anoxic waters (e.g. the Black Sea), but not in the waters of the ODZs in the open ocean.
p. 22, line 27 – citation for the need for multiple tracers?

p. 3, line 8 – stable and stability are repetitive. Be more clear about what is meant by stable chemical structure. Long  atmospheric lifetimes are indicative that a compound is probably stable to processes such as UV degradation of bonds even in the stratosphere.
p.3, line 20 – "with" not "to"

p 3, line 23+ - As the authors point out, a comparison of the surface saturations with $CCl_4$ only suggest that the medusa tracers are perhaps more conservative than $CCl_4$. That does not mean they are conservative. A better comparison would be to CFC-12 which is known to be conservative.

p.4, line 26 – Clarify that Medusa was developed and utilized to measure volatile gases in the atmosphere.

p. 5, line 16 – "empirically" implies that improvements were made without any logical thought.
p. 5, line 29 – I suggest "measured at both adjacent stations located 15 nm away along the cruise track"
p. 5, line 32 – Heating decreases the solubility  - this results in the gases leaving solution into the headspace over a long period of time, but also increases the fractionation between the purging gas bubbles and the sample.

p. 6, line 32 – the precisions seem reasonable (Table 1). Are they based upon only one duplicate pair of ampoules? (A separate  comment on Table 1 is that the number of significant figures seems rather high for the detection limits given the precision)

p. 7, line 10 – sentence needs an "and" for the final phrase
p. 7, line 13 – Interpretation of Transient Tracer Distributions ?
p. 7, line 16 – At the older end of the time range, the detection limit and precision are also important.
p. 7, line 21 – Others publications refer to the tracer age as the partial pressure age to distinguish it from other possible ages defined by the tracers.

p. 8, line 14 – add "and SF6 concentrations"
p. 8, line 19+ - I thought Atlantic Water entered the Mediterranean Sea at the surface. It is not clear to me why it should take a long time to equilibrate with the atmosphere.  The typical reasons for undersaturation are entrainment of waters from below the mixed layer or cooling of the surface layer, with n=both processes occurring at rates faster than gas exchange can re-establish equilibrium.
p. 8, line 27+ - I suggest the authors apply more rigor to the QC process.  What is the definition of "inconsistent"? Greater than 3 x precision difference? As presented, it seems arbitrary. Are there correlations between the concentrations of CFC-12 and the Medusa tracers in the samples labeled as 5 that could help identify the issues? When I plot the Aqua Medusa tracers vs the Aqua Medusa CFC-12, there is significant overlap between good (flag=2) and bad (flag=5) data for HCFC141a and HCFC142b  (i.e. the concentration ratios are consistent).  There is less correlation for the other medusa tracers.
p. 8, line 34 – The data in the supplemental spreadsheet  (Table S6) are reported in concentration units.  These values are for the partial pressures. You should make this explicit in the manuscript.

p.9, line 2 – consistent with your expectations of tracer concentrations in the deep waters.

p. 9, line 4 – This sentence needs to be re-written for clarity. Proximity to equilibrium with the atmosphere is not a factor – perhaps an indicator?

p. 9, line 12 – annual basis

p. 9, line 13 - Why are the CFC-12 saturations so high in the Baltic Sea? Why do the saturations of the other gases vary so greatly? Without some explanation, I have little confidence in the data quality. Entrainment and surface warming would affect all of the tracers equally.

P. 9-10 – If the 1-G TTD was representative of the processes controlling the distributions of the tracers in the Mediterranean Sea, all of the data plotted in the panels of Fig. 10 should fall along the same $\Delta/\Gamma$ line. Even the $SF_6$ - CFC-12 data fall into the region of $\Delta/\Gamma > 1.8$.

I agree with some of what the authors conclude about the feasibility of using these tracers in future oceanographic studies. The real issue for me is whether the ampoule sampling combined with the Aqua-Medusa system is capable of being utilized for these measurements. This is where I would focus my efforts in the future.

---

## Editor Decision (ED1)

PC comments on version 6

Following a second review by reviewers, the authors have made an extensive series of changes to the paper (thank you!) and I believe it is now suitable for publication. There are still obviously problems with the measurements of these compounds, and I doubt the reviewers will be completely convinced by their arguments, but there seems little point in delaying the paper any longer. There is certainly a need for additional transient tracers for oceanographic use, and this paper, even with the limitations described, will help other researchers improve our measurement capability of these compounds.

I have a number of suggested changes to the wording that I believe will improve the readability of the paper. If the authors agree with these, then I do not need to see it again.

Abstract, line 20-21: "for the first time, with reproducibility studies using Baltic Sea water. Based on historical data, the surface saturation …"
Line 22: Delete "based on historical data."
Line 23: "found currently to be" line 25: "HCFC-22 is likely unstable….which compromises its potential…"
Lines 27-29: "We were not able to fully evaluate the potential of HFC-125 and HFC-23 as tracers due to inconclusive results regarding their solubility and stability in seawater, along with potential analytical challenges."
Line 30: "alternative compounds with similar…"

P2, line 1: "is based on"
Line 7: Delete "as oceanic transient tracers"
Line 8: "ventilation, mixing and circulation."
Line 15: Delete "oceanographic transient"
Line 19: "concentration" not "concentrations"

P3, lines 4-5: "histories of these potential alternative tracers … allow us to examine using one or several of them to replace…"
Line 11: "1200°C, while the rate of"
Line 26: "suitable" not "suited"

P6, lines 27-28: "in the sample; samples with a higher amount of tracer have better…"

P7, last sentence section 3.5: "two different instruments, at roughly the same depth and ignoring their different station positions, is 5.9 +/- 4.6%"

P8, line 11: "adding the other compound"

P12, line 2: "in this study, giving us confidence in our ability to accurately measure"

P13, line 28: "potential;" not "potentials"

Line 30: "the measurements although it has not been identified as unstable;"
Lines 34-35: "analytical methods, using either a more sensitive"

---

## Author Response (AR2)

**Topic Editor Decision: Reconsider after major revisions (04 Jun 2020) by Piers Chapman**

Comments to the Author:

I have now received three reviews of the revised paper, two of which suggest that it needs considerable work before it can be published. In the circumstances, I ask the authors to consider carefully the comments of the reviewers and then submit a revised manuscript, as the information is certainly of considerable importance to researchers wanting to use similar tracers to investigate water movement and mixing rates. At this stage I am not sure if the work required counts as a major or minor revision, and the paper may need to be sent out again for review.

**Dear editor and referees,**

We greatly thank three referees for your constructive suggestions and comments. Below, we address all the comments and describe our responses to them one by one. In the revised manuscript, all changes from the original text are marked.

The main result from this study is that although a couple of the investigated tracers might be useful as oceanic tracers, at least under some circumstances such as cold waters, etc., none of the tracers would fully qualify at this time. Of the four basic requirements, 1) we have a good handle of the input function in the atmosphere through the work of the "atmospheric community". 2) for the solubility, we recognize that targeted work on this is needed for all compounds, 3) for stability we have an idea which ones are possibly stable, but more work is needed here too (laboratory experiments under controlled conditions could be one way), and 4) the analytical limitations in the current configuration are described, also here laboratory experiments are needed to fully understand/improve the system. In summary, we are not quite yet in the position to recommend targeted observations of these tracers, but this study provides a useful summary of the current knowledge, including the new research reported on in this manuscript, to guide further, more targeted experiments.

With this in mind, we hope that the manuscript is now acceptable for publication in Ocean Science.

**os-2019-101-referee-report-1**

This manuscript has improved significantly and I think now suitable for publication. I only have a few minor suggestions.

Pg 7 lines 21-25. This is material more suited for introduction and not methods section. I think could just start with statement on using TTD framework to interpret data

Response: The texts to introduce ventilation have been moved as the first paragraph in the Introduction.

Pg 13, line 28. I think more accurate to replace "considered" with "used only", as the short lifetime could be very useful if combine with age from CFC-12 or $SF_6$.

Response: Done.

Pg 14, last paragraph. This is a bit repetitive as just mentioned above and will then be mentioned again in conclusions. I think should be removed or used to replace text in conclusions (I like description here better than longer discussion in conclusions).

Response: The last paragraph has replaced some texts in conclusions. And the conclusion section has been rewritten.

**os-2019-101-referee-report-2**

Much of the interpretation of tracer ages and TTDs would not warrant publication on their own. It is the development of the methods to focus on the potential new transient tracers that would be the important contribution of this research. However, the authors have not presented enough information on the analytical techniques to convince me that the compounds can be measured at the precision and accuracy necessary to identify the compounds as useful tracers.

This manuscript is a companion piece to a previous study by these two authors (along with additional co-authors) to identify potential new anthropogenic transient tracers for studies of ocean ventilation on decadal timescales. While the first manuscript (Li et al., 2019) focused on reconstructing the atmospheric histories and estimating the solubilities in water and seawater for several halogenated compounds, this manuscript is focused on the feasibility of measuring these compounds in seawater, demonstrating whether they behave conservatively, and finally rating their potential as tracers. In my view, the key to the publication of this manuscript is to have demonstrated the capability to measure the seawater concentration of these tracers. As such, the authors should have focused on the methods necessary to produce high-quality "Medusa" tracer data. In the data presented from the Mediterranean Sea, only 9 of the 21 samples are evaluated as

good. Another 10 are given the flag of 5. The data from the Baltic Sea may be better, but the authors spend very little time on their interpretation. The remainder of the manuscript consists of hypothetical data interpretation (e.g. Fig. 5, Figs. S1 and S2) that is not worthy of publication on their own. I cannot recommend that this manuscript be published without major revisions that may include further method development. Developing the implementation of new tracers for studying the ocean is important, and I encourage the authors to understand the issues affecting their measurements. I would suggest some laboratory measurements using seawater samples equilibrated with atmospheric gases as a potential method for future studies.

Response: Many thanks for your comment. Before the Medusa-Aqua system used for the measurement of seawater samples from the Mediterranean Sea (cruise MSM72) and the Baltic Sea (cruise AL516), it has been improved. The improved processes have been added in the Supplement (Section S3, Figures S1-S7, Tables S3-S5). For data from the Baltic Sea, we only collected a few samples for analysis of reproducibility and we have interpreted them all. The results can be found in the new Table 1 and Table S4b.

The suggestion for laboratory experiments using seawater with equilibrated sea-water samples is certainly a very good idea. The main result from this study is that although a couple of the investigated tracers might be useful as oceanic tracers, at least under some circumstances such as cold waters, etc., none of the tracers would qualify at this time. Of the four basic requirements, 1) we have a good handle of the input function in the atmosphere through the work of the "atmospheric community". 2) for the solubility, we know that targeted work on this is needed for all compounds, 3) for stability we have an idea which ones are possibly stable, but more work is needed here too (laboratory experiments under controlled conditions could be one way), and 4) the analytical limitations in the current configuration is described, also here laboratory experiments are needed to fully understand/improve the system. In summary, we are not quite yet in the position to recommend targeted observations of these tracers, but the current study provides a useful summary of the current knowledge to guide further, more targeted experiments. With this in mind, we argue that the results can, and should, be published.

In addition to the major scientific issue, the manuscript has many other issues with both writing and with some of the interpretation. I have listed many of them below:

p. 1, L21 – "potential be tracers"

Response: "potential as tracers"

p. 1, L31 - The source function for bomb tritium is not well constrained for much of the ocean. It was the 3He-3H tracer pair which are used as transient tracers. On the pedantic side, $^{39}$Ar is not a "transient" tracer as it's assumed to be at steady-state.

Response: "Tritium ($^3$H)" has been changed to "Tritium-Helium ($^3$H-$^3$He)", and similar change for Figure 1. For $^{39}$Ar, that is obviously correct. In a previous paper (Stöven et al., 2015), we used the terms "natural radioactive transient tracers" and "chronological transient tracers" to make this distinction. Maybe not perfect as the term "transient" is still in there, but at least it is a step in the right direction. We have now made this clear in the manuscript by adding ", although $^{39}$Ar is assumed to be in steady-state and cannot be regarded as a transient racer in the true meaning of the word" at the end of the sentence.

p.2, L 3 – the industrial use of CFC-12 was phased out; assign a year to the Montreal Protocol

Response: The sentence "the use of CFC-12 was phased-out as a result of the implementation of the Montreal Protocol" has been changed to "the industrial use of CFC-12 and CFC-11 were phased out as a result of the implementation of the 1987 Montreal Protocol …."

p.2, L9 – Although restrictions on $SF_6$ may be implemented, the concentrations in the atmosphere will continue to rise for the foreseeable future due to its long atmospheric lifetime. Note that the PFCs also have high GHG potential, yet the authors conclude that they should be considered as transient tracers.

Response: $SF_6$ is the compound that has the highest Global Warming Potential (GWP) among halocarbons and related compounds (including CFCs, HCFCs, HFCs, Halons, PFCs, HFEs, HGs, etc.). It has a GWP of 23,507 over 100 years, while PFC-14 and PFC-116 have GWPs of 6,626 and 11,123 respectively (Hodnebrog et al., 2013). Compared to $SF_6$, the GWPs of PFCs are relatively lower.

The corresponding sentences "However, some local restrictions are in place for the production and use of $SF_6$ due to its very high global warming potential that may restrict $SF_6$ to be an oceanic tracer in the future." have been revised as "Due to its very high global warming potential, some local restrictions on the production and use of $SF_6$ implemented. However, the concentrations of $SF_6$ in the atmosphere continue to rise due to its long atmospheric lifetime."

p. 2, L15 – Note that CFC-11 is a Level 1 (required) measurement for the US GO-SHIP program (https://usgoship.ucsd.edu/level-1-data/). The measurements of tracers are complicated – fortunately, the Bullister and Wisegarver (2008) paper referenced by the authors describes an analytical system capable of measuring seawater concentrations of CFC-11, CFC-12, $SF_6$, and $CCl_4$ precisely and accurately aboard a research vessel.

Response: Although CFC-11 is a Level 1 required measurement for the GO-SHIP program, the application of CFC-11 as an oceanic transient tracer has been largely reduced during the past decade compared to CFC-12 and $SF_6$, mostly due to the complications on simultaneous observations (for one sample) in seawater. Even though Bullister and Wisegarver solved the problem, few others have used that set-up. We acknowledge that CFC-11 is, and has been, an important transient tracer.

We changed the related sentences "In addition, trichlorofluoromethane (CFC-11), 1,1,2-trichloro-1,2,2-trifluoroethane (CFC-113), and carbon tetrachloride ($CCl_4$) had been extensively used as transient tracers, but have largely been discarded. CFC-11 was found to be degraded in anoxic marine waters (Bullister and Lee, 1995) and has a time-history similar to that of CFC-12. Besides, the simultaneous

measurement of $SF_6$ and CFC-11 is complicated. Both CFC-113 and $CCl_4$ have been found to be degraded in warm waters (Roether et al., 2001) as well as in low oxygen waters (Wallace and Krysell, 1989; Huhn et al., 2001)." to "CFC-11 has been extensively used as a transient tracer but found to has a time-history similar to that of CFC-12 and be degraded in anoxic marine waters; the simultaneous measurement of $SF_6$ and CFC-11 is complicated, so that the use of CFC-11 is being reduced. In addition, 1,1,2-trichloro-1,2,2-trifluoroethane (CFC-113) and carbon tetrachloride ($CCl_4$) have been extensively used as transient tracers, but have now been largely discarded since they have been found to be degraded in warm waters (Roether et al., 2001) as well as in low oxygen waters (Wallace and Krysell, 1989; Huhn et al., 2001).".

p. 2, L20 – There should never be a 1-sentence paragraph

Response: The sentence has been added to the end of the last paragraph.

p.2, L 24 – "Well-quantified sources and sinks" instead of "known input function"

Response: "known input function" has been changed to "well-quantified sources and sinks".

p.2, L 29 - two phrases beginning with "as" makes the sentence confusing to read

Response: "As the replacements of CFCs, the atmospheric abundances of most HCFCs and HFCs are increasing, as are the concentrations of PFCs." has been changed to "The atmospheric abundances of most HCFCs and HFCs are increasing, as are the concentrations of PFCs."

P.2, L 35 - CFC-12 is difficult to use by itself as a tracer; with some caveats, it can be used as a tracer pair with e.g. $SF_6$. Based upon the reported blank level (0.48 fmol/kg), the utility of $SF_6$ extends back about 50 y instead of 100 y.

Response: We are sorry for your misunderstanding. "1-100 years" here does not mean the validation period of $SF_6$ but means that in this seawater timescale range only $SF_6$ is a relatively reliable transient tracer while $^3H$-$^3He$ and CFC-12 are with constraints. This is also the reason why we explore and evaluate the potential tracers in this range considering that tracer pairs are more validated to estimate ages.

Blank values of $SF_6$ are different depending on various analytical technologies. The blank value of $SF_6$ is 0 fmol/kg and the detection limit is 0.03 fmol/kg when measured by PT-GC-ECD (Stöven, 2011). Therefore, based upon the detection limit, blank value, and source function, the tracer potential timescale range of $SF_6$ was estimated as 1-60 years as shown in Figure 1.

In the revised manuscript, "timescale range of 1-100 years" has changed to "timescales less than 100 years". We note that several groups working with GC-ECD systems achieve considerable lower blank levels, although the reported blank level (0.48 fmol/kg), i.e., the detection limit mentioned in the original Table 3 (now Table 1), is relatively high value because such a value is 19 % of the maximum value of $SF_6$ (2.52 fmol/kg) from measurements on cruise MSM72. Therefore, the current Medusa-Aqua system is still not suitable to measure $SF_6$ in seawater.

p. 3, L8 – see https://www.ncbi.nlm.nih.gov/pubmed/12608868 for a marine natural product with a C-F bond.

Response: "no known marine natural products contain C-F bonds and" has been removed.

p. 3, L9 – grammar: "reasoning true for PFC-116"

Response: "This is reasoning true for PFC-116" has been removed.

p.3, L 19 – replace of with to; where is "here"?

Response: The corresponding sentences "Judged against their environmental total lifetimes, the oceanic contributions of these compounds are small enough to be neglected. But here the ocean partial lifetimes were calculated only considering the chemical degradation process." have been changed to "Judged against their environmental total lifetimes, the oceanic contributions to these compounds are small enough to be neglected. Note that the partial atmospheric lifetimes with respect to oceanic uptake in Table S1 were calculated only considering the chemical degradation process."

p. 3, L 24 – when I went to this ftp site, the data were available through 2004 and the only HCFC data were for HCFC-22

Response: You could find data for HCFC-22 and HCFC-142b from the GOMECC-1 in 2007. The corresponding text "National Oceanic and Atmospheric Administration (NOAA) cruises in 1992-2008 (ftp://ftp.cmdl.noaa.gov/hats/ocean/, last access: 20 January 2020)." has been changed to "National

Oceanic and Atmospheric Administration (NOAA) cruises in 1992-2004 (ftp://ftp.cmdl.noaa.gov/hats/ocean/, last access: 20 January 2020) and the Gulf of Mexico and East Coast Carbon Cruise (GOMECC) in 2007 (https://seabass.gsfc.nasa.gov/cruise/gomecc-1, last access: 10 June 2020)."

p. 3, L27+ - If these studies of degradation in soils have no relevance for CFC-12 in the seawater, why should they have relevance for the Medusa tracers? This contributes nothing to this manuscript.

Response: Although these studies focused on biodegradation in freshwater and soils, these are the only biodegradation information related to the Medusa tracers we could find in the related environment. Anyway, the paragraph related to the biodegradation of Medusa tracers and the original Table 2 have been moved to Supplement (Section S2 and Table S2).

p. 3, L. 36 – "not enough information…"

Response: "no enough information" has been changed to "not enough information".

p.4, L 13 – grammar: "…rendering transient tracers to penetrate…"

Response: "rendering" has been changed to "which causes".

p. 4, L14 – TTDs assume a time-invariant circulation. In other words, they should not be expected to produce useful information (i.e. Sec. 5.5 is just a data exercise)

Response: Although the assumption of time-invariant ventilation is not valid for the Mediterranean Sea, the TTD model can produce indicative results to understand the mean ages estimated from Medusa tracers (assuming $\Delta/\Gamma$ =1.0) and their comparison to those estimated from CFC-12 and SF$_6$. Based on this, we have moved Section 5.5 and Figure 11 to the Supplement as Section S5 and Figure S11.

p.4, L 25 – Lobert et al., 1995 only reported HCFC-22; Beyer et al., 2014 only report CFC-1301. How are these relevant specifically to HFC-134a and HFC-125?

Response: Please see Table B1 in Beyer et al., (2014) for the information of HFC-134a and HFC-125. The original publication (Sousa and Bialkowski, 2001) reported HFC-134a and HFC-125 has also been added in the revised manuscript.

p.4, L 27-28 – citation for the precision of MS vs. ECD?

Response: The sentence "The MS as a detector is becoming increasingly popular since the sensitivity is approaching that of an ECD." has been replaced by "HCFCs and HFC-134a measurements by GC-MS in seawater samples have also been reported in previous studies (Lobert et al., 1996; Ooki and Yokouchi, 2011)."

p. 4, L 28-29 – The last sentence in this paragraph seems to be misplaced.

Response: The last sentence has been removed.

p. 4, L 30 – Shortened not shorted

Response: "shorted" has been changed to "shortened".

p. 4, L 32 – An important component of the Medusa system is the trapping system – I wouldn't say the system is "based" upon it.

Response: "This system is based on trapping of the volatile gases on two traps kept at accurately controlled temperatures" has been changed to "This system includes two traps kept at accurately controlled temperatures to trap the volatile gases".

p. 5, L 28 – This is the first mention of the samples from the Baltic in the manuscript. It seems like a late addition.

Response: In fact, the Baltic Sea samples were first mentioned on L21 on this page (in Section 3 not only Section 3.1). Samples took from here were only for reproducibility. But now we added a sentence on sampling in the Baltic Sea in Section 1.4: "In addition, we report from sampling at a shallow station in the south-west Baltic Sea.", which is the last sentence of the paragraph.

5, L 32 – I doubt that the glass ampoule was closed with a screw.

Response: Followings are the photo and the schematic figure of the stainless-steel mounting system for sampling onboard. The left end was first connected to the outlet of the Niskin bottle. After seawater flushing the glass ampoule three times, the right end was closed with a screw. In order to avoid

misunderstanding, "After removing and closing the ampoule with a screw, the ampoule was flame-sealed as soon as possible under…" has been changed to "The ampoules were flame-sealed immediately after sampling under…".

[Figure]

p. 6, L 10 – Is the difference in bubble size noticeable over the depth of the ampoule?

I could understand placing the purge tube near the bottom of the ampoule to increase the physical stirring of the entire water sample. In addition, the bubbles are primarily responsible for stripping the compounds out of the sample – exchange across the gas-water interface would be extremely slow.

Response: It depends on the contact state of the purge tube with the ampoule bottom. Bubble sizes are usually similar over the depth of the ampoule, except for the nitrogen-seawater interface where the bubbles may be connected as a big one and then disappear.

p. 6 and Fig. 2 – The manuscript and/or figure caption need more details. For example, I assume N1, N2 and N3 are the Nafion driers mentioned in the text. If the Methods section is the major component of the paper, elaborate.

Response: For the manuscript, "When introduced into the Medusa, two Nafion dryers of 1.8 m length and one Nafion dryer of 0.6 m length are used to remove water vapor from the samples" has been changed to "Water vapor is removed from the sample by passing the gases through two Nafion dryers (N1 and N2) of 1.8 m length and one (N3) of 0.6 m length".

We also added more descriptions at the end of Figure 2 caption: "N1-N3 are the Nafion driers and V1-V8 are multiport valves. P1 and P2 are pressure transducers (100PSI-A-DO, All Sensor Corporation, Morgan Hill, CA)."

p. 6, L13 – In this method, approximately 2 L of gas is used to purge 1.3 L of sample. In comparison, the Bullister and Wisegarver CFC/$SF_6$ system purges a 200-cc sample with approximately a liter of UHP N2. It's not surprising that it takes multiple purges.

Response: We also tried to purge a ~1.3 L sample with approximately 6 L of $N_2$ for a single purge. It still needs another 1-2 purges to complete the whole purge process. One of the reasons is the larger solubility of the slightly higher polarity of the (most) of the compounds in this study.

p. 6, L 33 – What is the "stripping efficiency" for the first purge? i.e. how much of the total tracer remains in solution?

Response: "The averaged stripping efficiencies for the first purge of seawater samples from cruise MSM72 are 92.5 ± 5.4 % for CFC-12, 76.6 ± 7.9 % for HCFC-22, 87.5 ± 8.3 % for HCFC-141b, 83.2 ± 10.6 % for HCFC-142b, 83.1 ± 7.5 % for HFC-134a, and 88.9 ± 7.5 % for HFC-125.", which has been added in the revised manuscript.

p. 7, L3 – Clarify what is meant by "noises"

Response: The "noise" here means the one in "signal-to-noise". In order to avoid misunderstanding, "The detection limits… based on the signals corresponding to the blank values or noises plus ten standard deviations." has been changed to "The detection limits… based on the signals corresponding to the blank values plus ten standard deviations.".

p. 7, L16 – "Larger" rather than "bigger"; a single purge

Response: "due to using a bigger sampling volume if only considering purge once" has been changed to "due to its larger sampling volume if only considering a single purge".

p. 7 L 18 – Are the blanks additive? Since each analysis of a single purge has an error associated with it, don't those errors add up (or at least the square root of the sum of the blanks squared)?

Response: Good question. It depends on if the blank is in the Medusa system itself, in which case they do add up, or if it is associated with blank values from the ampoule and cracker, in which case they don't add up. In our case, it is only HFC-23 that falls in the second category. For other Medusa tracers, it is already the total uncertainty considered since the precision was calculated as the reproducibility of two measurements at the same depth, which has included the multiple purge errors.

p. 8, L 6 – monotonically

Response: The sentence "Atmospheric histories of tracers should be monotonic increasing in the atmosphere for ideal applicability" has been changed to "For ideal applicability, atmospheric histories of tracers should increase monotonically in the atmosphere".

p. 8, L8 - Why does the HFC-125 source function start in the mid-1990s? In Li et al. (2019) there is a reconstructed atmospheric concentration prior to that.

Response: The reconstructed atmospheric history of HFC-125 started from the 1970s in Li et al. (2019), but atmospheric mole fractions of HFC-125 before 1990 were very close to zero so that they do not show well in Figure 4.

p. 8, L25 – What sets the maximum useful age? Blank level? Source function?

Response: The maximum useful age is decided by the detection limit, blank value (i.e. the quantification limit), and source function.

p. 8, L31 – I'm not sure what evaluated to be transient tracers means. I would probably say demonstrated to be useful (and that includes easily measured).

Response: The sentence "As CFC-12 is limited to be used as a tracer in the upper ocean, PFCs will obtain more attention if they are evaluated to be transient tracers in the ocean." has been removed.

p. 9 first paragraph – If both figures referred to in a paragraph are in the supplemental material, then perhaps that is where the paragraph belongs as well.

Response: The paragraph has been moved to Supplement as part of Section S4.

Section 5.1 – The authors stress the importance of the WML for determining the long-term average saturation, then never return to this concept. I'm not sure why this is here, since they evaluate the Medusa tracers by their surface saturation (Section 5.3)

Response: We are sorry for your misunderstanding. In Section 5.1, we addressed the WML and estimated the historical surface saturation of CFC-12 and $SF_6$ in the WML from 1987 to 2018. Such saturation value 94 % was also used for Medusa tracers for following calculations of their tracer age and mean age. While in Section 5.3, only surface saturation of Medusa tracers from cruises in 2018, not the historical one, was described. In order to make it clearer, we have changed "For the following calculations…" to "For the following calculations of ages and evaluation of stability…" for the last sentence in Section 5.1.

In fact, the result of the "historical surface saturation" of transient tracers (i.e. CFC-12 and $SF_6$) in the Mediterranean Sea is a useful number for other studies working on ventilation times in the Mediterranean Sea. This is thus relevant to report on by its own right.

p. 9, L 13 – Examples of density profiles – rather than exemplary

Response: "two exemplary density profiles" has been changed to "two examples of density profiles".

p. 9, L 23 – Takes a long time or has a long time or has been isolated from the atmosphere for a long time?

Response: "where the inflowing Atlantic Water (to the Mediterranean Sea) has a long time to equilibrate with the atmosphere" has been changed to "where the inflowing Atlantic Water (to the Mediterranean Sea) takes a long time to equilibrate with the atmosphere."

p.9, L29 – "generally comparable" is meaningless. The authors should plot CFC-12 (MS) vs. CFC-12 (ECD), and let the reviewer decide if they are comparable.

Response: "generally comparable with" has been changed to "compared with". Note that the comparison is given in Figure 8.

p. 9, L31 – Is 5.9% an absolute difference? This is much larger than the precision of the measurements. How are data flagged as "good"? Note that only 9 of 21 samples are classified as "good".

Response: 5.9 % is the average value of the following 18 errors in Table A1 (below) ignoring pressure differences and distance differences of closed stations.

**Table A1.** Misfit of CFC-12 observations (ppt) measured by Medusa-Aqua system and PT-GC-ECD at closed stations

| Medusa-Aqua system | | PT-GC-ECD | | | PT-GC-ECD | | |
|---|---|---|---|---|---|---|---|
| Profile 52 | | Profile 51 | | | Profile 53 | | |
| pressure | CFC-12 | pressure | CFC-12 | Error52-51 | pressure | CFC-12 | Error52-53 |
| 103.86 | 470.4 | 103.4 | 482.5 | 2.5% | 104.4 | 462.3 | 1.7% |
| 508.79 | 315.5 | 508.4 | 315.8 | 0.1% | 507.3 | 327.7 | 3.8% |
| Profile 84 | | Profile 83 | | | Profile 85 | | |
| pressure | CFC-12 | pressure | CFC-12 | Error84-83 | pressure | CFC-12 | Error84-85 |
| 206.2 | 388.8 | 203.4 | 381.1 | 2.0% | 207.2 | 413.1 | 6.1% |
| 1266.5 | 164.0 | 1265.0 | 183.1 | 11.0% | 1264.5 | 186.5 | 12.9% |
| 2025.9 | 174.0 | 2023.9 | 197.4 | 12.6% | | | |
| 2298.7 | 209.8 | 2279.1 | 228.1 | 8.3% | | | |
| Profile 106 | | Profile 105 | | | Profile 107 | | |
| pressure | CFC-12 | pressure | CFC-12 | Error106-105 | pressure | CFC-12 | Error106-107 |
| 507.6 | 269.3 | 507.9 | 308.8 | 13.7% | 5073 | 274.0 | 1.7% |
| 759.8 | 230.5 | 759.6 | 252.7 | 9.2% | 7594 | 236.4 | 2.5% |
| 1012.0 | 224.8 | 1012.6 | 232.2 | 3.2% | 1012.3 | 231.9 | 3.1% |

We added descriptions before 5.9 %: "We performed a two-step quality control procedure on the medusa data where, in the first step outliers were flagged, and in a second step we flagged data where the Medusa-CFC-12 values are inconsistent with the CFC-12 values from PT-GC-ECD (Fig. 8). This process led to 9 samples in the Mediterranean that meets all these criteria and have similar concentration as the PT-GC-ECD observations.". Part of the text now in Section 5.2 was moved from Section 5.3.

Data flagged as "good" or "2" are data for sampling sites that measured CFC-12 by Medusa-Aqua system matched the one by PT-GC-ECD, see description in caption of Figure 8.

Section 5.2 – This section is entitled "Observations of Medusa Tracers in Seawater" yet the Medusa tracers are never discussed. The final sentence points to Fig. 9.

Response: At the end of the paragraph in Section 5.2, we added sentences "Concentration ranges of Medusa tracers are 16.1-129.4 ppt for HCFC-22, 2.1-18.7 ppt for HCFC-141b, 5.0-29.0 ppt for HCFC-

142b, 41.0-217.4 ppt for HFC-134a and 4.9-12.8 ppt for HFC-125. The concentrations of Medusa tracers decreased from the surface to the intermediate layer and then increased in the deep layer, consistent with the well-ventilated Mediterranean deep waters.".

p. 10, L 5 – In Section 5.3, the authors wrote that there is too much variability in the surface saturation to be useful.

Response: We did not write just that, we are trying to tell readers that surface saturation is influenced by many factors. More measurements are needed to better discuss the results of surface saturation. Note that all near-surface sampling points are flagged "5", a fact that, although described, was not clear enough described. We have now removed the discussion of surface saturation for the Mediterranean samples.

p. 10, L7 – Do not forget bubble injection and/or vertical mixing

Response: "bubble injection and/or vertical mixing" has been added.

p. 10, L10 – If the Medusa Aqua measurements of CFC-12 are 20 % too low at the surface of the Mediterranean Sea, how can the reviewer have confidence in any of the other tracer measurements? Unless the authors can provide a reasonable explanation for this discrepancy for CFC-12, the remainder of the data are suspect in my opinion. Likewise, the surface CFC-12 saturations of 122% in the Baltic Sea also seem unreasonable. Since data from only one station in the Baltic Sea is reported, I assume that the uncertainties in the surface saturation are an indication of the precision of the Medusa tracer measurements. These are also not very reassuring for the most part (except for HCFC-142b at 2 %).

The manuscript would greatly benefit from measurements of the Medusa tracers in the laboratory under controlled settings. I need to be convinced that Medusa Aqua a reliably measure the tracer concentration under laboratory conditions in order to have confidence in the reported measurements from the Mediterranean and Baltic Seas.

Response: Since all surface samples from the Mediterranean Sea have been flagged as "5", we should not have discussed the surface saturation of Medusa tracers in the Mediterranean Sea in the first place. This has nows been removed from the original Table 5 (now Table 3). We now focus on samples from the Baltic Sea where we have surface saturation of Medusa tracers in one surface sample and two bottom samples. Note that the bottom waters (at about 23.5 meters depth) can be considered as recently ventilated as this water is ventilated on an annual base.

We agree that the reviewer's suggestion to work with a number of seawater samples in a controlled laboratory environment would be the best solution, and something we will try to realize in the future. We suggest that the information in the current article is useful for reporting on the initial field observations to better guide future work, in addition to be (in our opinion) relevant results by its own right.

p. 10, L 25 – grammar: easy to soluble

Response: It has been changed to "ease of solubility".

p. 11, L 21 – Why not take the approach of finding an IG-TTD with a delta/gamma ratio that matches CFC-12 and SF$_6$ together (instead of assuming the ratio is 1), and then applying this IG-TTD to the Medusa tracers?

Response: In Figure 10f, the $\Delta/\Gamma$ ratio for most sampling sites, especially the surface ones, are over 1.8. Due to the time-variant ventilation, and possibly the influence of more than one major water mass (leading to a bi-modal shape of the TTD), we were not able to find any suitable D/G based on IG-TTD for the Mediterranean Sea based on the CFC-12/SF$_6$ tracer couple. But from Figure 10, we could find that the D/G ratio determined by CFC-12/HCFC-141b and CFC-12/HCFC-142b tracer pairs located closer to that by CFC-12/SF$_6$, pointing out the more promising tracer pairs. This is also another expression of your suggested method.

p. 11, L 30 – The discussion of solubilities should have been in Li et al. (2019)

Response: Section 5.5 has been removed.

**os-2019-101-referee-report-3**

Difficult to follow text, but a worthwhile effort. Would like to see some added discussion into what next steps will be / should be taken

This is a worthwhile study, and I think it should see the light of day (i.e. be published) after some work on presentation. The authors pursue the potential of synthetic compounds that are regularly measured/monitored in the atmosphere as new tracers for quantifying ocean ventilation timescales. New tracers are needed as the most commonly used tracers, the chlorofluorocarbons, are now decreasing in the atmosphere, and their replacement/complement, Sulfur Hexafluoride, is much more difficult to measure and is likely to suffer from regulatory limits in the near future as well. To evaluate alternate synthetic tracers (almost too many to list), the authors used the analytical equipment built to quantify their atmospheric abundances by adapting/attaching a sparging apparatus to the intake. This allowed the authors to evaluate a number of potential tracers in terms of measurement feasibility, likely stability in seawater, and known solubility.

One issue of presentation has to do with the discussion. HFC-125 is easily quantifiable with the Ampule-Medusa system, and based on one solubility estimate, may be stable in seawater. Rather than discount the tracer (p. 15, lines 35 onward), one would ordinarily recommend that empirical (laboratory) studies of HFC-125 solubility (a la Warner and Weiss, 1998) in seawater be performed. With known solubility, then, HFC-125's stability can be addressed. (The lower mean ages from TTDs are not a hindrance, newer tracers will be expected to yield younger ages). The authors don't discuss anything about possible next steps towards pursuing the promising tracers.

Response: The empirical laboratory studies of HFC-125 solubility would be one of the next steps. The possible next steps have been added in the rewritten Section 7 (Conclusions and outlook).

Figure 1 can be improved by dropping $^{39}$Ar and $^{14}$C – allowing the distinction between the anthropogenic gases to be easier to see. Tritium-helium might be a good addition.

Response: We thank the reviewer for the suggestion and have now removed $^{39}$Ar and $^{14}$C from that figure which allows us to scale up the x-axis for better visibility.

The same perspective could be applied to PFC-14 and PFC-16. Although Aqua-Medusa is not sensitive enough for these gases, perhaps they could be pursued with an electron-capture detector? ECDs love C-

F bonds, after all. Looking at Table 5, one concludes that PFC-14 and PFC-16 are too hard to measure, but one wonders if that's true using 'traditional' purge-trap GC-ECD techniques.

Response: We found previous publications that PFC-14 ($CF_4$) analyzed by Gas Chromatography (GC) detector (Varian Model 90P) (Cosgrove and Walkley, 1981) and gas-solid chromatography (Smith et al., 1981). But we have not found any publications where PFCs were measured by PT-GC-ECD in seawater, which could be tried in the near future.

The discussion would benefit, greatly, with some discussion of what to do next, is what I'm saying. The second author, in particular, is a known leader in developing custom analytical system for aquatic trace gases. It is disappointing to have that expertise not weighing in on the potential of PFCs as future tracers. The bottom line is that Aqua-Medusa's utility on in oceanography is limited by its poor sensitivity (relative to ECDs). This lower sensitivity requires a water budget (5 liters, by my estimation) that is simply not sustainable on regular oceanographic cruises.

Response: Measurement of PFCs should be first resolved. We have rewritten the conclusion and outlook section.

My recommendation is that a few hard passes through the text are required before the paper is ready for publication. The figure and analyses all make sense. The text is confusing and hard to follow, and the conclusions rely too much on "applicability within Aqua-Medusa" of these tracers, rather than on the perspective of "applicability within oceanography". I applaud the intent of the effort and am happy to review a draft that is easier to follow.

Response: We thank the topic editor who helped us went through the text. The conclusion and outlook section have been rewritten. We also did the language check several times.

[revised manuscript text omitted]

---

## Author Response (AR4)

**os-2019-101-EC1**

**Topic editor:** The authors have responded to the comments of the reviewers and made a large number of changes to the manuscript. As a result, the paper is somewhat shorter and easier to read than previously and more of the details have been placed in the supplementary material. I believe that the paper now needs only a few relatively minor changes to be suitable for publication, but would like to give reviewers #2 and #3 a chance to comment on the revised version.

Dear editor and referees,

We thank the editor and two referees for your constructive suggestions and comments. Below, we address all the comments and describe our responses to them one by one. In the revised manuscript, all changes from the original text are marked.

To quantify the reproducibility of the analytical set-up and make our results more convincing, we took seawater from the tropical Atlantic Ocean, and let it equilibrate with the atmosphere in the laboratory. The water was then sampled from Niskin bottles in the same way as during a cruise and flame sealed in ampoules, although for this experiment we used 300 mL samples. The reproducibility for CFC-12, HCFC-22, HCFC-141b, and HCFC-142b measurements from four duplicate samples are 3.7%, 2.0%, 3.5%, and 3.4%, respectively. The added results validate the feasibility of the analytical method of ampoule sampling combined with the Aqua-Medusa system and make our results more credible.

With this in mind, we hope that the paper is now suitable for publication in Ocean Science.

Some comments I have are:

1. The abstract does not mention the samples from the Baltic; this seems odd given that these samples are used to justify some of the conclusions in the paper.

Response: The underlined part has been added to the sentence "*HCFC-22, HCFC-141b, HCFC-142b, HFC-134a, and HFC-125 have been measured in depth-profiles in the Mediterranean Sea for the first time, and reproducibility in the Baltic Sea*".

2. Nowhere in the paper do the authors state specifically that at present none of the suggested compounds can be considered to be suitable transient tracers. The last paragraph of the authors' overall response to reviewer #2 is a very clear statement of this (starting "The main result of this study…" and finishing "to guide further, more targeted experiments."). This could be added to the conclusion.

Response: Thanks for the suggestion. The mentioned paragraph has been added as the third paragraph in the Conclusion section.

3. In section 5.2 the authors discuss variability in their CFC-12 measurements. Reviewer #2 queried this, and the authors produced Table A1 in response. While these numbers can presumably be calculated from Table S6, a bit more explanation in the text would help understanding.

Response: This part has been moved up to Sect. 3.5. The underlined part has been added to the sentence "*For such samples, the averaged difference of CFC-12 concentrations measured by the two different instruments is 5.9 ± 4.6 % at roughly the same depth by ignoring their distance differences*.".

4. Like reviewer #3, I feel the paper seems to suggest liabilities of the present Medusa system for seagoing use as much as the problems of using these particular tracers. The authors have made some suggestions for future improvement; these could perhaps be expanded.

Response: The expanded part has been added in the third and fourth paragraphs in the Conclusion section, see the revised manuscript.

**os-2019-101-referee-report-1**

Presentation is greatly improved, however it is still difficult to determine what exactly the authors are trying to say.

This is a worthwhile study, and I think it should see the light of day (i.e., be published) after some work on presentation.

This draft is greatly improved over the prior. Unfortunately, the ms. is not currently ready. Although the text has been streamlined considerably (and I applaud the first author on the effort) the text is still confusing to follow.

The discussion of TTDs, for example, has been omitted from Section IV and moved to the supplement. Please, at least introduce, in the main manuscript, the idea (somewhere in section 4) of using D/G = 0.2 and D/G = 1.8 as a validity range (Waugh et al., 2003, or one of Hall's or Stoven's or Sonnerup's papers. Oh, and incidentally, this "valid range" is based on what is visible using the CFC / SF6 pair and does not reflect what may actually be happening in the ocean.) Without this introduction, the TTD-based curves in Fig. 10, and the discussion of TTDs on p. 9 (line 32) and p. 10 (line 8) have little explanation and appear to come out of nowhere.

Response: Many thanks for your nice reminder. We have added a paragraph (shortly described the TTD and mean age) to the Sect. 4.2 and changed the subtitle "*4.2 Tracer age*" to "*4.2 Tracer age and the Transit Time Distribution (TTD)*". For the "valid range", we do make a point in the manuscript that due to the time-variant ventilation, the validity range are not hard boundaries, and deviations can be conceived, see the first two sentences in the second paragraph in Sect. 5.4. In addition, we revised some sentences in the Sect. 5.4, see the revised manuscript.

The summary section (and Fig. 10) often disagrees with the final sentence of section 5.3, in which are listed sea surface saturations from the Baltic station:

CFC-12 122 +- 8 %

HCFC-22 77 +- 8 %

HCFC-141b 74 +- 12 %

HCFC-142b 114 +- 2 %

HFC – 134a 125 +- 23 %

HFC – 125 252 +- 35 %

One would ordinarily conclude, from this list, that HCFC-142b, HFC-134a have solubility functions that are approximated reasonably, and that they are stable in seawater.

Response: We agree with the reviewer's statement and have revised words in the "**HCFC-142b**" and "**HFC-134a**" paragraphs in Sect. 6, see the revised manuscript.

Jumping ahead to page 11:

For HCFC-141b one should note that the low saturation measured in this paper points to a problem with the solubility function, or degradation in seawater, or both.

Response: We have added a sentence in the "**HCFC-141b**" paragraph in Sect. 6. "*The low surface saturation measured in the Baltic Sea suggests that there is likely an issue with the solubility function.*"

For HCFC-142b – are the Baltic data omitted from Fig. 10? I interpret the Baltic results as promising and wonder if some of the analytical issues identified in the Med. are the reason for the high (they are profoundly high, not 'slightly', as suggested) values in Fig. 10?

I would recommend that the Baltic measurements be included in the 'Synopsis' for each tracer.

Response: The results from cruise AL516 in the Baltic Sea have been added to Fig. 10 as black dots. We removed the word "slightly" from section 6.

I started the process of identifying individual sentences that are confusing or could use some clarification, but concluded that the list would be too exhaustive. This is the authors' responsibility.

In summary, while the ms. is greatly improved, I would appreciate it if the authors would take three steps: 1) put some more effort into clarifying the presentation, 2) have peers/ colleagues go over it once or twice, and 3) let it sit for a few days, then go over it again before resubmitting. I stand by my original review statement that this is a worthwhile effort that should be published eventually.

Response: We did the language check again following these suggestions and hope that it has been improved.

**os-2019-101-referee-report-2**

I have concerns with the accuracy of the measurements.

Review of Li & Tanhua (OS-2019-101, revised)

This version of the manuscript is much improved in focus. Many of my previous concerns have been addressed. However, my major issue is with the quality of the reported measurements from the Aqua-Medusa system. As a proof of concept that this instrument can detect the proposed tracers in the water column, I think the manuscript is fine. However, I have low confidence in the quality of the data - primarily based upon the inability to accurately measure CFC-12. The decision on which data are good (quality flag = 2) seems to be based on an undefined cutoff for consistency with the CFC-12 concentrations measured using the standard PT-ECD instrumentation. The assumption follows that the accuracy issue is due to the sampling or extraction, such that good CFC-12 concentrations are a necessary and sufficient condition for the other measured Medusa concentrations to also be good. I would prefer an explanation, supported by data, for the "bad" CFC-12 data. As Tanhua is aware, it is possible to measure one compound accurately but still have issues with other compounds using a PT-ECD system. Without confidence in the accuracy of the tracer data, I find the interpretation of the data to be interesting but not convincing. In my opinion, laboratory experiments conducted using seawater equilibrated with ambient air, sampled into glass ampoules and analyzed in the same manner as the samples from the Mediterranean and Baltic Sea, would greatly improve my confidence in their water column measurements. If I return to their list of five requirements for a transient tracer, I am not convinced of #4 and cannot therefore evaluate #5.

Response: To quantify the reproducibility of the analytical set-up and make our results more convincing, we took seawater from the tropical Atlantic Ocean, and let it equilibrate with the atmosphere in the laboratory. The water was then sampled from Niskin bottles in the same way as during a cruise and flame sealed in ampoules, although for this experiment we used 300 mL samples. The reproducibility for CFC-12, HCFC-22, HCFC-141b, and HCFC-142b measurements from four duplicate samples are 3.7%, 2.0%, 3.5%, and 3.4%, respectively. The added results validate the feasibility of the analytical method of ampoule sampling combined with the Aqua-Medusa system and make our results more credible. The samples show saturations very close to 100% for CFC-12, and over 100% for HCFC-22, HCFC-141b and HCFC-142b. For HFC-134a, we observed very high concentrations, undoubtedly from contamination by nearby cold-labs. Therefore, we do not use the saturations for this experiment to evaluate the tracers in this manuscript. But these measurements do increase our confidence in the analytical procedure, although it does not really explain why we had issues in the upper water column of the Mediterranean Sea.

We are currently setting up an experiment to reliable (we hope) formulate the solubility for those tracers. This will hopefully be conducted during 2021, but we are experiencing some difficulties to make a reliable standard as some of these compounds are hard to come by.

While we agree that it is indeed possible, and not uncommon, to be able to reliably measure one compound correctly and others not, we prefer to be on the safe side in this manuscript. That means that we used the criteria of "large" deviations from expected CFC-12 concentration to flag ALL

data for those samples. Note that we still show samples flagged "5" in the figures, although with a different symbol.

Other specific comments – both editorial and scientific – are listed below.

p. 1, Line 29 - Ventilation is usually defined as a process, not a time (e.g. the process that transports water and climatically important trace gases from the surface mixed layer into the ocean interior).

Response: "*Ventilation is defined as the time elapsed since the water parcel has left the mixed layer and been transported to the ocean interior. Ocean ventilation and mixing processes play significant roles in climate as they are important processes to propagate perturbations on the ocean surface to the interior.*" has been changed to "*Ocean ventilation is defined as the process that transports perturbations such as the concentrations of tracer gases from the surface mixed layer into the ocean interior. Ocean ventilation and mixing processes play significant roles in climate.*".

p. 2, line 12 – add the word "respectively" for clarity

Response: "*, respectively*" has been added.

p.2, line 16+– grammar ("have been" or "might be" imposed); will continue to rise?; "have readily measured"

Response: The target sentence "*Due to its very high global warming potential, some local restrictions on the production and use of $SF_6$ implemented. However, the concentrations of $SF_6$ in the atmosphere continue to rise due to its long atmospheric lifetime.*" has been changed to "*Due to its very high global warming potential, some local restrictions on the production and use of $SF_6$ has been imposed since 2006. However, the concentrations of $SF_6$ in the atmosphere is still increasing, partly due to its long atmospheric lifetime.*"

p. 2, line 19 – None of the US tracer groups have stopped measuring CFC-11. It is clear that the Tanhua group has stopped measuring it. I'd just leave out the discussion entirely. In addition, Lee & Bullister found evidence for the degradation of CFC-11 in permanently anoxic waters (e.g. the Black Sea), but not in the waters of the ODZs in the open ocean.

Response: The target sentences have been removed and the related sentences have been changed to "*CFC-12, $SF_6$, and CFC-11 can readily be measured onboard research vessels at a reasonable rate from one seawater sample. 1,1,2-trichloro-1,2,2-trifluoroethane (CFC-113) and carbon tetrachloride ($CCl_4$) have previously been used as oceanic transient tracers, but…*".

p. 22, line 27 – citation for the need for multiple tracers?

Response: "*(Stöven and Tanhua, 2014; Holzer et al., 2018)*" has been added.

p. 3, line 8 – stable and stability are repetitive. Be clearer about what is meant by stable chemical structure. Long atmospheric lifetimes are indicative that a compound is probably stable to processes such as UV degradation of bonds even in the stratosphere.

Response: "*One indication of the stability is the stable chemical structure*" has been changed to "*One indication of the stability is the chemical structure and the atmospheric lifetime*".

p.3, line 20 – "with" not "to"

Response: "*associated to their estimates*" has been removed.

p 3, line 23+ - As the authors point out, a comparison of the surface saturations with CCl4 only suggest that the medusa tracers are perhaps more conservative than CCl4. That does not mean they are conservative. A better comparison would be to CFC-12 which is known to be conservative.

Response: We compared the average surface saturation of HCFCs and CFC-12. As seen in the newly added **Table S2 in the Supplement**, the average surface saturation of HCFCs tends to be higher than those of CFC-12. This may suggest that HCFCs are stable enough to be suited as tracers in the ocean. The corresponding paragraph has been revised: "*Another route is to compare surface saturations of a tracer with unknown stability to those of a compound that is known to be stable in seawater, e.g., CFC-12. The average surface saturation of HCFCs tends to be higher than those of CFC-12. This may suggest that HCFCs are stable enough to be suited as tracers in the ocean.*".

**Table S2.** Average surface saturation (%) of CFC-12, HCFC-22, and HCFC-142b from cruises BLAST III, GasEx98, PHASE1, and GOMECC

| Cruise name | Sampling year | CFC-12 | HCFC-22 | HCFC-142b |
|---|---|---|---|---|
| BLAST III [a] | 1996 | -3.0 ± 10.4 | 21.6 ± 74.2 | |
| GasEx98 [a] | 1998 | 0.3 ± 5.8 | 7.6 ± 21.0 | |
| PHASE1 [a] | 2004 | -22.8 ± 114.0 | 1.5 ± 6.7 | |
| GOMECC [b] | 2007 | -0.4 ± 8.6 | 6.8 ± 108.7 | 6.0 ± 13.2 |

[a] National Oceanic and Atmospheric Administration (NOAA) cruises in 1992-2004 (ftp://ftp.cmdl.noaa.gov/hats/ocean/, last access: 20 January 2020); [b] Gulf of Mexico and East Coast Carbon Cruise (GOMECC) in 2007 (https://seabass.gsfc.nasa.gov/cruise/gomecc-1, last access: 10 June 2020).

p.4, line 26 – Clarify that Medusa was developed and utilized to measure volatile gases in the atmosphere.

Response: "*volatile trace gases has been ...*" has been changed to "*volatile trace gases in the atmosphere has been ...*".

p. 5, line 16 – "empirically" implies that improvements were made without any logical thought.

Response: "*empirically*" has been removed.

p. 5, line 29 – I suggest "measured at both adjacent stations located 15 nm away along the cruise track"

Response: "*although they were measured for the nearby stations, 15 nm (nautical miles) away from either direction*" has been changed to "*although they were measured at both adjacent stations located 15 nm away along the cruise track*".

p. 5, line 32 – Heating decreases the solubility - this results in the gases leaving solution into the headspace over a long period of time, but also increases the fractionation between the purging gas bubbles and the sample.

Response: Thanks. We changed the sentence to make this clear. "*Before measurement, each ampoule sample was immersed in a warm water bath at 65 °C overnight to support the transfer of the gases into the headspace and to enhance the purging efficiency.*"

p. 6, line 32 – the precisions seem reasonable (Table 1). Are they based upon only one duplicate pair of ampoules? (A separate comment on Table 1 is that the number of significant figures seems rather high for the detection limits given the precision)

Response: Yes, the first "precision" column showed only one duplicate pair of ampoules from the Baltic Sea. But now we have added the second "precision" column and presented four duplicated pairs of ampoules from the tropical Atlantic Ocean. We have deleted one number of significant figures for detection limits.

p. 7, line 10 – sentence needs an "and" for the final phrase

Response: "*and*" has been added.

p. 7, line 13 – Interpretation of Transient Tracer Distributions?

Response: "*Transient tracer interpreting methods*" has been changed to "*Interpretation of transient tracer distributions*".

p. 7, line 16 – At the older end of the time range, the detection limit and precision are also important.

Response: Yes, that is obviously correct. We added the sentence "*For older waters, and low tracer concentrations, the precision and detection limits will be limiting factors (Tanhua et al., 2008; Stöven et al., 2015)*".

p. 7, line 21 – Others publications refer to the tracer age as the partial pressure age to distinguish it from other possible ages defined by the tracers.

Response: "*Tracer age*" has been changed to "*Tracer age (or partial pressure age)*" on the first line of Sect. 4.2.

p. 8, line 14 – add "and SF6 concentrations"

Response: We added this.

p. 8, line 19+ - I thought Atlantic Water entered the Mediterranean Sea at the surface. It is not clear to me why it should take a long time to equilibrate with the atmosphere. The typical reasons for undersaturation are entrainment of waters from below the mixed layer or cooling of the surface layer, with n=both processes occurring at rates faster than gas exchange can re-establish equilibrium.

Response: We do not really know why we see a constant undersaturation of 6% over time, while in the Atlantic we see time-dependent saturation. But we are offering a possible explanation that is now better explained: "*For CFC-12, this is different from the situation in the North Atlantic Ocean (Tanhua et al., 2008), and could be an indication of the different oceanographic settings where the inflowing Atlantic Water (to the Mediterranean Sea) take a long time to equilibrate with the atmosphere. The constant undersaturation through time is then mainly a function of rapid cooling during winter and entrainment of water from below, rather than a rapid change of the atmospheric concentration that can drive undersaturation that varies over time.*".

p. 8, line 27+ - I suggest the authors apply more rigor to the QC process. What is the definition of "inconsistent"? Greater than 3 x precision difference? As presented, it seems arbitrary. Are there correlations between the concentrations of CFC-12 and the Medusa tracers in the samples labeled

as 5 that could help identify the issues? When I plot the Aqua Medusa tracers vs the Aqua Medusa CFC-12, there is significant overlap between good (flag=2) and bad (flag=5) data for HCFC141a and HCFC142b (i.e. the concentration ratios are consistent). There is less correlation for the other medusa tracers.

Response: We added a new section (Sect. 3.5) to illustrate the QC procedure. We calculated the three times the standard deviations (3σ) of CFC-12 observations for profile pairs 51-53, 83-85, and 105-107. The "inconsistent" means that the misfit of CFC-12 concentrations measured by the Medusa-Aqua system and PT-GC-ECD is more than the 3σ.

While we agree that it is indeed possible, and not uncommon, to be able to reliably measure one compound correctly and others not, we prefer to be on the safe side in this manuscript. That means that we used the criteria of "large" deviations from expected CFC-12 concentration to flag ALL data for those samples. Note that we still show samples flagged "5" in the figures, although with a different symbol. The inconsistency between ECD and Medusa measurements for CFC-12 does indicate an issue somewhere along the chain of operations, likely in sampling or storing that would affect tracers somewhat equal.

p. 8, line 34 – The data in the supplemental spreadsheet (Table S6) are reported in concentration units. These values are for the partial pressures. You should make this explicit in the manuscript.

Response: The last sentence in Sect. 3.4 has been changed to "*The observations of CFC-12, HCFC-22, HCFC-141b, HCFC-142b, HFC-134a and HFC-125 (in pmol kg$^{-1}$) measured by the Medusa-Aqua system in seawater from both cruises are shown in Table S6 with quality flags marked.*" by adding "*(in pmol kg$^{-1}$)*".

p.9, line 2 – consistent with your expectations of tracer concentrations in the deep waters.

Response: "*consistent with the well-ventilated Mediterranean deep waters*" has been changed to "*consistent with our expectations of tracer concentrations, and indicating the well-ventilated Mediterranean deep waters as interpreted by CFC-12 and SF$_6$ (Li and Tanhua, 2020)*".

p. 9, line 4 – This sentence needs to be re-written for clarity. Proximity to equilibrium with the atmosphere is not a factor – perhaps an indicator?

Response: Rewritten as "*The surface saturation of seawater can be an indicator of the stability of a compound in surface seawater or the correctness of the seawater solubility function*".

p. 9, line 12 – annual basis

Response: "*annual base*" has been changed to "*annual basis*".

p. 9, line 13 - Why are the CFC-12 saturations so high in the Baltic Sea? Why do the saturations of the other gases vary so greatly? Without some explanation, I have little confidence in the data quality. Entrainment and surface warming would affect all of the tracers equally.

Response: We are not quite sure why we see these high values in the Baltic Sea for CFC-12. This is an active area, and we do agree that the saturation should be similar for all tracers, so that the saturation of HCFC-142b and HFC-134a looks "about right". We realize that that is not a very satisfactory statement. As mentioned above, we have initiated trials to determine the solubility coefficient of these compounds.

The experiments in the laboratory in January 2021 (Table A1) indicate CFC-12 concentrations very close to saturation, as an indication of reliable measurements. We agree that the variability of the three samples is higher than hoped for.

**Table A1**. Concentrations of CFC-12, HCFC-22, HCFC-141b, HCFC-142b and HFC-134a (in pmol kg$^{-1}$) in seawater samples equilibrated with the atmosphere in the laboratory conducted in January 2021

| Ampoule number | CFC-12 | HCFC-22 | HCFC-141b | HCFC-142b | HFC-134a |
|---|---|---|---|---|---|
| 310 | 1.39 | 20.13 | 0.99 | 0.78 | 1163.50 |
| 105 | 1.50 | 20.90 | 1.04 | 0.84 | 1305.40 |
| 134 | 1.45 | 20.44 | 0.96 | 0.80 | 1205.50 |
| 372 | 1.51 | 21.02 | 1.03 | 0.84 | 1324.60 |
| Average | 1.46 | 20.62 | 1.00 | 0.81 | 1249.75 |
| Precision (%) | 3.7 | 2.0 | 3.5 | 3.4 | 6.2 |
| Expected conc. [a] | 1.40 | 9.81 | 0.78 | 0.35 | 2.03 |

[a] The expected concentrations of tracers are calculated based on their extrapolated atmospheric mole fractions in the Northern Hemisphere background sites in the year 2021.125 (only CFC-12 in the year 2021.5) and seawater solubility functions (Li et al., 2019) by setting salinity 34.4, temperature 17 °C and pressure 5 dbar, although the seawater samples were contaminated after equilibrated with the laboratory air.

P. 9-10 – If the 1-G TTD was representative of the processes controlling the distributions of the tracers in the Mediterranean Sea, all of the data plotted in the panels of Fig. 10 should fall along the same Δ/Γ line. Even the SF6 - CFC-12 data fall into the region of Δ/Γ>1.8.

Response: We were aware that IG-TTD is not so suitable to constrain ventilation in the Mediterranean Sea (Stöven and Tanhua, 2014). There are two reasons for this as described in that

paper: 1) It is most likely a mix of two water masses so that a "2IG-TTD" would be needed to describe the TTD, and 2) the time-variant ventilation of the Mediterranean Sea that invalidated the TTD concept. Indeed, we were not able to fit the $SF_6$ and CFC-12 measurements to the IG-TTD successfully from that cruise. However, as the first step, we would like to see the results based on different $\Delta/\Gamma$ ratios under IG-TTD for various tracer pairs. We do expect that the tracer pairs should fall within, or at least, close to the "validity area", just as the $SF_6$/CFC-12 pair.

We realize that the actively ventilated Mediterranean Sea is interesting from a ventilation perspective, see for instance Li and Tanhua (2020), and that we can expect to find these tracers through the water column. But that the time-variant ventilation and mixing of two or three major water masses with different ages and $\Delta/\Gamma$ makes the constraint difficult. So there are pros and cons of this area.

I agree with some of what the authors conclude about the feasibility of using these tracers in future oceanographic studies. The real issue for me is whether the ampoule sampling combined with the Aqua-Medusa system is capable of being utilized for these measurements. This is where I would focus my efforts in the future.

Response: To answer this question, we took seawater from the tropical Atlantic Ocean, and let it equilibrate with the atmosphere in the laboratory. The water was then sampled from Niskin bottles in the same way as during a cruise and flame sealed in ampoules, although for this experiment we used 300 mL samples. The reproducibility for CFC-12, HCFC-22, HCFC-141b, and HCFC-142b measurements from four duplicate samples are 3.7%, 2.0%, 3.5%, and 3.4%, respectively. The added results validate the feasibility of the analytical method of ampoule sampling combined with the Aqua-Medusa system and make our results more convincing.

**References**

Li, P., Mühle, J., Montzka, S. A., Oram, D. E., Miller, B. R., Weiss, R. F., Fraser, P. J., and Tanhua, T.: Atmospheric histories, growth rates and solubilities in seawater and other natural waters of the potential transient tracers HCFC-22, HCFC-141b, HCFC-142b, HFC-134a, HFC-125, HFC-23, PFC-14 and PFC-116, Ocean Sci., 15, 33–60, https://doi.org/10.5194/os-15-33-2019, 2019.

Li, P., and Tanhua, T.: Recent changes in deep ventilation of the Mediterranean Sea; evidence from long-term transient tracer observations, Frontiers in Marine Science, 7, 594, https://doi.org/10.3389/fmars.2020.00594, 2020.

Stöven, T., and Tanhua, T.: Ventilation of the Mediterranean Sea constrained by multiple transient tracer measurements, Ocean Sci., 10, 439–457, https://doi.org/10.5194/os-10-439-2014, 2014.